# Crumbs organizes the transport machinery by regulating apical levels of PI(4,5)P$_2$ in *Drosophila*

Johanna Lattner[1], Weihua Leng[1], Elisabeth Knust[1], Marko Brankatschk[2]*, David Flores-Benitez[1]*

[1]Max-Planck Institute of Molecular Cell Biology and Genetics (MPI-CBG), Dresden, Germany; [2]The Biotechnological Center of the TU Dresden (BIOTEC), Dresden, Germany

**Abstract** An efficient vectorial intracellular transport machinery depends on a well-established apico-basal polarity and is a prerequisite for the function of secretory epithelia. Despite extensive knowledge on individual trafficking pathways, little is known about the mechanisms coordinating their temporal and spatial regulation. Here, we report that the polarity protein Crumbs is essential for apical plasma membrane phospholipid-homeostasis and efficient apical secretion. Through recruiting β$_{Heavy}$-Spectrin and MyosinV to the apical membrane, Crumbs maintains the Rab6-, Rab11- and Rab30-dependent trafficking and regulates the lipid phosphatases Pten and Ocrl. Crumbs knock-down results in increased apical levels of PI(4,5)P$_2$ and formation of a novel, Moesin- and PI(4,5)P$_2$-enriched apical membrane sac containing microvilli-like structures. Our results identify Crumbs as an essential hub required to maintain the organization of the apical membrane and the physiological activity of the larval salivary gland.

## Introduction

**\*For correspondence:**
marko.brankatschk@tu-dresden.de (MB);
flores@mpi-cbg.de (DF-B)

Epithelia can organize as layers or tubes, which form barriers and thus separate internal biological compartments from the environment. Many epithelia are specialized for absorption or secretion by performing selective and directional transport of nutrients, enzymes and waste products, which is essential for metazoan life (*Cereijido et al., 2004*; *Rodriguez-Boulan and Macara, 2014*; *Lemaitre and Miguel-Aliaga, 2013*). To perform these functions, epithelial cells are highly polarized: plasma membrane proteins and lipids are distributed asymmetrically into an apical domain facing the environment or a lumen, and a basolateral domain that contacts the neighboring cell and/or a basal lamina. In addition, polarity is manifested by uneven distribution of organelles, asymmetric cytoskeleton organization and directed trafficking (*Rodriguez-Boulan and Macara, 2014*; *Knust and Bossinger, 2002*; *Eaton and Martin-Belmonte, 2014*). The latter is particularly obvious in secretory epithelia, for example the salivary glands, which produce vast amounts of material that is secreted into the gland lumen (*Blasky et al., 2015*; *Iruela-Arispe and Beitel, 2013*; *Eaton and Martin-Belmonte, 2014*; *Chung et al., 2014*; *Miguel-Aliaga et al., 2018*).

Several evolutionarily conserved proteins regulate epithelial cell polarity. These include members of the apical Crumbs- and PAR-complexes, and the basolateral Scrib-Dlg-Lgl module (reviewed in *Flores-Benitez and Knust, 2016*; *Román-Fernández and Bryant, 2016*). The Crumbs (Crb) protein has a large extracellular domain (>2000 aa), and a small intracellular domain (37 aa) (*Tepass et al., 1990*; *Wodarz et al., 1993*), which harbors two protein-protein interaction motifs, a C-terminal PDZ (**P**ostsynaptic density/**D**iscs large/**Z**O-1)-domain binding motif (PBM) and a juxtamembrane FERM (protein **4**.1/**e**zrin/**r**adixin/**m**oesin)-domain binding motif (FBM). The PBM is important for cell polarity and can bind Stardust (Sdt) and Par-6 (*Li et al., 2014*; *Roh et al., 2002*; *Bulgakova et al., 2008*;

*Bachmann et al., 2001*; *Hong et al., 2001*; *Kempkens et al., 2006*; *Ivanova et al., 2015*). The FBM can directly interact with Yurt (Yrt), Expanded (Ex) and Moesin (Moe) (*Klebes and Knust, 2000*; *Laprise et al., 2006*; *Ling et al., 2010*; *Wei et al., 2015*), FERM-proteins that act as adaptors between membrane proteins and the actin cytoskeleton (*Bennett and Baines, 2001*; *Lemmon et al., 2002*; *McClatchey, 2014*; *Sauvanet et al., 2015*). The FBM of Crb is also important for $\beta_{Heavy}$-Spectrin ($\beta_H$-Spec) recruitment to the apical plasma membrane, and thereby supports the polarized organization of the membrane-associated cytoskeleton (cytocortex) (*Wodarz et al., 1995*; *Richard et al., 2009*; *Pellikka et al., 2002*; *Lee et al., 2010*; *Lee and Thomas, 2011*; *Médina et al., 2002b*).

Several epithelia of *crb* or *sdt* mutant *Drosophila* embryos show severe polarity defects, disruption of cell-cell adhesion and loss of tissue integrity. On the other hand, over-expression of Crb in the embryonic epidermis increases the size of the apical membrane (*Tepass and Knust, 1993*; *Grawe et al., 1996*; *Tepass et al., 1990*; *Das and Knust, 2018*; *Tepaß and Knust, 1990*). Similar phenotypes have been reported in mouse embryos mutant for *Crb2* or *Crb3* (*Charrier et al., 2016*; *Szymaniak et al., 2015*; *Whiteman et al., 2014*; *Xiao et al., 2011*; *Ramkumar et al., 2016*). In addition, *Drosophila* Crb has been associated with other functions, which are independent of its roles in epithelial integrity, such as regulation of tissue growth via the Hippo pathway, regulation of Notch signaling (*Das and Knust, 2018*; *Nemetschke and Knust, 2016*; *Perez-Mockus et al., 2017*; *Herranz et al., 2006*), as well as photoreceptor morphogenesis and survival under light stress (reviewed in *Pocha and Knust, 2013*; *Bulgakova and Knust, 2009*; *Genevet and Tapon, 2011*).

Apico-basal polarity is also essential for polarized membrane traffic. Directed trafficking depends on the phosphoinositide composition of the plasma membrane, the cytocortex and various Rab (**Ra**s-related in **b**rain) proteins. All of these are closely interconnected to organize and maintain the identity of apical and basolateral membranes (*Weisz and Rodriguez-Boulan, 2009*; *Eaton and Martin-Belmonte, 2014*; *Blasky et al., 2015*; *Rodriguez-Boulan et al., 2005*; *Croisé et al., 2014*). Epithelial cell polarity and polarized membrane traffic require differential enrichment of phosphatidylinositol 4,5-bisphosphate (PI(4,5)$P_2$) and phosphatidylinositol 3,4,5-trisphosphate (PI(3,4,5)$P_3$) in the apical and basolateral membranes, respectively (*Di Paolo and De Camilli, 2006*; *Martin-Belmonte and Mostov, 2007*). PI(4,5)$P_2$ levels are controlled by Pten (Phosphatase and tensin homolog deleted on chromosome ten), which converts PI(3,4,5)$P_3$ into PI(4,5)$P_2$, by the type I phosphatidylinositol 4-phosphate 5-kinase Skittles (Sktl), which produces PI(4,5)$P_2$ from phosphatidylinositol 4-phosphate (PI4P), and by Ocrl (Oculocerebrorenal syndrome of Lowe), which dephosphorylates PI(4,5)$P_2$ into PI4P (*de Renzis et al., 2002*; *Knirr et al., 1997*; *Maehama et al., 2004*; *Claret et al., 2014*; *Gervais et al., 2008*; *Worby and Dixon, 2014*; *Balakrishnan et al., 2015*; *Weixel et al., 2005*). Pten activity is antagonistic to that of the type IA phosphatidylinositol three kinase (Pi3K), which is enriched at basolateral membranes and converts PI(4,5)$P_2$ into PI(3,4,5)$P_3$ (*Gassama-Diagne et al., 2006*; *Peng et al., 2015*; *Balakrishnan et al., 2015*; *Gao et al., 2000*; *Goberdhan et al., 1999*; *Huang et al., 1999*). PI(4,5)$P_2$ can bind to pleckstrin homology (PH)-domains of FERM proteins and β-Spectrins (*Yoon et al., 1994*; *Harlan et al., 1995*), thereby linking the plasma membrane to the cytocortex and to the trafficking machinery (*Barroso-González et al., 2009*; *Ramel et al., 2013*; *Beck and Nelson, 1998*; *Holleran and Holzbaur, 1998*; *Kang et al., 2009*). Moreover, PI(4,5)$P_2$ is directly implicated in the regulation of exocytosis (*Milosevic et al., 2005*; *Gong et al., 2005*; *Massarwa et al., 2009*; *Rousso et al., 2013*) and in all forms of endocytosis (*Antonescu et al., 2011*; *Mayinger, 2012*; *Jost et al., 1998*).

Here, we studied the functions of Crb in a differentiated, highly polarized secretory epithelium, namely the salivary gland (SG) of the *Drosophila* larva, to decipher its possible role in polarized trafficking. We identified Crb as a novel regulator of apical secretion and maintenance of the apical microvilli in SG cells. We show that loss of Crb in SGs disrupts the apical cytocortex, apical secretion and the apical trafficking machinery, including the organization of Rab6-, Rab11- and Rab30-positive apical compartments, and the localization of their effector Myosin V (MyoV) (*Lindsay et al., 2013*). Our results show that Crb controls the apical secretion machinery via regulation of phosphoinositide metabolism. Loss of Crb increases apical levels of PI(4,5)$P_2$, a phenotype that requires the activity of Pten, and impairs the function of the apical secretory machinery. These defects are accompanied by the formation of a novel apical membrane compartment, which emerges as a solitary intracellular sac of PI(4,5)$P_2$- and phospho-Moe-enriched apical membrane containing microvilli. This compartment is reminiscent to intracellular vacuolar structures found in patients with MVID (microvillus

inclusion disease), a fatal genetic disease characterized by lack of microvilli on the surface of entero-cytes (www.omim.org/entry/251850). We conclude that Crb acts as an apical hub to couple phospholipid metabolism and cytoskeleton scaffolds with apical membrane traffic. Our work sheds light on the mechanism behind the determination of the apical membrane by Crb and its possible implications in different pathologies.

## Results

### The Crb complex is dispensable for maintenance of apico-basal polarity in larval salivary glands (SGs)

To investigate the role of the Crb protein complex in a differentiated secretory epithelium, we silenced Crb or its binding partner Sdt in the larval SG by RNAi-mediated knock-down (KD) using the SG-specific driver *fkh*-GAL4 (*Zhou et al., 2001*). We took advantage of the fact that this strategy does not affect embryonic development (data not shown). The larval SG consists of two tubes composed of columnar epithelial cells, each with a central lumen (*Figure 1A*). Strikingly, although the KD of Crb effectively reduces apical levels of Crb, Sdt and *D*Patj (*Figure 1B–C'* and *Figure 1—figure supplement 1C,D,Q*), it does not affect the overall morphology of SGs, as determined by phalloidin staining (*Figure 1—figure supplement 1A,B*). Yet, the SGs lacking Crb are shorter when compared to their control counterparts (*Figure 1—figure supplement 1R*, *Figure 1—figure supplement 1—source data 1*). Similar results were observed upon RNAi-mediated KD of Sdt (*Figure 1—figure supplement 1R–X*). Interestingly, KD of Crb or Sdt does not alter the polarized distribution of any canonical apical or basolateral polarity marker tested, including Bazooka (Baz, *Figure 1D,E*), aPKC (*Figure 1—figure supplement 1E,F*), Par-6 (*Figure 1—figure supplement 1G,H and Y,Z*), Disc large (Dlg, *Figure 1F,G* and *Figure 1—figure supplement 1AA, BB*), Yurt (Yrt) (*Figure 1—figure supplement 1I,J*) and Coracle (Cora, *Figure 1—figure supplement 1K,L*). Taken together, these results show that the Crb protein complex is dispensable for maintenance of tissue integrity and overall epithelial cell polarity of larval SGs.

### The Crb protein complex is required for proper apical secretion in larval SGs

Because depletion of the Crb protein complex does not affect the overall polarity or integrity of the larval SGs, we analyzed whether it plays any role in maintaining their physiological functions. SGs of feeding larvae produce saliva required to digest food, whereas in later stages they produce and secrete predominantly glue proteins required to attach the pupae to surfaces (*Thomopoulos, 1988*; *Chung et al., 2014*; *Maruyama and Andrew, 2012*; *Csizmadia et al., 2018*; *Gregg et al., 1990*; *Fraenkel and Brookes, 1953*). Thus, we speculated that any defect in saliva secretion could result in less food intake and hence delayed larval development. In fact, when compared to control larvae, the time necessary to reach the pupal stage is prolonged upon depletion of Crb (*Figure 1H*, *Figure 1—source data 1*) or Sdt (*Figure 1—figure supplement 1KK*, *Figure 1—figure supplement 1—source data 2*).

To test whether the delay in pupation correlates with defects in apical membrane transport, we analyzed the localization of Cadherin99C (Cad99C), an apical transmembrane protein involved in regulation of microvillar length (*Chung and Andrew, 2014*), and CD8-RFP, a heterologous transmembrane protein normally targeted to the apical membrane (*Xu et al., 2002*; *Lee and Luo, 1999*). We found that upon silencing of Crb or Sdt, Cad99C and CD8-RFP do not localize properly at the apical membrane but instead localize in intracellular vesicles (*Figure 1I,J* and *Figure 1—figure supplement 1CC,DD,M,N and EE,FF*).

To evaluate apical secretion, we analyzed the expression of Sgs3-GFP. However, the glue proteins are not expressed at the feeding stage we study here (beginning of the 3$^{rd}$ instar) but almost 2 days later (*Tran and Ten Hagen, 2017*). Indeed, at the stage of glue secretion, vesicle delivery appears normal in Crb-deficient SGs (*Videos 1* and *2*) (*Tran et al., 2015*). Furthermore, several proteins that are known to be apically secreted in other tubular epithelia, like Piopio, Vermiform and UAS-driven secreted proteins (cherry-sec, GFP-tagged wheat germ agglutinin) (*Jaźwińska et al., 2003*; *Luschnig et al., 2006*; *Brankatschk and Eaton, 2010*) were not suitable for our studies since they could not be detected in the lumen of wild-type feeding larval SGs (not shown). Therefore, we

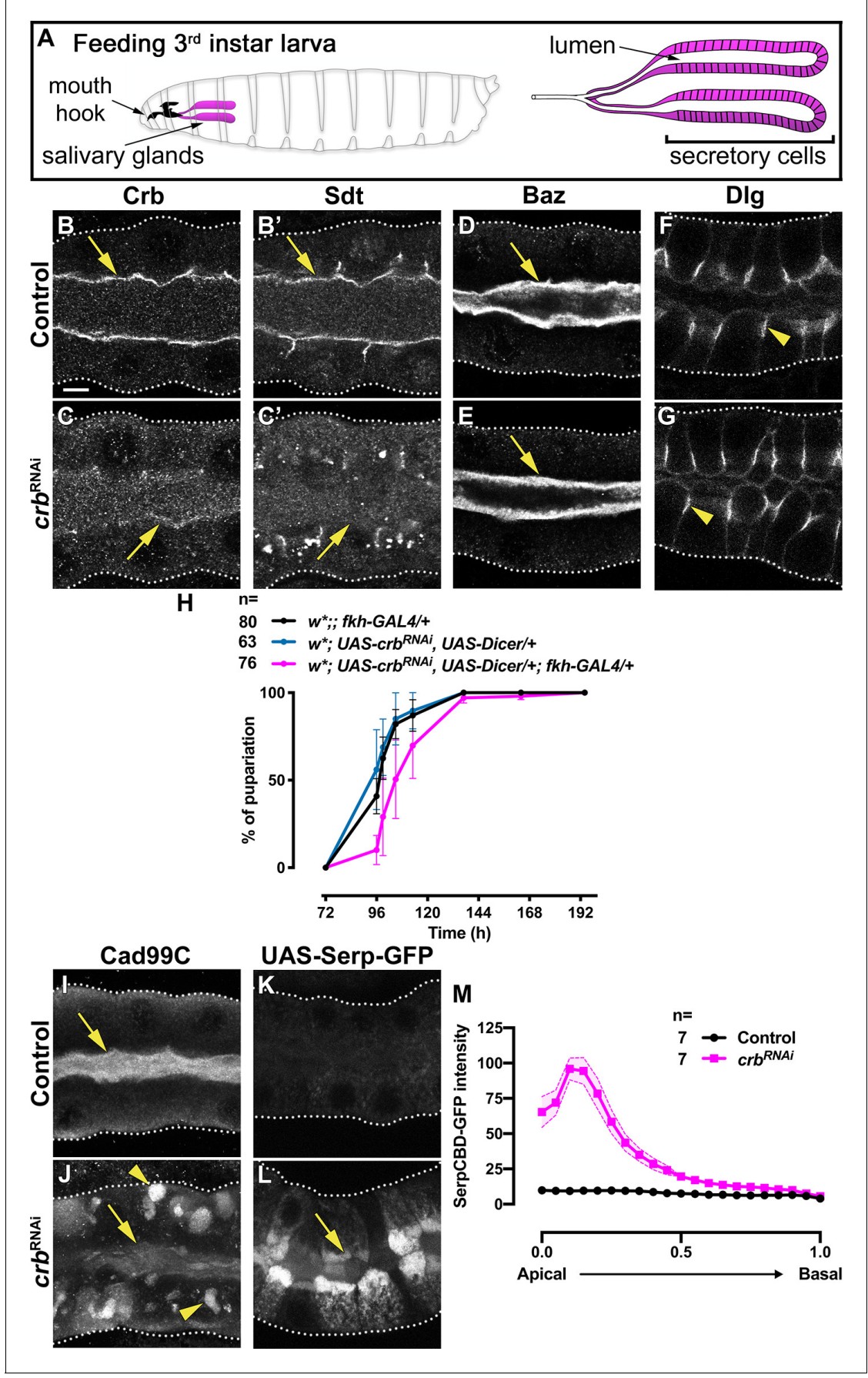

**Figure 1.** Crb is required for efficient apical secretion in SG cells. (**A**) Scheme indicating the anatomic location of the SG in the larval stage. (**B-G**) Localization of Crb (**B,C**), Sdt (**B',C'**), Baz (**D,E**) and Dlg (**F,G**) in control (**B,B',D,F**, *fkh>/+*) and Crb KD (**C,C',E,G**, *fkh >UAS crb^RNAi*) animals. H. Pupariation efficiency of controls (black and blue) and larvae with reduced levels of Crb (magenta) at 29 ˚C. Error bars indicate the standard error of the mean, n indicates number of traced individual larvae of the corresponding genotypes in three independent experiments. (**I,J**) Localization of the apical transmembrane protein Cadherin99C in SGs from control (**I**) and Crb KD (**J**) animals. (**K,L**) Localization of the secreted apical cargo SerpCBD-GFP in live SGs of control (**K**, *fkh >UAS SerpCBD-GFP*) and Crb KD (**L**, *fkh >UAS crb^RNAi; UAS-SerpCBD-GFP*) animals. Arrows indicate the apical plasma membrane. Arrowheads mark the lateral plasma domain. Dotted lines indicate the basal membrane. Scale bar in A indicates10 µm applies to all panels. (**M, M**) Plotted is the fluorescence intensity (arbitrary units) of SerpCBD-GFP along the apical-to-basal direction in live SGs of control (black, *fkh >UAS SerpCBD-GFP*) and Crb KD (magenta, *fkh >UAS crb^RNAi; UAS-SerpCBD-GFP*). Error bars indicate the standard error of the mean, n indicates number of glands from the corresponding genotypes.

The online version of this article includes the following source data and figure supplement(s) for figure 1:

**Source data 1.** Dataset for tracking of larval development.
**Source data 2.** Dataset for SerpCBD-GFP fluorescence intensity in control glands.
**Source data 3.** Dataset for SerpCBD-GFP fluorescence intensity in Crb KD glands.
**Figure supplement 1.** Knock-down of the Crb protein complex in larval SGs disrupts apical secretion.
**Figure supplement 1—source data 1.** Dataset for salivary gland lengths.
**Figure supplement 1—source data 2.** Dataset for tracking of larval development.
**Figure supplement 1—source data 3.** Dataset for SerpCBD-GFP fluorescence intensity in control glands (note is the same dataset for *Figure 1M* control).
**Figure supplement 1—source data 4.** Dataset for SerpCBD-GFP fluorescence intensity in Sdt KD glands.
**Figure supplement 2.** The Crb protein complex is dispensable for maintenance of cell-cell junctions in larval SGs.

used the chitin-binding domain of Serpentine tagged with GFP (UAS-SerpCBD-GFP), a well-established marker to evaluate apical secretion (*Luschnig et al., 2006*; *Kakihara et al., 2008*; *Förster et al., 2010*; *Petkau et al., 2012*; *Dong et al., 2013*; *Dong et al., 2014*; *Bätz et al., 2014*). Notably, while SerpCBD-GFP is barely detectable upon overexpression in control glands, loss of Crb or Sdt results in an obvious intracellular retention of SerpCBD-GFP at the apical aspect (*Figure 1K–M*, *Figure 1—source data 2* and *3*; and *Figure 1—figure supplement 1 GG, HH, and LL*, *Figure 1—figure supplement 1—source data 3* and *4*). In support of the idea that Crb is necessary for efficient apical secretion, we also found that glycoprotein secretion is impaired upon loss of Crb or Sdt, as revealed by intracellular retention of peanut-agglutinin-GFP (PNA-GFP, *Figure 1—figure supplement 1O,P and II,JJ*), which can bind to glycoproteins produced by the SGs (*Korayem et al., 2004*; *Theopold et al., 2001*; *Tian and Ten Hagen, 2007*). Taken together, these results show that the Crb protein complex is required for proper apical membrane protein delivery and protein secretion in SGs of feeding larvae.

## The Crb protein complex is dispensable for maintenance of cell-cell junctions in larval SGs

Impaired apical secretion after KD of Crb could be related to defects in cell-cell junctions. In particular, the pleated septate junctions (pSJs) are involved in apical secretion in the embryonic tracheae (*Wang et al., 2006*; *Laprise et al., 2010*; *Nelson et al., 2010*). Therefore, we examined the SGs by transmission electron microscopy (TEM). We did not find any abnormalities in the localization of the *zonula adherens* (ZA) of Crb-deficient SG cells (*Figure 1—figure supplement 2A',C'* arrowheads).

In contrast to ZA, pSJs are morphologically abnormal in SGs of Crb KD animals, showing many interruptions (*Figure 1—figure supplement 2C*, green highlight) and disorganized regions (*Figure 1—figure supplement 2D*). In contrast, control SG cells, pSJs run uniformly along the lateral membrane with few interruptions (*Figure 1—figure supplement 2A,B*). Defects in pSJs were corroborated by reduced immunostaining of some pSJ components, including Sinuous (Sinu, *Figure 1—figure supplement 2E,F*), Kune-kune (Kune, *Figure 1—figure supplement 2G,H*), while others, such as Fasciclin3 (Fas3, *Figure 1—figure supplement 2I,J*), Dlg (*Figure 1F,G*), Lachesin-GFP and Nervana2-GFP (not shown) were not affected. Given the defects observed in pSJs, we analyzed their permeability by monitoring any luminal appearance of fluorescently labeled 10 kDa-Dextran ex vivo (*Lamb et al., 1998*). Interestingly, KD of Crb does not increase dye penetration into the lumen when compared to control glands (*Figure 1—figure supplement 2K–L'*), suggesting that the epithelium is tight. In contrast, KD of Fas3-GFP, used as a positive control, enhances the diffusion of 10 kDa-Dextran into the gland lumen (*Figure 1—figure supplement 2M,M'*).

## Sgs3-GFP

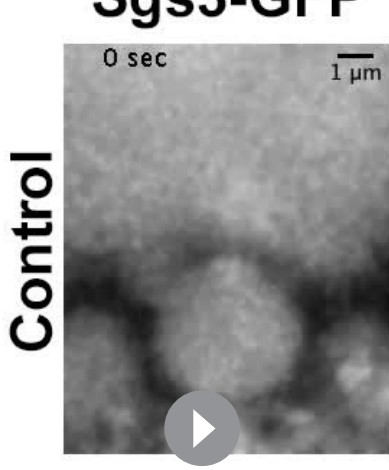

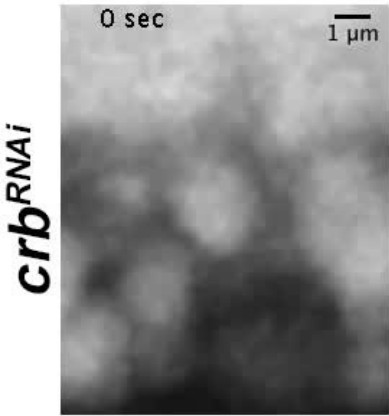

**Video 1.** Fusion of a glue vesicle followed by expulsion of the cargo Sgs3-GFP into the lumen SG lumen of control (*fkh>+*, top) and Crb KD (*fkh >UAS crb*[RNAi], bottom) animals.
https://elifesciences.org/articles/50900#video1

## Sgs3-GFP

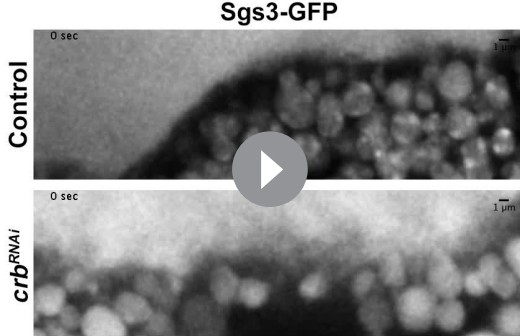

**Video 2.** Overview showing the fusion of glue vesicles followed by expulsion of Sgs3-GFP into the SG lumen of control (*fkh>+*, top) and Crb KD (*fkh >UAS crb*[RNAi], bottom) animals. Note that the increase of fluorescence in the vesicle occurs when they open to the lumen.
https://elifesciences.org/articles/50900#video2

Taken together, these results suggest that loss of Crb does not affect adherens junctions or the epithelial barrier function of SGs.

## Crb regulates apical membrane organization via the apical cytocortex

Crb recruits Moesin (Moe) and $\beta_H$-Spectrin ($\beta_H$-Spec, encoded by the gene *karst -kst*) to the apical membrane (*Richard et al., 2009*; *Lee et al., 2010*; *Lee and Thomas, 2011*; *Médina et al., 2002b*; *Kerman et al., 2008*), where they mediate interactions between transmembrane proteins and the apical cytocortex (reviewed in *Fehon et al., 2010*; *Baines et al., 2014*). Therefore, we analyzed whether Crb KD affects the organization of the apical cytocortex in SG cells, and if so, whether this relates to the defects in apical secretion.

We found that KD of Crb decreases apical levels of F-actin (*Figure 2A–C*, *Figure 2—source data 1*) and $\beta_H$-Spec (*Figure 2D–F*, *Figure 2—source data 2*). Similarly, silencing a knock-in Crb tagged with GFP on the extracellular domain, Crb-GFP-A (*Huang et al., 2009*), using *fkh >gfp*[RNAi] as an alternative approach for the KD of the Crb protein complex (*Figure 2—figure supplement 1A–D*) also decreases apical levels of F-actin (*Figure 2—figure supplement 1E,F*). Moreover, KD of Crb-GFP-A induces accumulation of Moe, as well as its active form phospho-Moe, into a single sac per cell localized right below the apical domain (*Figure 2G,H*, arrows, *Video 3* and not shown). These sacs are also positive for the apical transmembrane protein Stranded at second tagged with YFP (*Firmino et al., 2013*) (Sas-YFP, *Figure 2I,J*) suggesting that they have an apical plasma membrane identity. On the other hand, KD of Crb has no evident effects on the organization of $\alpha$-Tubulin or $\alpha$-Spectrin (*Figure 2—figure supplement 1G–J*). These results show that Crb is required to maintain the organization of the apical cytocortex and the morphology of the apical membrane in larval SGs.

To examine in more detail the morphology of the apical aspect of Crb-deficient cells, we prepared SGs for TEM analysis by employing the high-pressure freezing technique. This technique immobilizes complex macromolecular assemblies in their native state and helps to preserve cytoskeleton-rich structures like microvilli (*Studer et al., 2008*). Strikingly, cells from Crb-depleted SGs display intracellular vesicles containing microvilli (*Figure 2L'*, arrowheads and *Figure 2—figure*

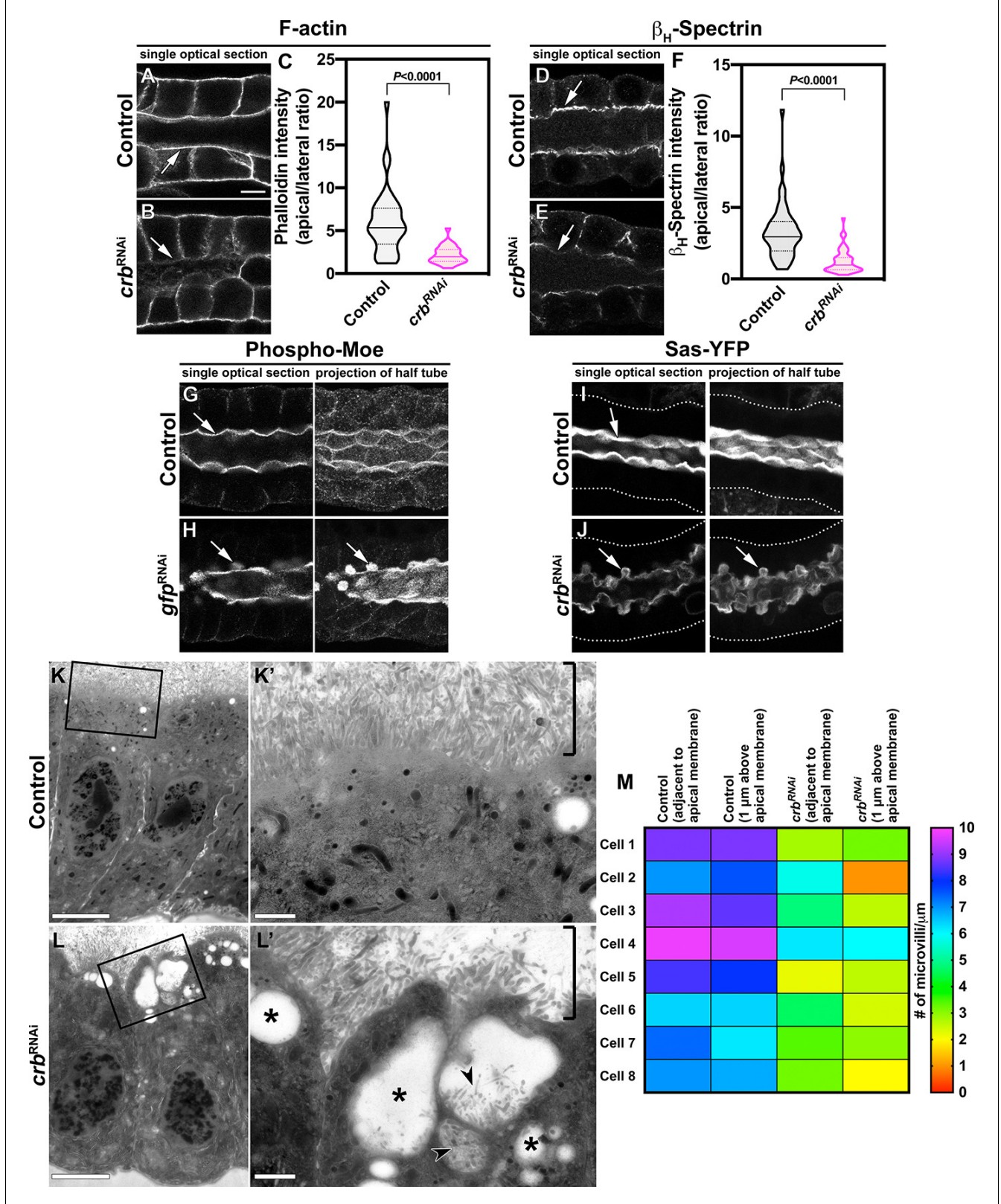

**Figure 2.** Crb is necessary to specifically maintain the apical cytoskeleton and the morphology of the apical membrane. (A-F) Localization and quantification of F-actin (phalloidin staining, **A-C**) and β$_H$-Spec (**D-F**) in control (**A,D**, *fkh>/+*) and Crb KD (**B,E**, *fkh >UAS crb$^{RNAi}$*) SGs. Violin graphs (**C,F**) show the fluorescence intensity (apical vs lateral ratio) indicating the mean and quartiles for F-actin (**C**, n = 36 cells for control and 28 cells for Crb KD)
*Figure 2 continued on next page*

*Figure 2 continued*

and β$_H$-Spec (F, n = 44 cells for control and 40 cells for Crb KD). Statistical significance was analyzed in an unpaired two-tailed *t*-test. (G-H) Localization of phospho-Moe in control (G, *Crb-GFP, fkh>/+*) and Crb KD (H, *Crb-GFP, fkh >UAS gfp$^{RNAi}$*) SGs. (I,J) Localization of the apical protein Stranded at second (Sas-YFP) in live SGs of control (I, *fkh>/+*) and Crb KD (J, *Crb-GFP, fkh >UAS gfp$^{RNAi}$*) animals. Shown are single optical slices and maximal projections of half of the z-stack (half SG-tube). Arrows point to the apical domain of the cell. Dotted lines indicate the basal membrane. Scale bar in (A) displays 10 μm and applies to panes (A-J). (K-L') TEM images of SGs prepared using the high-pressure freezing technique, visualizing the apical aspect of SG cells of control (K,K', *fkh>/+*) and Crb KD (L,L', *fkh >UAS crb$^{RNAi}$*) animals. The brackets in K,L' indicate the apical microvilli. Asterisks in (L') mark large intracellular vesicles found in Crb-deficient glands. Arrowheads in L' indicate microvilli found inside vesicles. Scale bars in (K,L) indicate 5 μm and in (K',L') indicate 1 μm. (M, M) Mean number of microvilli following along the apical membrane over a distance of 1 μm, adjacent to the membrane and 1 μm above the apical membrane in SG cells of control (*fkh>/+*) and Crb KD (*fkh >UAS crb$^{RNAi}$*) animals. The heatmap indicates the scale bar for the number of microvilli/μm.

The online version of this article includes the following source data and figure supplement(s) for figure 2:

**Source data 1.** Dataset for phalloidin fluorescence intensity.
**Source data 2.** Dataset for β$_H$-Spec fluorescence intensity.
**Source data 3.** Dataset for microvilli quantifications.
**Figure supplement 1.** Crb is necessary to specifically maintain the apical membrane organization.
**Figure supplement 2.** TEM images of intracellular extensions of apical membrane in Crb-deficient glands.
**Figure supplement 3.** Increased lysosomal activity in Crb and Sdt deficient glands.
**Figure supplement 4.** KD of β$_H$-Spec induces the formation of PAMS.

supplement 2), which seem to correspond to the Sas-YFP positive sacs described above (*Figure 2J*). In fact, we also observed cases of intracellular sacs whose membrane were continuous with the apical membrane (*Figure 2—figure supplement 2A,A'*). Moreover, in Crb-deficient SGs, the density of apical microvilli is dramatically reduced (*Figure 2K',L'*, brackets, and M, *Figure 2—source data 3*). The number of microvilli per micron adjacent to the apical plasma membrane is 8.0 ± 1.219 in control vs. 4.125 ± 1.446 in Crb-deficient cells (mean ± SD, p<0.0001, n = 8). This difference is even bigger when measured at 1 μm above the plasma membrane, 7.75 ± 1.222 in control vs. 2.850 ± 1.441 in Crb-deficient cells (mean ± SD, p<0.0001, n = 8), indicating that microvilli are also shorter in Crb-deficient cells. In addition, Crb-deficient SG cells exhibit large intracellular vesicles not present in control SGs, which probably correspond to enlarged lysosomes (asterisks in *Figure 2L'* and in *Figure 2—figure supplement 2B'C'*; see also *Figure 1—figure supplement 2C* blue highlight). Indeed, live imaging of SGs incubated with Lysotracker showed that KD of Crb or Sdt increases lysosomal activity (*Figure 2—figure supplement 3*). This suggests that lysosomal activity increases due to impaired secretion upon loss of Crb.

Since these apical membrane invaginations are enriched in PI(4,5)P$_2$ (described below), we refer to them as PAMS: **p**hospho-Moe and PI(4,5)P$_2$-enriched **a**pical **m**embrane **s**acs. Given that silencing of Crb reduces apical β$_H$-Spec, we analyzed the effect of β$_H$-Spec KD on PAMS formation. Indeed, loss of β$_H$-Spec (*Figure 2—figure supplement 4A,B*) prompts formation of PAMS marked by phospho-Moe (*Figure 2—figure supplement 4C,D*). Moreover, in SGs deficient in β$_H$-Spec, Crb remains apical and additionally localizes to the PAMS (*Figure 2—figure supplement 4E,F*). These results indicate that Crb localizes to the apical domain independently of β$_H$-Spec while β$_H$-Spec requires Crb to be organized at the apical cytocortex.

Taken together, these results indicate that Crb is essential to maintain the proper amount and

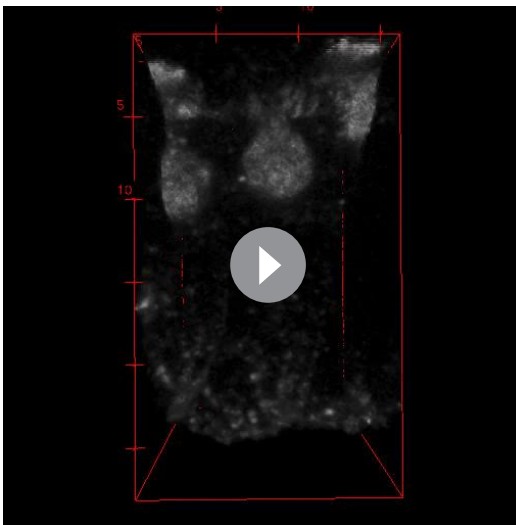

**Video 3.** 3D rendering of a SG from a Crb KD animal (*fkh >UAS crb$^{RNAi}$*) probed for phospho-Moesin. The extraction focuses on one cell to appreciate the accumulation of phospho-Moesin at the apical membrane. Apical is up.
https://elifesciences.org/articles/50900#video3

organization of the apical membrane by stabilizing the apical cytocortex.

## Crb regulates the apical membrane organization via MyosinV

The PAMS described above are reminiscent to microvilli-containing vesicles found in samples from MVID (microvillus inclusion disease) patients, which is linked to mutations in the *MYO5b* gene (*Müller et al., 2008*). Similar inclusions are found in animal models of MVID (*Sidhaye et al., 2016*). MyosinV (MyoV) is a processive motor that transports cargos along F-actin (*Reck-Peterson et al., 2000*) and is a component of the apical secretory machinery in epithelia (*Massarwa et al., 2009*; *Reck-Peterson et al., 2000*; *Li et al., 2007*; *Pocha et al., 2011a*).

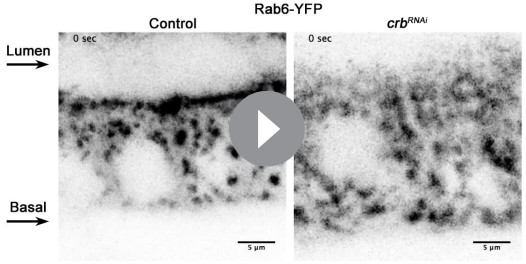

**Video 4.** Live imaging of endogenously expressed Rab6-YFP in SGs of control (left, *Rab6-YFP, fkh>/+*) and Crb KD (right, *Rab6-YFP, fkh >UAS crb^RNAi^*). 5 min recording, time lapse 5 s.
https://elifesciences.org/articles/50900#video4

Moreover, in photoreceptor cells, Crb regulates apical transport of Rhodopsin-1 by interacting with MyoV (encoded by the gene *didum*) (*Pocha et al., 2011a*). Therefore, we analyzed whether Crb regulates MyoV in the SGs. Indeed, the KD of Crb decreases apical MyoV (*Figure 3A,B,D*, *Figure 3— source data 1* and *2*). Importantly, overexpression of MyoV-GFP in Crb-deficient glands does not rescue its apical localization (*Figure 3—figure supplement 1A–C*, *Figure 3—figure supplement 1—source data 1* and *2*). Furthermore, KD of $\beta_H$-Spec also decreases apical MyoV (*Figure 3A,C,D*, *Figure 3—source data 1* and *3*) as well as apical secretion as revealed by the apical retention of SerpCBD-GFP (*Figure 3—figure supplement 2A–C*, *Figure 3—figure supplement 2—source data 1* and *2*). This suggests that $\beta_H$-Spec acts downstream of Crb to maintain apical MyoV.

To examine the role of MyoV in apical secretion and PAMS formation, we silenced MyoV expression in the SGs using a specific RNAi (*didum^RNAi^*). Analysis of Crb, phospho-Moe and Sas-YFP in MyoV-deficient SGs shows that while these proteins localize apically, they are also found in PAMS (*Figure 3E–J*). Additionally, live imaging of SGs expressing Sas-YFP shows large vesicles inside the cell (*Figure 3J*, arrowhead), which resemble similar structures seen in an organoid model for MVID established from mouse intestinal cells with impaired apical transport (*Mosa et al., 2018*). Indeed, we found that MyoV KD impairs secretion of SerpCBD-GFP, which in turn accumulates at the apical aspect of MyoV-deficient SG cells (*Figure 3K–M*, *Figure 3—source data 4* and *5*). These results suggest that formation of PAMS can be a consequence of defects in the apical secretory machinery.

Together, our results indicate that loss of Crb disrupts the apical $\beta_H$-Spec cytocortex. As a consequence, the apical localization of MyoV is reduced, apical secretion is impaired, and apical membrane morphology is defective, resulting in PAMS formation.

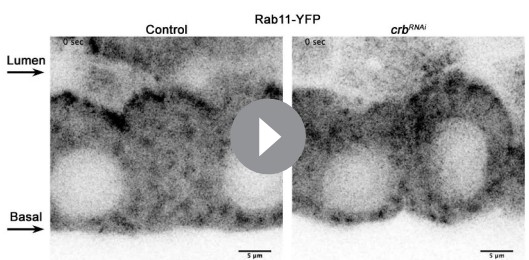

**Video 5.** Live imaging of endogenously expressed Rab11-YFP in SGs of control (left, *Rab11-YFP, fkh>/+*) and Crb KD (right, *Rab11-YFP, fkh >UAS crb^RNAi^*). 5 min recording, time lapse 5 s.
https://elifesciences.org/articles/50900#video5

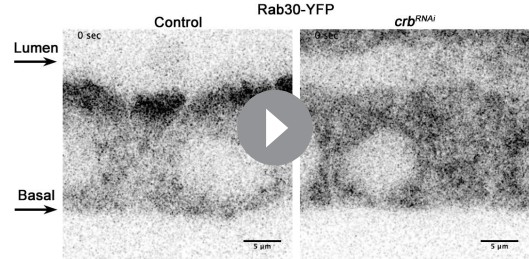

**Video 6.** Live imaging of endogenously expressed Rab30-YFP in SGs of control (left, *Rab30-YFP, fkh>/+*) and Crb KD (right, *Rab30-YFP, fkh >UAS crb^RNAi^*). 5 min recording, time lapse 5 s.
https://elifesciences.org/articles/50900#video6

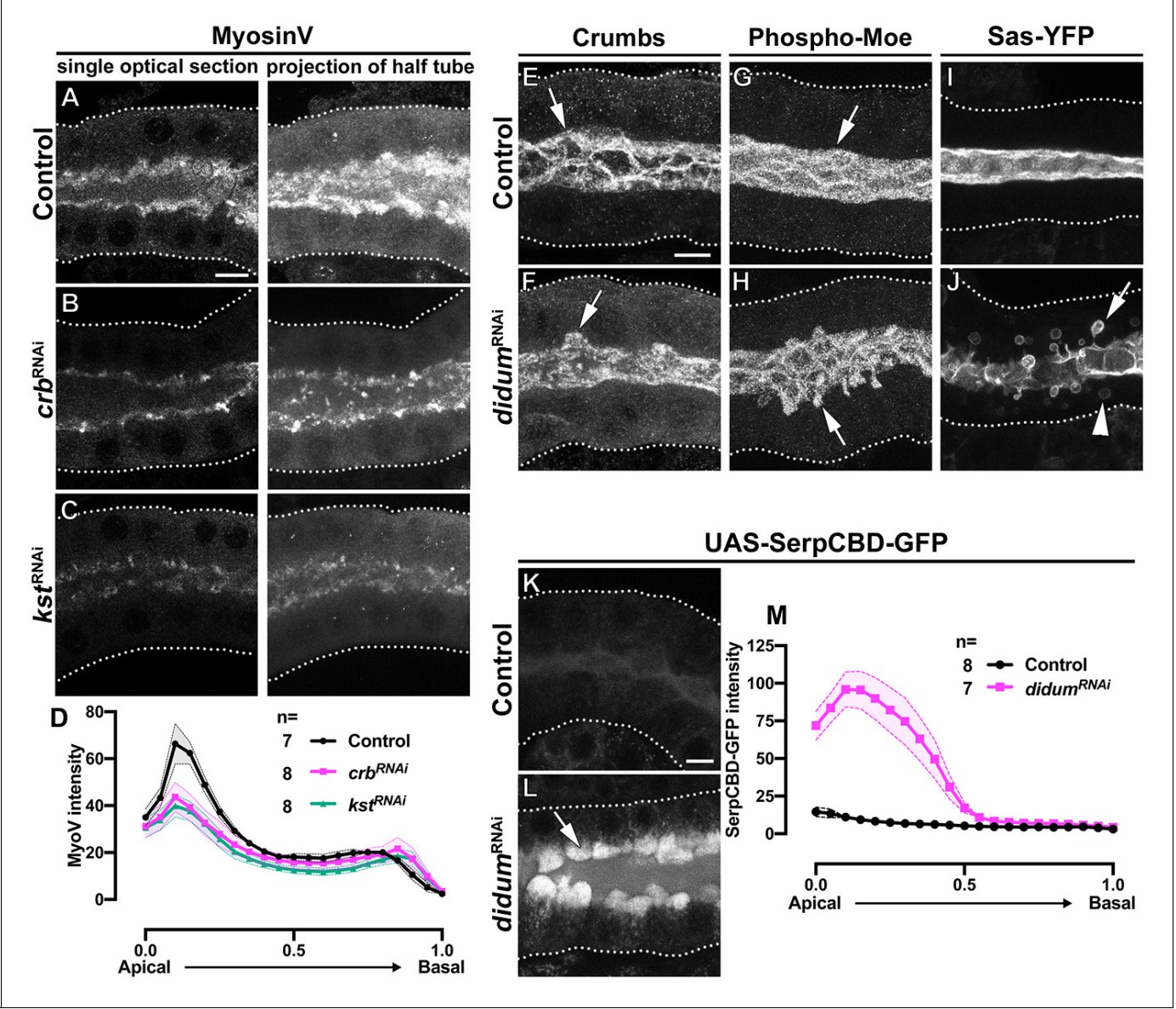

**Figure 3.** MyoV KD induces the intracellular extension of the apical membrane and disrupts apical secretion. (A-C) Single optical slices and maximal projection of half of the z-stack (half SG-tube) showing the localization of MyoV in fixed SGs of control (A, *fkh>/+*), Crb KD (B, *fkh >UAS crb^RNAi*) and β_H-Spec KD (C, *fkh >UAS kst^RNAi*) animals. (D, D) Plotted is the intensity (arbitrary units) of MyoV detected by immunofluorescence along the apical-to-basal direction in SGs of control (black, *fkh>/+*), Crb KD (magenta, *fkh >UAS crb^RNAi*) and β_H-Spec (green, *fkh >UAS kst^RNAi*) animals. Error bars indicate the standard error of the mean, n indicates number of glands from the corresponding genotypes. (E-J) Maximal projection of half of the z-stack (half SG-tube) showing the localization of Crb (E,F), Phospho-Moe (G,H) and Sas-YFP in SGs of control (E,G,I, *fkh>/+*) and MyoV KD (F,H,J, *fkh >UAS didum^RNAi*) animals. (K,L) Localization of SerpCBD-GFP in live SGs of control (K, *fkh >UAS* SerpCBD-GFP) and MyoV KD (L, *fkh >UAS didum^RNAi; UAS-SerpCBD-GFP*) animals. Arrows point to the apical and dotted lines indicate the basal membrane. Scale bars in (A,E,K) indicate 10 µm. (M, M) Plotted is the fluorescence intensity (arbitrary units) of SerpCBD-GFP along the apical-to-basal direction in live SGs of control (black, *fkh >UAS* SerpCBD-GFP), and MyoV KD (magenta, *fkh >UAS didum^RNAi; UAS-SerpCBD-GFP*) animals. Error bars indicate the standard error of the mean, n indicates number of glands from the corresponding genotypes.

The online version of this article includes the following source data and figure supplement(s) for figure 3:

**Source data 1.** Dataset for MyosinV fluorescence intensity in control glands.
**Source data 2.** Dataset for MyosinV fluorescence intensity in Crb KD glands.
**Source data 3.** Dataset for MyosinV fluorescence intensity in β_H-Spec KD glands.
**Source data 4.** Dataset for SerpCBD-GFP fluorescence intensity in control glands.
**Source data 5.** Dataset for SerpCBD-GFP fluorescence intensity in MyoV KD glands.
**Figure supplement 1.** Proper apical localization of MyoV requires Crb.

*Figure 3 continued on next page*

*Figure 3 continued*

**Figure supplement 1—source data 1.** Dataset for MyosinV-GFP fluorescence intensity in control glands.
**Figure supplement 1—source data 2.** Dataset for MyosinV-GFP fluorescence intensity in Crb KD glands.
**Figure supplement 2.** $\beta_H$-Spec is required for proper apical secretion.
**Figure supplement 2—source data 1.** Dataset for SerpCBD-GFP fluorescence intensity in control glands.
**Figure supplement 2—source data 2.** Dataset for SerpCBD-GFP fluorescence intensity in βH-Spec KD glands.

## Crb is a novel regulator of the apical Rab machinery in larval SGs

Other works have provided genetic evidence that links the presence of microvilli-containing inclusions to defects in the apical Rab trafficking machinery (*Feng et al., 2017*; *Knowles et al., 2015*; *Knowles et al., 2014*; *Sato et al., 2007*). The Rab protein family is a major regulator of intracellular membrane traffic routes (*Wandinger-Ness and Zerial, 2014*; *Pfeffer, 2013*) and MyoV is known to interact with Rab6 and Rab11 (*Lindsay et al., 2013*; *Li et al., 2007*; *Iwanami et al., 2016*), which play an important role in apical membrane trafficking and recycling (*Khanal et al., 2016*; *Iwanami et al., 2016*; *Chung and Andrew, 2014*; *Li et al., 2007*; *Satoh et al., 2005*; *Pelissier et al., 2003*). Therefore, to evaluate the effects of Crb depletion on the Rab machinery, we took advantage of the recently published library of Rab proteins endogenously tagged with YFP (*Dunst et al., 2015*). We knocked-down Crb in larval SG cells and systematically screened the expression of all Rab proteins (*Figure 4—figure supplement 1*). Strikingly, we found that loss of Crb affects the localization of a subset of Rab proteins, namely Rab6-YFP, Rab11-YFP and Rab30-YFP. Specifically, the apically localized pools of these Rab proteins are reduced (*Figure 4A–F'*, and *Videos 4–6*), while the basal pools are not affected significantly. The effects on this subset of Rab proteins are specific, as Crb KD does not alter the organization of other Rab proteins, like Rab1-YFP (*Figure 4G–H'*, *Video 7*, and *Figure 4—figure supplement 1*). Similar results were obtained in Sdt KD glands (data not shown). Importantly, total protein levels of these Rab proteins do not change significantly upon Crb KD (*Figure 4I*).

As shown above, KD of $\beta_H$-Spec affects MyoV localization and apical secretion similarly to Crb KD. Therefore, we tested the effects of $\beta_H$-Spec KD on the localization of Rab6-YFP, Rab11-YFP, Rab30-YFP and Rab1-YFP. Strikingly, KD of $\beta_H$-Spec only removes the apical pools of Rab6-YFP and Rab11-YFP (*Figure 4—figure supplement 2A–D*), while the apical Rab30-YFP and the intracellular Rab1-YFP compartments are not affected (*Figure 4—figure supplement 2E–H*). Thus, the apical localization of Rab6 and Rab11 require a functional apical cytocortex.

To examine whether the reduction in Rab6-YFP or Rab11-YFP relates to the formation of PAMS, we silenced them individually using a *gfp*$^{RNAi}$ and analyzed CD8-RFP localization. CD8-RFP accumulates intracellularly and localizes to the PAMS in Crb- and Sdt-deficient SGs (*Figure 1—figure supplement 1M,N* and EE,FF, and not shown). We found that KD of Rab6-YFP severely affects the morphology of the SGs and produces intracellular accumulation of CD8-RFP in large vesicles (*Figure 4—figure supplement 3A–B''*), which agrees with the general requirement of Rab6 in secretion (*Homma et al., 2019*). KD of Rab11-YFP also affects the morphology of the SGs, although a single lumen is still patent (*Figure 4—figure supplement 3D'*, asterisk). More importantly, loss of Rab11 results in formation of PAMS in larval SG cells (*Figure 4—figure supplement 3D''*, arrows). Hence, defects in the apical secretory machinery can induce the formation of PAMS.

Together, our results show that Crb is a novel regulator of apically localized Rab6-YFP, Rab11-YFP and Rab30-YFP. Moreover, $\beta_H$-Spec acts downstream of Crb to organize the apical localization of Rab6-YFP and Rab11-YFP. Therefore, the stabilization of $\beta_H$-Spec by Crb is essential to organize aspects of the apical Rab machinery for efficient apical secretion in larval SGs.

## Crb regulates apical membrane levels of PI(4,5)P$_2$

As we describe above, depletion of Crb, Sdt, $\beta_H$-Spec or MyoV induces accumulation of phospho-Moe in a subapical structure that we termed PAMS. Phospho-Moe can bind to PI(4,5)P$_2$ via its PH-domain (*Yonemura et al., 2002*; *Fiévet et al., 2007*; *Fehon et al., 2010*; *Roch et al., 2010*) and the phosphoinositide composition of a membrane regulates Rab protein activity, as well as the localization of cytoskeleton proteins (*Wandinger-Ness and Zerial, 2014*; *Tan et al., 2015*; *Mayinger, 2012*; *Liem, 2016*; *Bennett and Healy, 2009*; *Fehon et al., 2010*). Therefore, we explored whether loss of

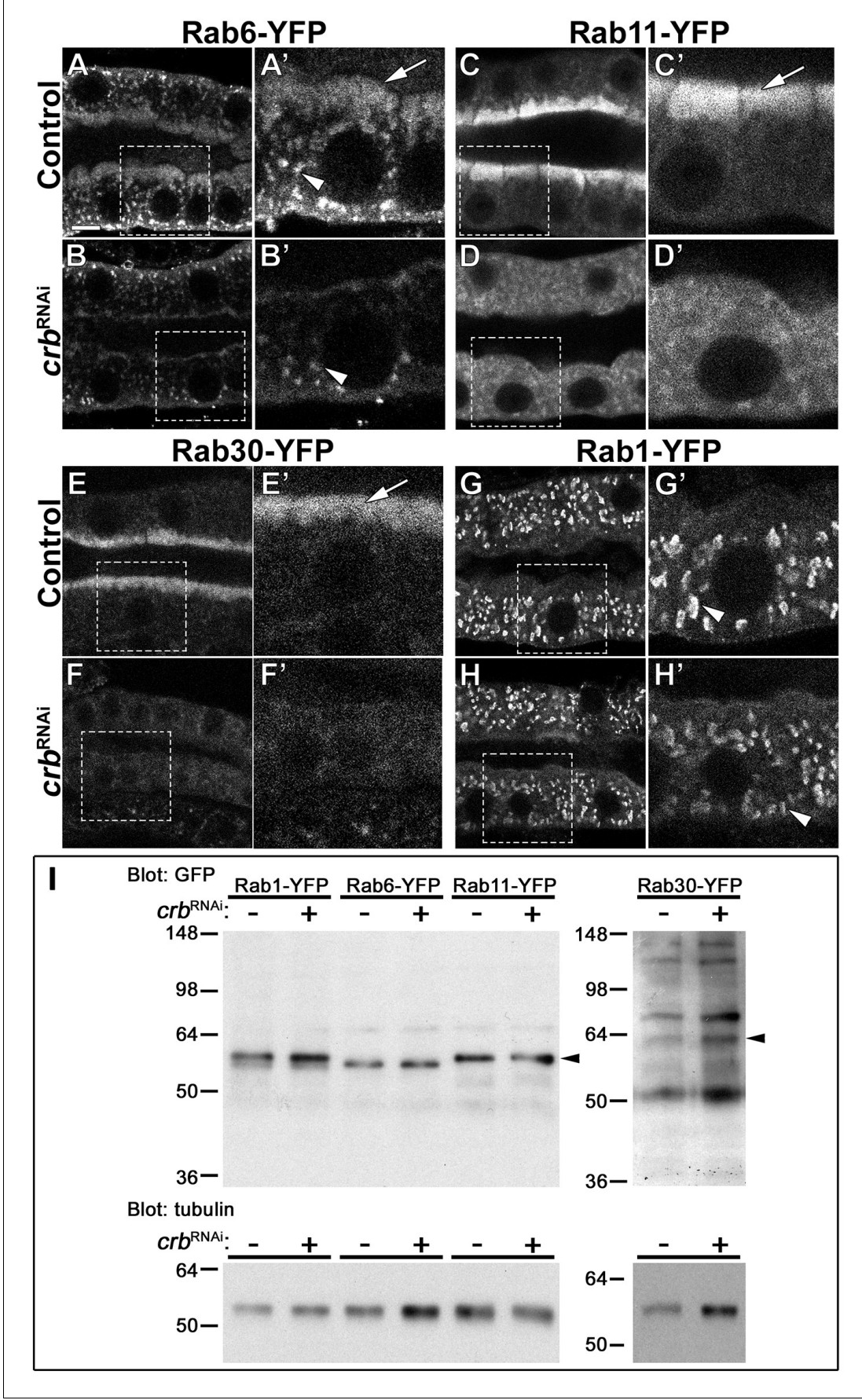

**Figure 4.** Crb organizes the apical Rab machinery in larval SG cells. (**A-H'**) Confocal images of SGs to localize endogenously expressed Rab-YFP proteins. Rab6-YFP (**A-B'**), Rab11-YFP (**C-D'**), Rab30-YFP (**E-F'**) and Rab1-YFP (**G-H'**) in control (**A,C,E,G**, *fkh>/+*) and Crb KD (**B,D,F,H**, *fkh >UAS crb^RNAi*) SGs. Dotted-line squares in A-H indicate the area blown-up to the right of the respective panel (**A'-H'**). Arrows point to the apical pool of Rab6-YFP (**A'**), Rab11-YFP (**C'**) and Rab30-YFP (**E'**). Arrowheads mark the intracellular vesicular localization of Rab6-YFP (**A',B'**) and Rab1-YFP (**G',H'**). Scale bar (**A**) indicates 10 μm. (**I, I**) Western blot of endogenously expressed Rab-YFP proteins. Rab1-YFP, Rab6-YFP, Rab11-YFP, and Rab30-YFP in control (*fkh>/+*) and Crb KD (*fkh >UAS crb^RNAi*) SGs, indicated as *crb^RNAi* – or +, respectively. Membranes were probed for tubulin (loading control) and for GFP; arrowheads point to Rab-YFP proteins.

The online version of this article includes the following figure supplement(s) for figure 4:

**Figure supplement 1.** Localization of Rab-YFP proteins after KD of Crb in larval SGs.

**Figure supplement 2.** The apical cytocortex is necessary for the organization of apical Rab6 and Rab11 trafficking machinery.

**Figure supplement 3.** Loss of Rab11 in larval SG induces the formation of PAMS.

Crb modulates the phosphoinositide composition of the apical membrane. For this, we monitored PI(4,5)P$_2$ localization by employing a well-established reporter containing the PI(4,5)P$_2$-specific PH-domain of phospholipase Cδ fused to GFP (PLCδ-PH-EGFP) (*Gervais et al., 2008*; *Rousso et al., 2013*; *Balla et al., 1998*; *Várnai and Balla, 1998*; *Rescher et al., 2004*).

Live imaging of larval SGs shows that PI(4,5)P$_2$ is enriched in the apical membrane (*Figure 5B,H*, *Figure 5—source data 1*), as previously observed in late 3$^{rd}$ instar SGs (*Rousso et al., 2013*). Importantly, quantification of PLCδ-PH-EGFP fluorescence intensity of Crb-deficient SGs shows an increase in apical levels of PI(4,5)P$_2$ (*Figure 5C,H*, *Figure 5—source data 2*). Additionally, PI(4,5)P$_2$ localizes in the PAMS (*Figure 5C*), which are also positive for phospho-Moe (*Video 8*). Similar results were observed in Sdt KD glands (*Figure 5—figure supplement 1A,B*).

To analyze whether β$_H$-Spec or MyoV participate in the accumulation of PI(4,5)P$_2$, we analyzed the distribution of PLCδ-PH-EGFP upon β$_H$-Spec or MyoV depletion. Indeed, KD of β$_H$-Spec or MyoV induces accumulation of PI(4,5)P$_2$ in the PAMS (*Figure 5—figure supplement 1C,D*), suggesting that loss of β$_H$-Spec and MyoV facilitates the increase of apical PI(4,5)P$_2$ levels and formation of PAMS.

We noted that PAMS are very heterogenous structures that are poorly preserved during fixation for immunohistochemistry. Therefore, we made use of live imaging to assess the frequency and morphology of the PAMS in the different genetic backgrounds. We used the signal from PLCδ-PH-EGFP and *D*E-cadherin-mTomato to measure the apical membrane area and volume (see Materials and methods). Our measurements show that KD of Crb, β$_H$-Spec or MyoV do not significantly change the amount of apical membrane surface or its volume, except for Crb-deficient cells, which have a slightly increased volume (*Figure 5—figure supplement 1E*, *Figure 5—figure supplement 1—source data 1*). We found that, when PAMS appear (% of cells with PAMS: 0% in control n = 322 cells; 46.7% in Crb KD n = 417 cells, 49.3% in β$_H$-Spec KD n = 503 cells; and 41,9% in MyoV KD n = 393 cells), there is a single sac per cell, which localizes toward the center of the apical domain. The PAMS diameter varies between 1.737 μm to 11.52 μm (mean ± SD: 5.325 ± 1.552 μm in Crb KD, 4.718 ± 1.382 μm in β$_H$-Spec KD, 5.012 ± 1.544 μm in MyoV KD; *Figure 5—source data 12*), suggesting that they could be dynamic. However, following up on single sacs by live imaging for 20 min revealed that these structures are rather steady (*Video 9*). Nevertheless, PAMS are not present in late 3$^{rd}$ instar SGs of wandering larvae (*Figure 5—figure supplement 1F,G*). Taken together these results indicate that Crb is essential to control the levels of PI(4,5)P$_2$ at the apical membrane. Moreover, our results suggest that at least part of this control is exerted by organizing β$_H$-Spec and MyoV at the apical aspect.

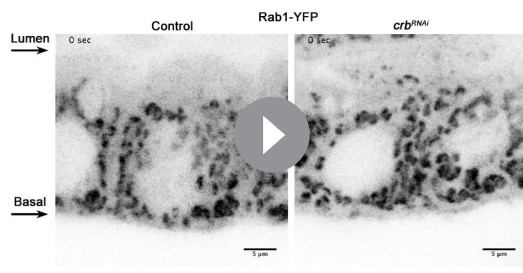

**Video 7.** Live imaging of endogenously expressed Rab1-YFP in SGs of control (left, *Rab1-YFP, fkh>/+*) and Crb KD (right, *Rab1-YFP, fkh >UAS crb^RNAi*). 5 min recording, time lapse 5 s.

https://elifesciences.org/articles/50900#video7

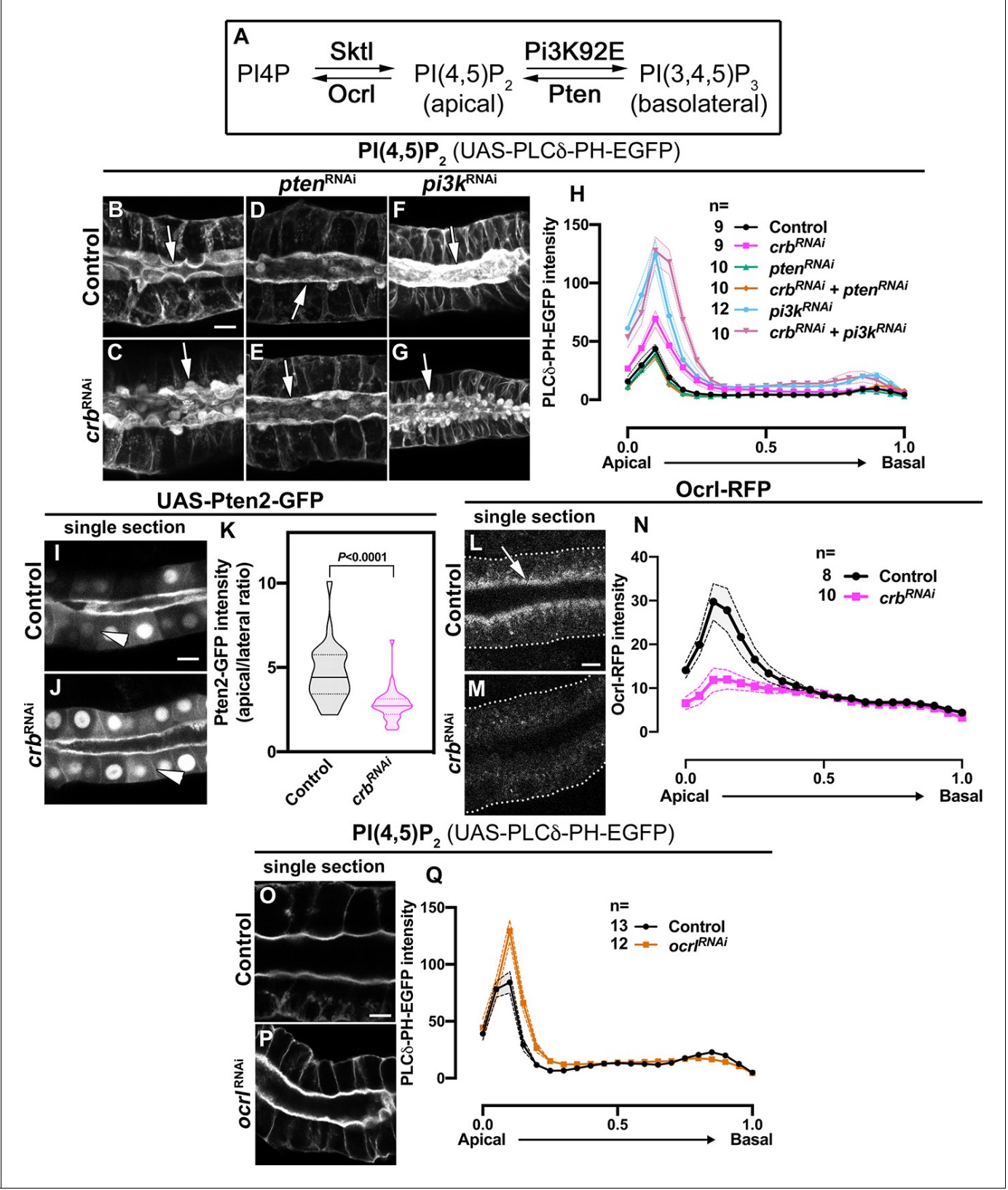

**Figure 5.** Crb organizes the apical secretory machinery by negatively regulating Pten A. (**A**) Simplified scheme of PI(4,5)P$_2$ biosynthesis. (**B-G**) Maximal projection of half of the z-stack (half SG-tube) showing the localization of PI(4,5)P$_2$ (PLCδ-PH-EGFP reporter) in live SGs of control (**B**, *fkh >UAS-PLCδ-PH-EGFP*), Crb KD (**C**, *fkh >UAS crb^RNAi; UAS-PLCδ-PH-EGFP*), Pten KD (**D**, *fkh >UAS pten^RNAi; UAS-PLCδ-PH-EGFP*), double KD of Crb and Pten (**E**, *Figure 5 continued on next page*

*Figure 5 continued*

*fkh >UAS crb^{RNAi}, UAS-pten^{RNAi}; UAS-PLCδ-PH-EGFP)*, Pi3K92E KD (F, *fkh >UAS-pi3k92E^{RNAi}; UAS-PLCδ-PH-EGFP*) and double KD of Crb and Pi3K92E (G, *fkh >UAS crb^{RNAi}, UAS-pi3k92E^{RNAi}; UAS-PLCδ-PH-EGFP*) animals. (H, H) Plotted is the fluorescence intensity (arbitrary units) of PLCδ-PH-EGFP along the apical-to-basal axis in live SGs of the genotypes indicated in (B-G), respectively. Error bars indicate the standard error of the mean, n indicates number of glands for the corresponding genotype. (I-K) Localization and quantification of over-expressed Pten2-GFP in SGs of control (I, *fkh >UAS-Pten2-GFP*) and Crb KD (J, *fkh >UAS crb^{RNAi}; UAS-Pten2-GFP*) animals. Violin graph (K) indicates the fluorescence intensity (apical vs lateral ratio) indicating the mean and quartiles (n = 28 cells for control and 36 cells for Crb KD). Statistical significance was analyzed in an unpaired two-tailed *t*-test. (L-N) Localization and quantification of Ocrl-RFP fluorescence intensity detected along the apical-to-basal axis in live SGs of control (black, *fkh>/+*) and Crb KD (magenta, *fkh >UAS crb^{RNAi}*) animals. Error bars indicate the standard error of the mean, n indicates number of glands of the corresponding genotypes. (O-Q) Localization and quantification of PLCδ-PH-EGFP fluorescence intensity detected along the apical-to-basal axis in live SGs of control (black, *fkh>/+*) and Ocrl KD (orange, *fkh >UAS ocrl^{RNAi}*) animals. Error bars indicate the standard error of the mean, n indicates the number of glands of the corresponding genotypes. Arrows point to the apical membrane domain. Arrowheads point to the lateral membrane. Dotted lines indicate the basal membrane. Scale bars in (B,I,L,O) indicate 10 μm.

The online version of this article includes the following source data and figure supplement(s) for figure 5:

**Source data 1.** Dataset for PLCδ-PH-EGFP fluorescence intensity in control glands (corresponding to panel H).
**Source data 2.** Dataset for PLCδ-PH-EGFP fluorescence intensity in Crb KD glands (corresponding to panel H).
**Source data 3.** Dataset for PLCδ-PH-EGFP fluorescence intensity in Pten KD glands (corresponding to panel H).
**Source data 4.** Dataset for PLCδ-PH-EGFP fluorescence intensity in glands with double KD of Crb and Pten (corresponding to panel H).
**Source data 5.** Dataset for PLCδ-PH-EGFP fluorescence intensity in Pi3K92E KD glands (corresponding to panel H).
**Source data 6.** Dataset for PLCδ-PH-EGFP fluorescence intensity in glands with double KD of Crb and Pi3K92E (corresponding to panel H).
**Source data 7.** Dataset for Pten2-GFP fluorescence intensity (corresponding to panel K).
**Source data 8.** Dataset for Ocrl-RFP fluorescence intensity in control glands (corresponding to panel N).
**Source data 9.** Dataset for Ocrl-RFP fluorescence intensity in Crb KD glands (corresponding to panel N).
**Source data 10.** Dataset for PLCδ-PH-EGFP fluorescence intensity in control glands (corresponding to panel Q).
**Source data 11.** Dataset for PLCδ-PH-EGFP fluorescence intensity in Ocrl KD glands (corresponding to panel Q).
**Source data 12.** Dataset for number of PAMS and diameter of PAMS.
**Figure supplement 1.** The Crb protein complex regulates apical levels of PI(4,5)P$_2$ and the secretory activity of SGs.
**Figure supplement 1—source data 1.** Dataset for apical surface quantifications.
**Figure supplement 1—source data 2.** Dataset for salivary gland lengths.
**Figure supplement 1—source data 3.** Dataset for PLCδ-PH-EGFP fluorescence intensity in control glands.
**Figure supplement 1—source data 4.** Dataset for PLCδ-PH-EGFP fluorescence intensity in Crb KD glands.
**Figure supplement 1—source data 5.** Dataset for PLCδ-PH-EGFP fluorescence intensity in Sktl KD glands.
**Figure supplement 1—source data 6.** Dataset for PLCδ-PH-EGFP fluorescence intensity in glands with double KD of Crb and Sktl.
**Figure supplement 1—source data 7.** Dataset for PLCδ-PH-EGFP fluorescence intensity in control glands incubated with vehicle.
**Figure supplement 1—source data 8.** Dataset for PLCδ-PH-EGFP fluorescence intensity in Crb KD glands incubated with vehicle.
**Figure supplement 1—source data 9.** Dataset for PLCδ-PH-EGFP fluorescence intensity in control glands incubated with VO-OHpic.
**Figure supplement 1—source data 10.** Dataset for PLCδ-PH-EGFP fluorescence intensity in Crb KD glands incubated with VO-OHpic.
**Figure supplement 2.** Pten2 over-expression induces formation of PAMS.
**Figure supplement 2—source data 1.** Dataset for GPR1-PH-EGFP fluorescence intensity.

## Crb controls apical membrane homeostasis by regulating phosphoinositide metabolism

To understand how the loss of Crb results in accumulation of PI(4,5)P$_2$, we explored the involvement of Pten, Pi3K, Sktl and Ocrl, key enzymes regulating PI(4,5)P$_2$ levels (*Figure 5A*). Expression of *pten*^{RNAi} (*Ramachandran et al., 2009*) in Crb KD glands effectively suppresses both the accumulation of PI(4,5)P$_2$ as measured by PLC-PH-EGFP fluorescence, and PAMS formation (*Figure 5D,E,H*, *Figure 5—source data 3* and *4*), while expression of *pi3K92E*^{RNAi} enhances the accumulation of PI(4,5)P$_2$ and PAMS formation (*Figure 5F,G,H*, *Figure 5—source data 5* and *6*). The latter also results in smaller glands (*Figure 5—figure supplement 1H*, *Figure 5—*

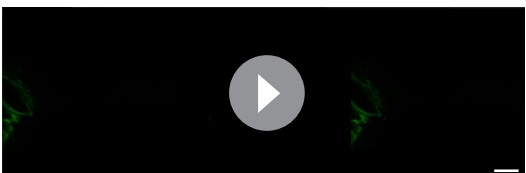

**Video 8.** 3D rendering of a fixed SG of a Crb KD animal expressing the PI(4,5)P$_2$ reporter PLCδ-PH-EGFP (green) and stained for phospho-Moesin (magenta). It is possible to appreciate the phospho-Moe and PI(4,5)P$_2$-enriched apical membrane sac (PAMS) below the apical membrane. Scale bar indicates 5 μm.
https://elifesciences.org/articles/50900#video8

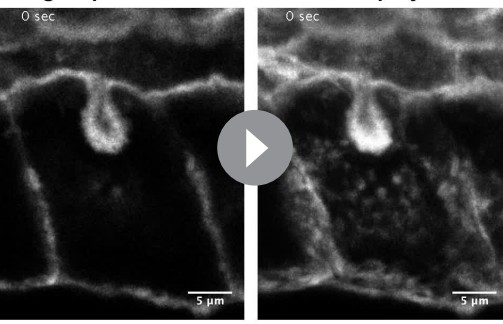

**single optical section** **maximal projection**

**Video 9.** Live imaging of a SG of a Crb KD animal expressing the PI(4,5)$P_2$ reporter PLCδ-PH-EGFP (*fkh >UAS crb$^{RNAi}$; UAS-PLCδ-PH-EGFP*). A single optical section is shown on the left. On the right, the maximal projection of the stack showing the whole PI(4,5)$P_2$-enriched apical membrane sac (PAMS). The arrowhead appearing at 660 s on the right panel points to an apparent opening of the sac to the lumen. It is worth noting that the PAMS are very stationary, as the movie shows 20 min recording, time lapse 20 s. Apical is up.

https://elifesciences.org/articles/50900#video9

figure supplement 1—source data 2), as expected due to the role of Pi3K in cell growth (*Huang et al., 1999*; *Goberdhan et al., 1999*; *Gao et al., 2000*; *Scanga et al., 2000*). Interestingly, KD of Sktl, another enzyme producing PI(4,5)$P_2$, is less effective in suppressing PAMS upon Crb KD than knocking-down Pten (*Figure 5—figure supplement 1I–M*, *Figure 5—figure supplement 1—source data 3–6*). To corroborate the importance for Pten to mediate the phenotype induced by loss of Crb, we found that over-expression of Pten2 induces accumulation of PI(4,5)$P_2$ and formation of PAMS (*Figure 5—figure supplement 2A,B*), while over-expression of Sktl results in strong defects in SG morphology (*Figure 5—figure supplement 2A, C*). Moreover, ex vivo incubation of SGs with VO-OHpic, a chemical inhibitor of Pten activity (*Mak et al., 2010*), eliminates the PAMS from Crb-deficient cells (*Figure 5—figure supplement 1N–R*, *Figure 5—figure supplement 1—source data 7–10*). Thus, our findings suggest that Pten is the main source of PI(4,5)$P_2$ involved in the formation of the PAMS upon Crb depletion.

Since apical Pten is important for restricting PI(3,4,5)$P_3$ to the basolateral membrane (*Worby and Dixon, 2014*; *Shewan et al., 2011*), we asked whether KD of Crb could affect PI(3,4,5)$P_3$ levels and Pten localization. We evaluated PI(3,4,5)$P_3$ levels using a probe containing the PH-domain of cytohesin tagged with GFP (*Pinal et al., 2006*). The signal of this probe at the plasma membrane is very weak and quantification of the fluorescence intensity revealed no significant change in the PI(3,4,5)$P_3$ apical-to-lateral ratio in Crb KD glands (*Figure 5—figure supplement 2D–F*, *Figure 5—figure supplement 2—source data 1*). Immunostainings to detect endogenous Pten were unsuccessful in our hands, therefore we expressed a UAS-transgene encoding the Pten2 isoform fused to GFP, which can rescue pupal eye development of *Pten* mutants (*Pinal et al., 2006*). Pten2-GFP over-expressed in larval SGs localizes to the apical domain in addition to the nucleus (*Figure 5I,J*). Interestingly, quantification of the Pten2-GFP fluorescence intensity revealed a decrease in the apical-to-lateral ratio in Crb and Sdt KD glands (*Figure 5K* and data not shown, *Figure 5—source data 7*), suggesting that Crb is required to ensure Pten levels at the apical membrane (*Figure 5I,J*, arrowheads). However, it is important to note that no PAMS were found in glands overexpressing Pten2-GFP, which is in contrast to the ones overexpressing Pten2 without a GFP tag (*Figure 5—figure supplement 2A,B*). Thus, the GFP tag could partially impair the phosphatase activity or expression levels could be lower than those achieved with Pten2 over-expression.

Besides Pten, Ocrl regulates PI(4,5)$P_2$ levels by dephosphorylating PI(4,5)$P_2$ into PI4P (*Balakrishnan et al., 2015*). Live imaging of Ocrl-RFP (knock-in allele) revealed its localization at the apical aspect in SG cells (*Figure 5L*). Moreover, KD of Crb severely decreases the apical localization of Ocrl (*Figure 5M,N*, *Figure 5—source data 8* and *9*). To evaluate the effect of Ocrl loss on PI(4,5)$P_2$ levels, we silenced the expression of Ocrl using a specific RNAi and quantified the fluorescence intensity of PLCδ-PH-EGFP. KD of Ocrl modestly increases the apical levels of PI(4,5)$P_2$ (*Figure 5O–Q*, *Figure 5—source data 10* and *11*), yet this is not accompanied by formation of PAMS.

Together, these results show that apical accumulation of PI(4,5)$P_2$ and formation of PAMS induced by the loss of Crb, seem to result from a combined effect of increased Pten activity and loss of Ocrl from the apical membrane upon loss of Crb.

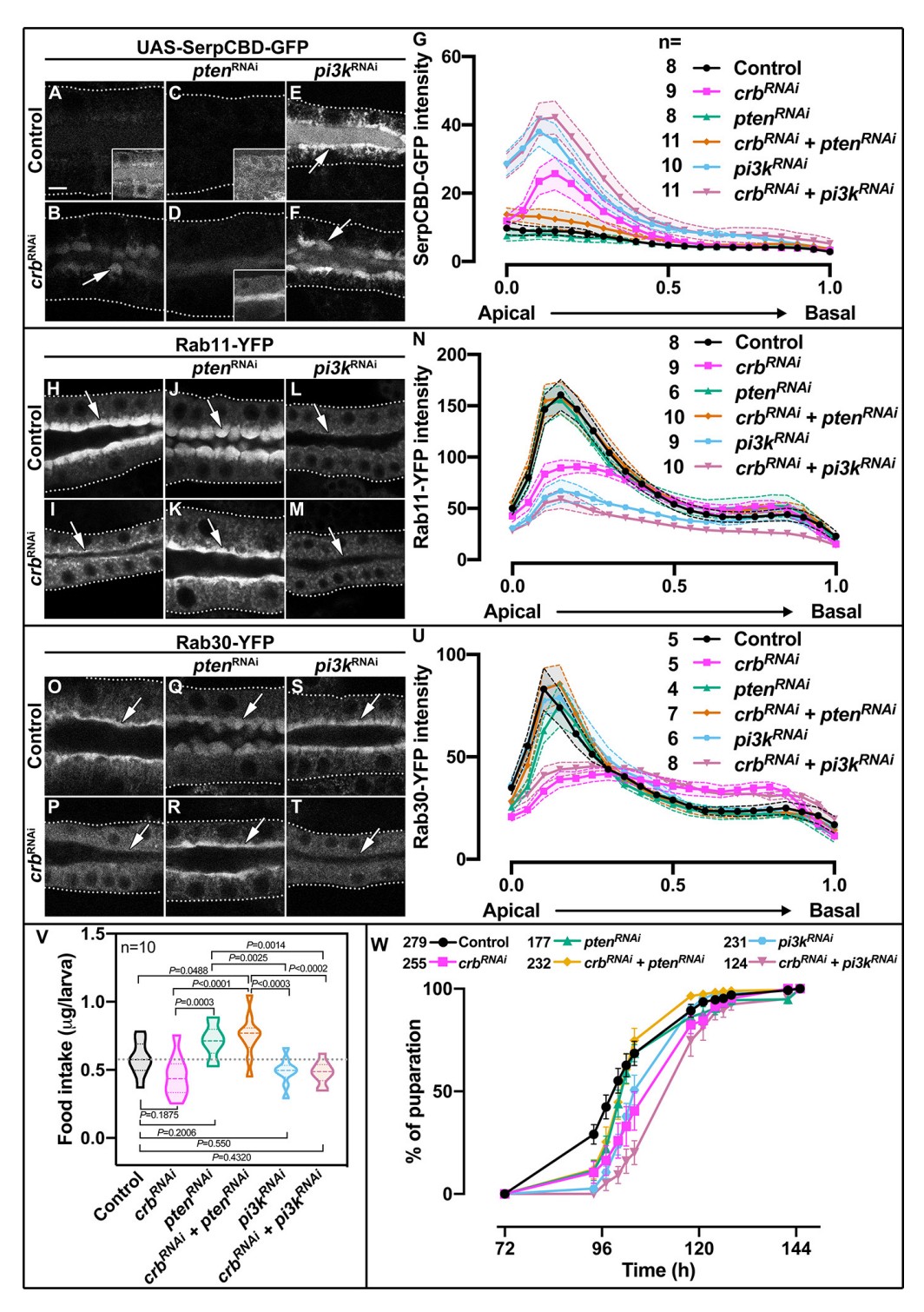

**Figure 6.** Control of apical secretion and localization of Rab11 and Rab30 by Crb requires Pten. (A–F) Maximal projection of 6.7µm through the SG lumen showing the localization of SerpCBD-GFP in live SGs of control (A, *fkh >UAS* SerpCBD-GFP), Crb KD (B, *fkh >UAS crb^RNAi; UAS-SerpCBD-GFP*), Pten KD (C, *fkh >UAS pten^RNAi; UAS-SerpCBD-GFP*), double KD of Crb and Pten KD (D, *fkh >UAS crb^RNAi, UAS-pten^RNAi; UAS-SerpCBD-GFP*), Pi3K92E KD (E, *fkh >UAS-pi3k92E^RNAi; UAS-SerpCBD-GFP*), and double KD of Crb and Pi3K92E (F, *fkh >UAS crb^RNAi, UAS-pi3k92E^RNAi; UAS-SerpCBD-GFP*), respectively. (H–M) Localization of endogenously expressed Rab11-YFP in live SGs. Shown are control (H, *Rab11-YFP, fkh>/+*), Crb KD (I, *Rab11-YFP, fkh >UAS crb^RNAi*), Pten KD (J, *Rab11-YFP, fkh >UAS pten^RNAi*), double KD of Crb and Pten (K, *Rab11-YFP, fkh >UAS crb^RNAi, UAS-pten^RNAi*), Pi3K92E KD (L, *Rab11-YFP, fkh >UAS-pi3k92E^RNAi*), and double KD of Crb and Pi3K92E (M, *Rab11-YFP, fkh >UAS crb^RNAi, UAS-pi3k92E^RNAi*) animals,

*Figure 6 continued on next page*

*Figure 6 continued*

respectively. (**O-T**) Localization of endogenously expressed Rab30-YFP in live SGs. Shown are control (**O**, *Rab30-YFP, fkh>/+*), Crb KD (**P**, *Rab30-YFP, fkh >UAS crb^{RNAi}*), Pten KD (**Q**, *Rab30-YFP, fkh >UAS pten^{RNAi}*), double KD of Crb and Pten (**R**, *Rab30-YFP, fkh >UAS crb^{RNAi}, UAS-pten^{RNAi}*), Pi3K92E KD (**S**, *Rab30-YFP, fkh >UAS-pi3k92E^{RNAi}*), and double KD of Crb and Pi3K92E (**T**, *Rab30-YFP, fkh >UAS crb^{RNAi}, UAS-pi3k92E^{RNAi}*) animals, respectively. Arrows point to the apical, and dotted lines to the basal membrane domain. Scale bar in (**A**) indicates 10 µm and applies to all panels. (**G,N,U**) Plotted is the fluorescence intensity (arbitrary units) of SerpCBD-GFP (**G**), Rab11-YFP (**N**) and Rab30-YFP (**U**), respectively, along the apical-to-basal axis in live SGs of the indicated genotypes. Error bars indicate the standard error of the mean, n indicates number of glands of the corresponding genotypes. (**V**) Violin graph of estimated food intake in control (first column), Crb KD (second column), Pten KD (third column), double KD of Crb and Pten (fourth column), Pi3K92E KD (fifth column), and double KD of Crb and Pi3K92E (sixth column) larvae. The dotted line indicates the mean value of the control. 60 larvae of the corresponding genotype were pooled in each biological replica. 10 biological replicas were analyzed distributed in three independent experiments. Statistical significance was tested in a one-way analysis of variance (ANOVA) followed by a Dunnett's multiple-comparison test. (**W**) Pupariation efficiency of control (black, *fkh>/+*), Crb KD (magenta, *fkh >UAS crb^{RNAi}*), Pten KD (green, *fkh >UAS pten^{RNAi}*), double KD of Crb and Pten KD (yellow, *fkh >UAS crb^{RNAi}, UAS-pten^{RNAi}*), Pi3K92E KD (blue, *fkh >UAS-pi3k92E^{RNAi}*), and double KD of Crb and Pi3K92E (, *fkh >UAS crb^{RNAi}, UAS-pi3k92E^{RNAi}*) animals. Error bars indicate the standard error of the mean, n indicates number of traced individual larvae of the corresponding genotypes in at least 15 independent experiments.

The online version of this article includes the following source data and figure supplement(s) for figure 6:

**Source data 1.** Dataset for SerpCBD-GFP fluorescence intensity in control glands.
**Source data 2.** Dataset for SerpCBD-GFP fluorescence intensity in Crb KD glands.
**Source data 3.** Dataset for SerpCBD-GFP fluorescence intensity in Pten KD glands.
**Source data 4.** Dataset for SerpCBD-GFP fluorescence intensity in glands with double KD of Crb and Pten.
**Source data 5.** Dataset for SerpCBD-GFP fluorescence intensity in Pi3K92E KD glands.
**Source data 6.** Dataset for SerpCBD-GFP fluorescence intensity in glands with double KD of Crb and Pi3K92E.
**Source data 7.** Dataset for Rab11-YFP fluorescence intensity in control glands.
**Source data 8.** Dataset for Rab11-YFP fluorescence intensity in Crb KD glands.
**Source data 9.** Dataset for Rab11-YFP fluorescence intensity in Pten KD glands.
**Source data 10.** Dataset for Rab11-YFP fluorescence intensity in glands with double KD of Crb and Pten.
**Source data 11.** Dataset for Rab11-YFP fluorescence intensity in Pi3K92E KD glands.
**Source data 12.** Dataset for Rab11-YFP fluorescence intensity in glands with double KD of Crb and Pi3K92E.
**Source data 13.** Dataset for Rab30-YFP fluorescence intensity in control glands.
**Source data 14.** Dataset for Rab30-YFP fluorescence intensity in Crb KD glands.
**Source data 15.** Dataset for Rab30-YFP fluorescence intensity in Pten KD glands.
**Source data 16.** Dataset for Rab30-YFP fluorescence intensity in glands with double KD of Crb and Pten.
**Source data 17.** Dataset for Rab30-YFP fluorescence intensity in Pi3K92E KD glands.
**Source data 18.** Dataset for Rab30-YFP fluorescence intensity in glands with double KD of Crb and Pi3K92E.
**Source data 19.** Dataset for food intake estimations.
**Source data 20.** Dataset for tracking of larval development.
**Figure supplement 1.** Overexpression of Pten leads to loss of Rab11 and Rab30 from the apical domain.

## Efficient apical secretion requires the control of PI(4,5)P$_2$ metabolism by Crb

To assess whether the secretion defects are a consequence of altered phosphoinositide metabolism we analyzed the secretion of SerpCBD-GFP and the organization of the apical Rab machinery. Live imaging analysis revealed that apical secretion of SerpCBD-GFP in Crb-deficient SGs is restored upon concomitant KD of Pten (*Figure 6A–D,G*, *Figure 6—source data 1* to 4), while KD of Pi3K92E alone, or in combination with Crb KD, induces a stronger apical retention of SerpCBD-GFP than the one observed in Crb-deficient SGs (*Figure 6B,E–G*, *Figure 6—source data 5* and *6*). Similar results were obtained using the probe for glycoproteins PNA-GFP (data not shown). Similarly, KD of Pten efficiently suppresses the loss of the apical pools of Rab11-YFP (*Figure 6H–K,N*, *Figure 6—source data 7* to 10) and Rab30-YFP (*Figure 6O–R,U*, *Figure 6—source data 13* to 16) observed upon Crb depletion. Interestingly, KD of Pi3K92E in control cells induces loss of apical Rab11-YFP (*Figure 6L, N*, *Figure 6—source data 11* and *12*), but has no effect on Rab30-YFP localization (*Figure 6S,U*, *Figure 6—source data 17* and *18*). Additionally, over-expression of Pten2 in the SGs induces the loss of apical pools of Rab11-YFP and Rab30-YFP (*Figure 6—figure supplement 1A–D*). This is in accordance with apical PI(4,5)P$_2$ levels regulating apical Rab proteins negatively. Unfortunately, the effects of Pten KD or over-expression on the apical pool of Rab6-YFP in the absence of Crb could not be studied due to lethality of the larvae. Thus, Crb function is required to organize the apical

cortex and to control the phosphoinositide metabolism, which in turn regulates the apical Rab protein machinery (Rab11-YFP, Rab30-YFP and possibly Rab6-YFP).

To assess the physiological relevance of Crb in SG secretion, we evaluated the larval food intake and tracked the pupariation time (*Deshpande et al., 2014*). We found that KD of Crb in the SGs, as well as KD of Pi3K92E, slightly reduces the amount of food intake (*Figure 6V*, *Figure 6—source data 19*), yet this reduction is not statistically significant (one-way ANOVA followed by Tukey's multiple comparisons test). Interestingly, concomitant KD of Crb and Pten significantly increases the larval food intake when compared to controls (*Figure 6V*). Importantly, these trends are reflected in the pupariation rate (*Figure 6W*, *Figure 6—source data 20*). Hence, while animals with SG-specific depletion of Crb take longer to pupariate than control animals (*Figure 6W*), those with additional Pten KD pupariate faster than those with Crb KD alone. Moreover, the pupariation of Pi3K92E KD animals is similar to the one of Crb-deficient animals, while concomitant KD of Crb and Pi3K92E delays the pupariation even more. Taken together, our results demonstrate that Crb is essential for apical membrane homeostasis, apical secretion and physiological function of larval SGs.

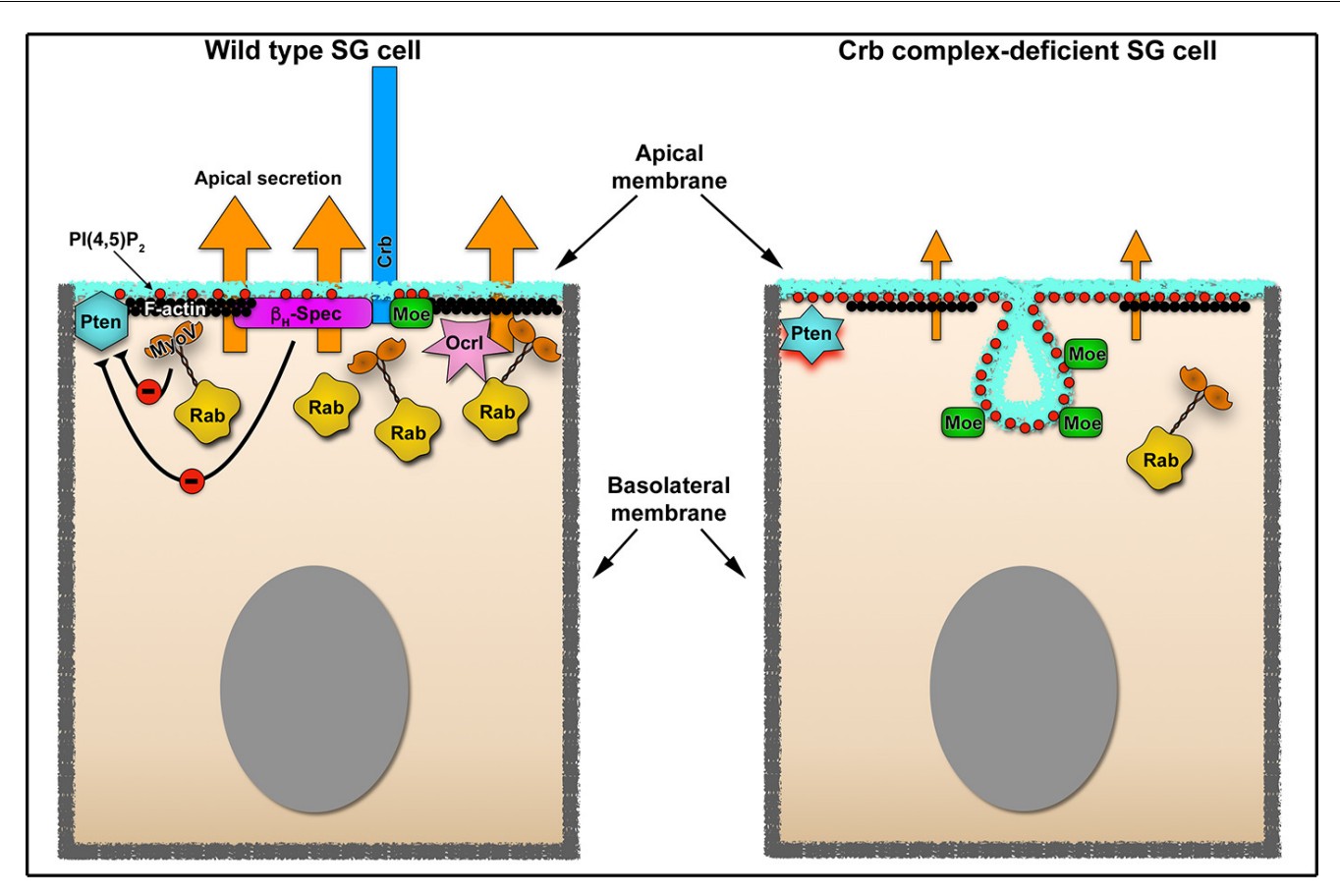

**Figure 7.** Crb-dependent regulation of apical secretion in SG cells Schematic representation of Crb-dependent regulation of apical secretion in SG cells. Under physiological conditions (left image), Crb mediates the apical localization of Moesin and $\beta_H$-Spec, which link the Crb protein (blue) to the apical F-actin cytoskeleton (black ribbon). This Crb-cytocortex complex is necessary for organization of the apical Rab-dependent traffic machinery (depicted as Rab vesicles in yellow). Under these conditions Crb negatively regulates the activity of Pten via $\beta_H$-Spec and MyoV. The precise molecular interactions involved in the negative regulation of Pten are not defined (see Discussion for details). The absence of Crb in the SG cells disrupts the efficient apical secretion (right image). The defects in apical secretion are a consequence of the disruption of the apical cytocortex (actin, $\beta_H$-Spec), the loss of MyoV and the excessive production of PI(4,5)P$_2$ (red dots) which require the activity of Pten. The loss of Ocrl form the apical membrane could also contribute to the increase in PI(4,5)P$_2$ apical levels. Another consequence is the formation of a novel apical membrane sac enriched in PI(4,5)P$_2$ (PAMS), Moe (green rectangles) and apical transmembrane proteins (not depicted).

## Discussion

In this work we identified unknown roles of Crb in constitutive apical secretion of larval SGs. Defects in apical secretion upon KD of Crb are not due to an overall disruption of epithelial cell polarity. Our results point to two major components acting downstream of Crb that regulate secretion. i) We found that the Crb complex is essential for Rab6-, Rab11- and Rab30-dependent, apical membrane transport machinery by ensuring the apical pools of these Rab proteins. This suggests that Crb maintains the active pool of these Rab proteins at the apical domain, as inactive GDP-bound Rab proteins associate with chaperone-like molecules, called GDP dissociation inhibitors (GDIs), and diffuse into the cytosol (*Goody et al., 2005*; *Grosshans et al., 2006*; *Müller and Goody, 2018*). ii) We show that Crb restricts the levels of PI(4,5)P$_2$ on the apical membrane by regulating apical activity and apical localization of Pten and Ocrl, respectively. As a consequence, Crb controls the size and organization of the apical membrane and efficient apical secretion, processes that are mediated in part by $\beta_H$-Spec and MyoV. From this we conclude that the Crb protein complex functions as an apical hub that interconnects and regulates these cellular machineries, which, in turn, are essential to maintain the physiological activity of the SGs (*Figure 7*).

### The roles of Crb in the regulation of constitutive saliva secretion

The late 3$^{rd}$ instar *Drosophila* SG has been extensively studied as a model for regulated exocytosis during the burst of glue granule secretion, which occurs at the onset of metamorphosis (reviewed in

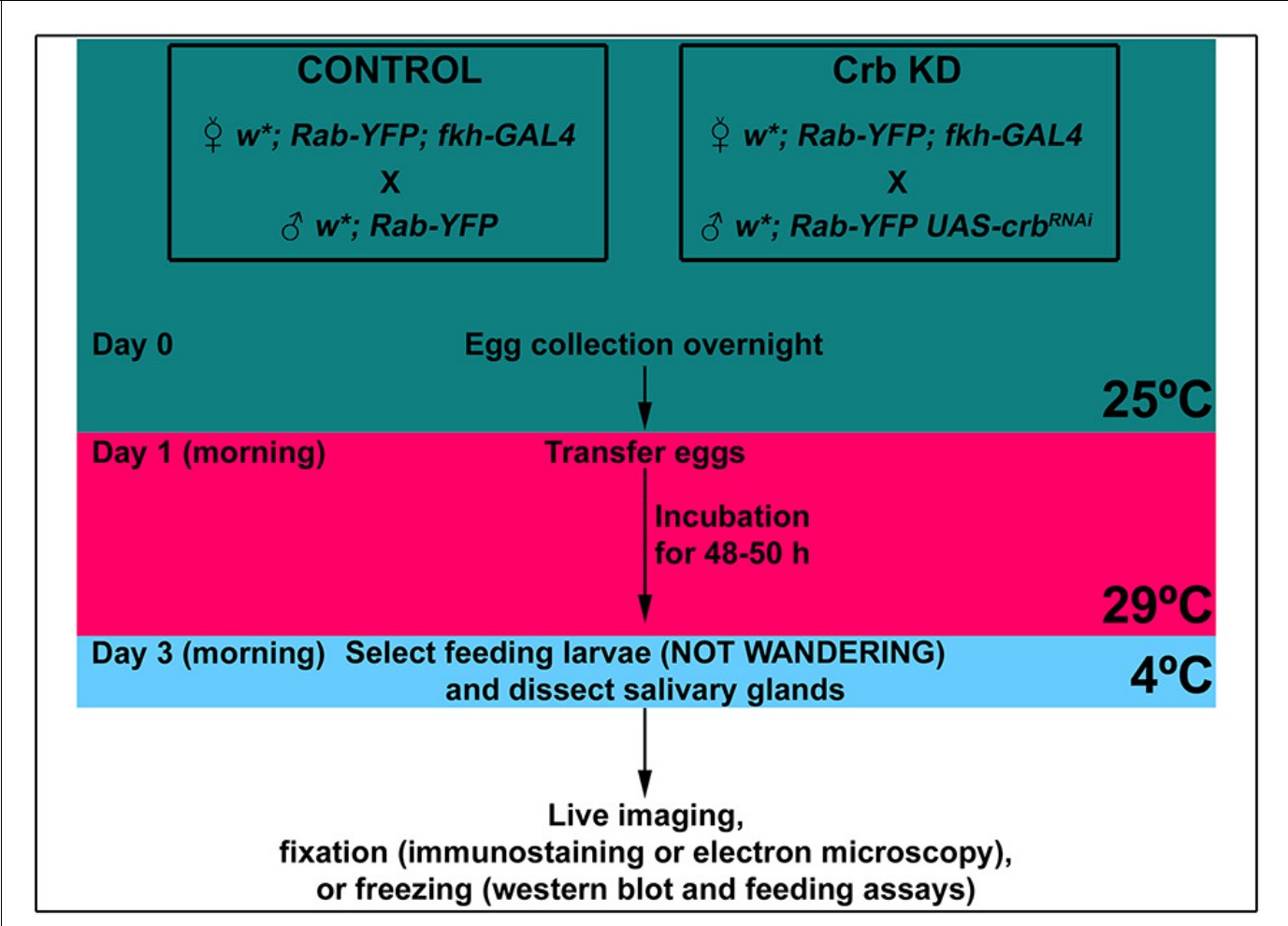

**Figure 8.** Schematic representation of the experimental setup. Indicated in the workflow are the times and incubation temperatures, as well as the time for dissections.

*Tran and Ten Hagen, 2017*). Here, we studied the roles of Crb in the regulation of constitutive saliva secretion in SGs at the beginning of the 3$^{rd}$ instar, while larvae are still feeding. At this stage there is a minimal synthesis of glue proteins (*Kodani, 1948*; *Rizki, 1967*; *Beckendorf and Kafatos, 1976*; *Korge, 1977*; *Zhimulev et al., 1981*), while salivary glycoproteins are actively secreted into the lumen (*Thomopoulos, 1988*; *Chung et al., 2014*; *Maruyama and Andrew, 2012*; *Csizmadia et al., 2018*; *Gregg et al., 1990*; *Fraenkel and Brookes, 1953*).

Loss of Crb or Sdt in SG cells results in hampered delivery of apical transmembrane proteins (Cad99C and CD8-RFP) as well as apical accumulation of secretion reporters (SerpCBD-GFP and PNA-GFP), which suggests at least two interpretations. Loss of Crb 1) hampers secretion, so that protein transport is jammed at the apical aspect, or 2) secretion is normal but endocytosis at the apical surface is strongly enhanced resulting in an immediate re-internalization of the secreted cargo. Loss of apical Rab6, Rab11, Rab30 and MyoV upon Crb KD supports the first interpretation. MyoV is a component of the apical secretory machinery (*Massarwa et al., 2009*; *Reck-Peterson et al., 2000*; *Li et al., 2007*) and known interactor of Crb, Rab6, Rab11 and possibly Rab30 (*Lindsay et al., 2013*; *Li et al., 2007*; *Iwanami et al., 2016*; *Pocha et al., 2011a*). Both Rab6 and Rab11 are known to facilitate apical transport and recycling (*Khanal et al., 2016*; *Iwanami et al., 2016*; *Chung and Andrew, 2014*; *Li et al., 2007*; *Satoh et al., 2005*; *Pelissier et al., 2003*), while Rab30 is suggested to be associated with the Golgi apparatus (*Kelly et al., 2012*). However, in larval SG cells Rab30 shows no co-localization with Golgi markers (*Dunst et al., 2015*) but instead localizes in a subapical pool. Interestingly, Rab30 was found as a potential MyoV-binding partner but later dismissed due to experimental threshold settings (*Lindsay et al., 2013*). Thus, although the functions of Rab30 in *Drosophila* are less clear (*Thomas et al., 2009*), our results suggest that active Rab30 contributes to MyoV-dependent transport.

A role of Crb in apical secretion rather than in apical endocytosis is further supported by our observations that the distribution of Rab proteins involved in endocytosis, namely Rab5, Rab7 and Rab21, is not affected by the loss of Crb (*Simpson et al., 2004*; *Chavrier et al., 1990*). This is also consistent with earlier observations that *crb* loss of function does not result in an overall increase in endocytosis in the eye imaginal disc epithelium (*Richardson and Pichaud, 2010*). Nevertheless, we cannot exclude the contribution of endocytosis completely, as inhibition of dynamin-dependent endocytosis seems to ameliorate the secretion phenotype of Crb-deficient glands, yet it does not block the formation of PAMS (data not shown). Moreover, by using dominant active or inactive forms of Rab5 (*Zhang et al., 2007*), we obtained inconsistent results (data not shown), probably due to pleiotropic effects of these versions of Rab5, which tend to titer effectors shared with other Rab proteins (*Pylypenko et al., 2017*; *Müller and Goody, 2018*). Although loss of β$_H$-Spec function has been linked to increased endocytosis in some *Drosophila* epithelia (*Williams et al., 2004*; *Pellikka et al., 2002*; *Richard et al., 2009*; *Phillips and Thomas, 2006*), or Crb mobility in the embryonic epidermis (*Bajur et al., 2019*) data presented here support the conclusion that in larval SGs Crb predominantly regulates apical membrane traffic and secretion, though we cannot completely rule out a minor contribution of endocytosis to the phenotypes observed.

We show that Crb is necessary to maintain the apical localization of MyoV. As mentioned above, MyoV is an interactor of Rab6, Rab11 and possibly Rab30 (*Lindsay et al., 2013*). This suggests that Crb can directly organize the apical secretion machinery by modulating the localization of MyoV. Additionally, our results also suggest that stabilization of β$_H$-Spec by Crb is important for organizing the apical Rab proteins. It is known that Crb regulates the actin cytoskeleton, and is necessary for recruitment of β$_H$-Spec and Moe to the apical cytocortex (*Flores-Benitez and Knust, 2015*; *Tsoumpekos et al., 2018*; *Salis et al., 2017*; *Röper, 2012*; *Sherrard and Fehon, 2015*; *Loie et al., 2015*; *Das and Knust, 2018*; *Médina et al., 2002a*; *Wei et al., 2015*; *Wodarz et al., 1995*). Unlike in other epithelia (*Wodarz et al., 1995*; *Pellikka et al., 2002*; *Médina et al., 2002b*; *Richard et al., 2009*), depletion of β$_H$-Spec in SGs does not result in loss of Crb or Sdt from the apical domain, but rather hampers apical secretion and induces the loss of Rab6- and Rab11-positive apical compartments. Therefore, the normal apical secretory activity of the SGs requires the Crb-dependent stabilization of β$_H$-Spec and MyoV at the apical cytocortex.

## Crb organizes the apical trafficking machinery by controlling apical PI(4,5)P$_2$ levels

Our results suggest that Crb is required to maintain the apical localization of Ocrl and to negatively regulate the activity of Pten, both key regulators of PI(4,5)P$_2$ levels (*Worby and Dixon, 2014*; *Balakrishnan et al., 2015*). Hence, apical PI(4,5)P$_2$ levels increase upon loss of Crb. Concomitantly, PI(4,5)P$_2$ as well as phospho-Moe and apical transmembrane proteins are found in a singular apical membrane extension, dubbed PAMS. Chemical inhibition or genetic ablation of Pten in Crb-deficient glands not only suppresses the formation of PAMS, but also restores the apical pools of Rab11, Rab30, apical secretion, larval food intake and timely pupariation. The relevance of PI(4,5)P$_2$ levels for proper secretion is highlighted by recent results demonstrating that the activity of *Drosophila* Crag (a Rab10 GEF) and Stratum (a Rab8 GEF) is regulated by the levels of PI(4,5)P$_2$ (*Devergne et al., 2014*; *Devergne et al., 2017*). For example, in the follicle epithelium, reduction of PI(4,5)P$_2$ levels results in defective secretion of basal membrane proteins, which then accumulate at the apical membrane (*Devergne et al., 2014*; *Devergne et al., 2017*). Moreover, recent work showed that another phosphoinositide species, PI(3,4)P$_2$, and the enzyme producing it, SHIP1, are key determinants of apical identity in a model of lumen formation (*Román-Fernández et al., 2018*). PI(3,4)P$_2$ was found to be an essential component of the pre-apical membrane and of Rab11a-positive recycling endosomes containing apical proteins that cluster together during de novo formation of the lumen. Indeed, perturbing PI(3,4)P$_2$ levels disrupts polarization through subcortical retention of vesicles at apical membrane initiation sites (*Román-Fernández et al., 2018*). Therefore, the control of PI(4,5)P$_2$ levels by Crb might impact the apical secretory machinery by altering the localization of specific effectors (GEFs or GAPs) of Rab6, Rab11 and Rab30.

Crb-mediated regulation of Pten partially depends on the organization of the apical cytocortex, but the precise molecular mechanism remains to be elucidated. So far, based on co-immunoprecipitation assays (data not shown) or on previous mass-spectrometry data (*Pocha et al., 2011b*) no direct interactions between Crb and Pten could be established. Yet, since apical localization of Baz, a binding partner of Pten (*von Stein et al., 2005*), is not affected by the loss of Crb, we suggest that the Baz-Pten interaction does not depend on Crb. On the other hand, KD of MyoV or β$_H$-Spec induces the formation of PAMS. Therefore, we favor the hypothesis that regulation of Pten might be mediated by β$_H$-Spec and MyoV acting downstream of Crb. Interestingly, an interaction between β$_H$-Spec and Pten was found by tandem affinity purification assays (*Vinayagam et al., 2016*), but whether this interaction regulates Pten activity was not analyzed. Furthermore, inhibition of MyoV-based transport increases the cell size of neurons, which mimics the PTEN-loss of function (*van Diepen et al., 2009*). Indeed, using immunoprecipitation and FRET analysis, it was shown that mammalian PTEN can interact directly with the MyoV C-terminal cargo-binding domain, yet the consequences of this interaction on PTEN activity or its localization were not evaluated (*van Diepen et al., 2009*). Therefore, it is plausible that Pten activity in the larval SGs can be regulated by interactions with MyoV and β$_H$-Spec. It is well-known that Pten regulation is very complex. Mammalian PTEN, for example, has more than 20 different sites, which can be subject to post-translational modifications (*Worby and Dixon, 2014*; *Gorbenko and Stambolic, 2016*). Therefore, it is likely that Crb can impinge on Pten activity via several different mechanisms, which can even be tissue- or developmental stage-specific. Indeed, it is well established that Crb as well as other polarity proteins have tissue specific functions, regulating cell signaling, cytoskeleton dynamics, cell division and cell adhesion as well as tissue growth and morphogenesis (reviewed in *Flores-Benitez and Knust, 2016*; *Tepass, 2012*). But even in one tissue like the larval SGs, Crb may control apical trafficking via additional mechanisms independent of Pten. For example, loss of apical Rab30 upon Crb KD is independent of β$_H$-Spec, suggesting that Crb can organize the apical trafficking machinery by additional effectors.

## PAMS – membrane entities dependent on PI(4,5)P$_2$ levels

It is well-known that Crb is a key determinant of the apical membrane and that over-expression of Crb in *Drosophila* embryos expands the apical membrane (*Wodarz et al., 1995*), Our findings on the functional link between Crb and Pten now provide a possible mechanism by which Crb exerts this function. In this context, the formation of the PAMS, apical membrane invaginations containing microvilli enriched in PI(4,5)P$_2$, phospho-Moe and apical transmembrane proteins (Sas, Crb and

CD8-RFP), offer an attractive model to study the regulation of apical membrane organization by Crb and Pten. Recently published data implicate PTEN in the regulation of apical membrane size. By using intestinal epithelial Ls174T:W4 cells in culture, Bruurs et al., showed that loss of PTEN results in formation of a larger brush border. In contrast, in mouse small intestinal organoids no change was observed (*Bruurs et al., 2018*), indicating that these effects can be tissue specific. Pten activity is necessary for the morphogenesis of rhabdomeres, a specialized apical membrane domain composed of a tightly packed stack of microvilli in *Drosophila* photoreceptors (*Pinal et al., 2006*). Indeed, Crb overexpression in *Drosophila* photoreceptor cells increases the amount of apical membrane (*Pellikka et al., 2002*; *Muschalik and Knust, 2011*). Therefore, it will be interesting to test whether this is mediated by Pten or by changes in intracellular trafficking.

It is important to note that the PAMS do not represent an expansion of the apical membrane upon loss of Crb (*Figure 5—figure supplement 1E*). On the contrary, based on our measurements of microvilli density (*Figure 2M*) and the disruption of Cad99C localization, the net effect of Crb KD is a reduction in the amount of apical membrane. Considering that one microvillus has a surface of approx. 0.55 $\mu m^2$ (roughly calculated from our EM images the height $\approx 2.5$ $\mu m$ and radius $\approx 35$ nm), 80 microvilli are found along 10 $\mu m$ of apical membrane thus 'contain' approx. 44 $\mu m^2$ of plasma membrane. Therefore, the reduction in microvilli number in Crb-deficient cells (without considering the reduction in their height) indicates that approx. 20 $\mu m^2$ of plasma membrane is found along the same length of the apical membrane, which is a loss of >50% of the apical membrane. Moreover, this loss of microvilli might be related to the defects in Cad99C localization upon loss of Crb, as Cad99C is important in maintenance of microvillar length (*Chung and Andrew, 2014*). Therefore, our results support the conclusion that Crb regulates the apical membrane architecture by maintaining the lipid homeostasis and the organization of the apical cytocortex. Upon loss of Crb, the collapse of the cytocortex together with the increase in PI(4,5)P$_2$ lead to the formation of PAMS and destabilization of the microvilli. Therefore, to understand how Crb regulates the proper proportions of apical vs. basolateral membranes, future studies need to address the biogenesis of the PAMS and whether their formation occurs at the expense of the basolateral membrane.

## Possible implications in human pathology

Defects in the membrane trafficking machinery are linked to a plethora of different pathologies, including immune syndromes, deafness, neuronal degeneration and cancer (reviewed in *Seabra et al., 2002*; *Holthuis and Menon, 2014*; *Krzewski and Cullinane, 2013*; *Bronfman et al., 2007*). The PAMS in Crb-deficient SG cells have striking similarities to the inclusion bodies observed in MVID patients carrying mutations in *MYO5b* (*Müller et al., 2008*; *Ruemmele et al., 2010*) or those found in animal models of MVID, like zebrafish mutant for *myosin Vb* (*Sidhaye et al., 2016*) and mice mutant for *Rab8a* and *Rab11a* (*Feng et al., 2017*; *Sato et al., 2007*). Moreover, recent data obtained in an intestinal organoid model of microvillus inclusion formation showed that these inclusion bodies are dynamic. Within hours, these inclusions can form and detach from the plasma membrane or collapse (*Mosa et al., 2018*). Furthermore, disruption of MyoVB, Rab8a, Rab11a, Syntaxin three and Syntaxin binding protein 2, all lead to defects similar to the ones observed in MVID enterocytes (*Sidhaye et al., 2016*; *Feng et al., 2017*; *Vogel et al., 2017*; *Schneeberger et al., 2015*; *Mosa et al., 2018*). Therefore, it is tempting to speculate that up-regulation of Pten activity could contribute to the pathogenesis of MVID.

Retinal degeneration is another pathological condition often caused by compromised trafficking machinery. Mutations in human *CRB1* induce retinal degeneration (*Richard et al., 2006b*; *Bulgakova and Knust, 2009*), similar as mutations in *Drosophila crb* (*Johnson et al., 2002*; *Pocha et al., 2011a*; *Pellikka et al., 2002*; *Izaddoost et al., 2002*; *Chartier et al., 2012*; *Spannl et al., 2017*) or overexpression of dominant negative versions (*Pellikka and Tepass, 2017*). Indeed, disruption of many of the proteins regulated by Crb in the SGs, including MyoV and Pten, can affect eye development, trafficking of Rh1 and ultimately photoreceptor survival in the fly (*Pocha et al., 2011a*; *Pellikka et al., 2002*; *Richard et al., 2009*; *Karagiosis and Ready, 2004*; *Iwanami et al., 2016*; *Pinal et al., 2006*; *Satoh et al., 2016*). Thus, it will be interesting to analyze whether Crb regulates the apical trafficking machinery in photoreceptor cells by modulating the phosphoinositide metabolism and how this is related to the pathogenesis of retinal degeneration.

In conclusion, data presented here reveal a role for the Crb complex beyond its canonical function as a polarity determinant in differentiating epithelial cells and show that Crb can fine-tune the

morphology and the molecular composition of the apical domain in a mature epithelium. In the future it will be interesting to explore whether the functional interactions described here are unique to early *Drosophila* SG cells or represent a conserved module also acting in other Crb-expressing epithelia.

## Materials and methods

### Fly stocks

Fly stocks (see *Table 1*) were maintained at room temperature (RT) on standard food. We employed the UAS-GAL4 system (*Elliott and Brand, 2008*) to drive the expression of different UAS-transgenes specifically in the salivary gland with the *fkh*-GAL4 driver (*Henderson and Andrew, 2000*). For detailed descriptions of the genotypes used in each figure see *Table 2*. The stocks in which the UAS-RNAi lines were recombined with the *fkh*-GAL4 driver together with the temperature sensitive repressor GAL80[ts] were maintained and expanded at 18°C. For experiments (see example in *Figure 8*), the crosses driving the different UAS-RNAi lines, and their corresponding controls, were done and maintained at 25°C. Eggs were collected overnight and then transferred to 29°C for approx. 48 hr. After this period, the feeding third instar larvae (not yet wandering) were collected for salivary gland dissections.

### Immunostaining of salivary glands

For all experiments, and in order to always compare equal timepoints of larval development, control and experimental genotypes were collected under the same conditions (see example in *Figure 8*). After growing at 29°C for approx. 50 hr, the salivary glands of non-wandering third instar larvae were dissected in ice cold Grace's medium (Thermo Fisher Scientific). Corresponding control and experimental glands were mounted together directly on a slide (previously coated with embryo-glue; *Figard and Sokac, 2011*) and then fixed. In this way, all staining-conditions were always identical for controls and experimental samples. Depending on the antigen (see *Table 3*), fixation was done in 100% methanol at −20°C for 5 min or in 6% formaldehyde in Grace's medium at RT for 15 min. For microtubule staining (*Riparbelli et al., 1993*), fixation was done in 100% methanol for 10 min followed by 5 min in acetone both at −20°C. Samples were washed at least 5 times with 0.1% Triton X-100 in 1xPBS (PBT) and blocked in 5% normal goat serum (NGS) in PBT (blocking solution) for 30 min at 4°C. Primary antibody staining was done in blocking solution over night at 4°C. Samples were washed at least 5 times with PBT before incubation with the appropriate secondary antibody in blocking solution for two hours at RT and washed again 5 times with PBT. The samples were covered with Vectashield (Vector Laboratories) and visualized using a Zeiss LSM 880 Airy upright single photon point scanning confocal system (ZEISS Microscopy, Jena, Germany) with a Zeiss iLCI Plan-Neofluar 63 × 1.3 Imm Korr DIC objective. In all cases, for any given marker, images were acquired under the same settings for laser power, PMT gain and offset. Maximal projections, merging and LUT-pseudocolor assignment were performed using Fiji (*Schindelin et al., 2012*). Image montage was done in Adobe Photoshop CS5 version 12.1 and when brightness, contrast and levels were adjusted, the modifications were linear and equally applied to the whole set of images. IMARIS 7.6 software was used to render the *Video 8*. Unless otherwise is stated, images are representative of at least three independent experiments, with at least three technical replicates in each experiment.

### Live imaging of salivary glands

Collection of control and experimental larvae was done as described above. For live imaging, the salivary glands were dissected in ice-cold Grace's medium, mounted on the bottom of a Petri dish previously coated with embryo-glue (*Figard and Sokac, 2011*) and imaged directly using a Zeiss LSM 880 Airy upright single photon point scanning confocal system (ZEISS Microscopy, Jena, Germany) with a Zeiss W Plan-Apochromat 40 × 1.0 objective. Excitation was performed with 488 nm for GFP or YFP from an Argon Multiline Laser, and 561 nm from a Diode Pumped Solid State (DPSS) Laser for RFP, mTomato and Dextran-Rhodamine. For time-lapse imaging of Rab-YFP proteins, 10 steps (0.67 μm/step) were acquired every 5 s for 5 min. Using FIJI software, the original stack was scaled 2X with a bicubic average interpolation, filtered with a Gaussian Blur (Sigma = 1) and animation speed set of 16 fps. Final montage and rendering were made in Photoshop CC 2018. Unless

**Table 1.** List of fly stocks used in this study.

| Designation | Genotype (as reported in FlyBase when available) | Description |
|---|---|---|
| Balancer | w[1118]; In(2LR)Gla, wg[Gla-1]/CyO, P{w[+mC]=GAL4 twi.G}2.2, P{w[+mC]=UAS-2xEGFP}AH2.2 | Balancer for 2nd chromosome; BSC 6662 |
| Balancer | w[1118]; Dr[Mio]/TM3, P{w[+mC]=GAL4 twi.G}2.3, P{UAS-2xEGFP}AH2.3, Sb[1] Ser[1] | Balancer for 3rd chromosome; BSC 6663 |
| Balancer | w[*]; ry[506] Dr[1]/TM6B, P{w[+mC]=Dfd-EYFP}3, Sb[1] Tb[1] ca[1] | Balancer for 3rd chromosome; BSC 8704 |
| crb[RNAi] | w[1118]; P{GD14463}v39177 | Expresses the RNAi against crb under the control of UAS sequences; VDRC 39177 |
| sdt[RNAi] | w[1118]; P{GD9163}v23822 | Expresses the RNAi against sdt under the control of UAS sequences; VDRC 23822 |
| sdt[RNAi] | y[1] sc[*] v[1]; P{y[+t7.7] v[+t1.8]=TRiP.HMS01652}attP40 | Expresses dsRNA for RNAi of sdt (FBgn0261873) under UAS control. BSC 37510 |
| gfp[RNAi] | y[1] sc[*] v[1]; P{y[+t7.7] v[+t1.8]=VALIUM20 EGFP.shRNA.3}attP40 | Expresses small hairpin RNA under the control of UAS for RNAi of EGFP and EYFP as well as fusion proteins containing these fluors, BSC 41559 |
| gfp[RNAi] | y[1] sc[*] v[1]; P{y[+t7.7] v[+t1.8]=VALIUM20 EGFP.shRNA.3}attP2 | Expresses small hairpin RNA under the control of UAS for RNAi of EGFP and EYFP as well as fusion proteins containing these fluors, BSC 41560 |
| moe[RNAi] | w[1118]; P{GD5211}v37917 | Expresses the RNAi against moe under the control of UAS sequences; VDRC 37917 |
| kst[RNAi] | y[1] v[1]; P{y[+t7.7] v[+t1.8]=TRiP.GLC01654}attP40 | Expresses dsRNA for RNAi of kst (FBgn0004167) under UAS control, BSC 50536 |
| ocrl[RNAi] | y[1] sc[*] v[1] sev[21]; P{y[+t7.7] v[+t1.8]=TRiP.HMS01201}attP2/TM3, Sb[1] | Expresses dsRNA for RNAi of Ocrl (FBgn0023508) under UAS control in the VALIUM20 vector. BSC 34722 |
| GAL80ts | w[*]; P{w[+mC]=tubP-GAL80[ts]}7 | Expresses temperature-sensitive GAL80 under the control of the alphaTub84B promoter; outcrossed from BSC 7018 |
| Dicer | w[1118]; P{w[+mC]=UAS-Dcr-2.D}2 | Expresses Dicer-2 under UAS control, BSC 24650 |
| myoV[RNAi] | y[1] sc[*] v[1]; P{y[+t7.7] v[+t1.8]=TRiP.HMC03900}attP40 | Expresses dsRNA for RNAi of didum (FBgn0261397) under UAS control; BSC 55740 |
| pten[RNAi] | y[1] w[1118]; P{w[+mC]=UAS Pten.dsRNA.Exel}2 | Expresses a snapback transcript for RNAi of Pten under the control of UAS. BSC 8549 |
| pten[RNAi] | w[1118]; P{w[+mC]=UAS Pten.dsRNA.Exel}3 | Expresses a snapback transcript for RNAi of Pten under the control of UAS. BSC 8550 |
| pi3k92E[RNAi] | y[1] sc[*] v[1]; P{y[+t7.7] v[+t1.8]=TRiP.HMC05152}attP40 | Expresses dsRNA for RNAi of Pi3K92E (FBgn0015279) under UAS control. BSC 61182 |
| pi3k92E[RNAi] | y[1] sc[*] v[1]; P{y[+t7.7] v[+t1.8]=TRiP .GL00311}attP2 | Expresses dsRNA for RNAi of Pi3K92E (FBgn0015279) under UAS control. BSC 35798 |

*Table 1 continued on next page*

*Table 1 continued*

| Designation | Genotype (as reported in FlyBase when available) | Description |
|---|---|---|
| sktl[RNAi] | y[1] sc[*] v[1]; P{y[+t7.7] v[+t1.8]=TRiP .GL00072}attP2 | Expresses dsRNA for RNAi of sktl (FBgn0016984) under UAS control. BSC 35198 |
| SerpCBD-GFP | w[*];; UAS-SerpCBD-GFP | Expresses the N-terminus of Serp including the signal peptide and chitin binding domain (CBD) fused to GFP (*Luschnig et al., 2006*), kindly provided by S. Luschning |
| MyosinV-GFP | w[*];; UAS-didum-GFP | Expresses full length didum (amino acids 1–1792) tagged at the C-terminal end with EGFP (*Krauss et al., 2009*), kindly provided by A. Ephrussi |
| Sas-Venus | w[*];; tub::Sas-Venus | Stranded at Second fused with Venus under tubulin promoter on 3rd chromosome (*Firmino et al., 2013*) |
| PNA-GFP | w[*]; M{w[+mC]=UAS PNA.GFP}ZH-86Fb | Expresses GFP-tagged peanut agglutinin under UAS control. BSC 55247 |
| CD8-RFP | w[*]; P{y[+t7.7] w[+mC]=10XUAS-IVS-mCD8::RFP}attP2 | Expresses mCD8-tagged RFP under the control of 10 UAS sequences. BSC 32218 |
| PI(4,5)P$_2$ sensor | y[1] w[*]; P{w[+mC]=UAS-PLCdelta-PH-EGFP}3 | Expresses GFP-tagged pleckstrin homology domain from human PLCδ. BSC 39693 |
| PI(3,4,5)P$_3$ sensor | w[*];; tub::GPR1-PH-EGFP | Expresses GFP-tagged pleckstrin homology domain from cytohesin/GRP1 (*Pinal et al., 2006*), kindly provided by F. Pichaud |
| Pten2-GFP | w[*]; UAS-Pten2-GFP | Expresses Pten2 isoform GFP-tagged under the control of UAS sequences (*Pinal et al., 2006*), kindly provided by F. Pichaud |
| Pten2 | w[*]; UAS-Pten2 | Expresses the Pten2 isoform under the control of UAS sequences (*von Stein et al., 2005*), kindly provided by A. Wodarz |
| fkhGAL4 | w[*]; fkh-GAL4 | On 3rd chromosome, expresses GAL4 under the control of the fkh promoter (*Henderson and Andrew, 2000*), kindly provided by K. Röpper |
| Fas3-GFP | w[*]; P{w[+mC]=PTT-GA}Fas3[G00258] | Fas3 fused with GFP protein trap. BSC 50841 |
| DE-cad-GFP | w*;DE-cad::GFP | DE-cadherin fused with GFP knock-in allele; homozygous viable (*Huang et al., 2009*), kindly provided by Y. Hong |
| DE-cad-mTomato | w*;DE-cad::mTomato | DE-cadherin fused with mTomato knock-in allele; homozygous viable (*Huang et al., 2009*), kindly provided by Y. Hong |

*Table 1 continued*

| Designation | Genotype (as reported in FlyBase when available) | Description |
| --- | --- | --- |
| Crb-GFP | w*;;crb::GFP-A | Crumbs fused with GFP knock-in allele; homozygous viable (*Huang et al., 2009*), kindly provided by Y. Hong |
| Lac-GFP | w*; lac::GFP | Protein trap line: *lachesin* fused with GFP under endogenous promoter on 2nd chromosome; homozygous viable (kindly provided by the Klämbt Protein trap consortium) |
| Nrv2-GFP | w*; nrv2::GFP | Protein trap line: *nervana2* fused with GFP under endogenous promoter on 2nd chromosome; homozygous viable (kindly provided by the Klämbt Protein trap consortium) |
| Ocrl-RFP | TI{T-STEP.TagRFP-T}Ocrl[KI] w[*] | A T-STEP cassette was knocked into Ocrl to tag the endogenous protein with TagRFP-T. BSC 66529 |
| Dlg-mTagRFP | Dlg-mTagRFP | On X chromosome, expresses Dlg-mTagRFP under the control of a ubiquitous promoter (*Pinheiro et al., 2017*), kindly provided by Y. Bellaïche |
| Rab-YFP | Rab-YFP | endogenously YFP::tagged Rab protein library generated in *Dunst et al. (2015)* |

BSC - Bloomington Drosophila stock Center

VDRC - Vienna Drosophila Resource Center.

otherwise is stated, images are representative of at least three independent experiments, with at least three technical replicates in each experiment.

## Image quantifications

The distribution and intensity levels of different markers were assessed using FIJI software. A flow-diagram of the analyses as well as all values obtained can be found in the accompanying Source Data. Briefly, to obtain the apical-to-basal fluorescence intensities of a particular marker, in a single-optical slice, individual straight lines (ROIs) were made from the apical membrane towards the basal membrane. The line width was set to 18 and all lines were arranged parallel to each other. Enough lines were made to cover the whole length of the salivary gland in the field of view (>70 μm) or a minimum of five cells per gland were covered (approx. 50 μm). The intensity values along the lines were obtained using the Multi Plot measurement option of FIJI. These intensity values were averaged along the length of the gland to obtain a single intensity distribution for one gland. The values for the line length were normalized to one and divided into 20 segments. The intensity values for each of the 20 segments was averaged and used to plot the final apical-to-basal fluorescence intensities.

To evaluate the apical-to-lateral ratios of a particular marker, in a single-optical slice, using the Multi-point tool of FIJI, a total of five dots (ROIs) were equally distributed along the apical membrane and five dots along the lateral membrane. The respective mean intensity values for apical and lateral membranes were obtained, averaged and the ratio was calculated. A minimum of four cells were evaluated for each gland.

For the quantification of the apical membrane (surface and volume), we analyzed the fluorescence of PLCδ-PH-EGF, to mark the plasma membrane including the PAMS, and *D*E-cadherin-mTomato, to distinguish the boundaries of the apical membrane, in Z-stacks acquired by confocal microscopy as described above. The plasma membrane was manually segmented using the Segmentation Editor

**Table 2.** List of detailed genotypes analyzed in each figure.

*Figure 1*

| | |
|---|---|
| B,B' | *w\*; UAS-crb[RNAi]/+* |
| C,C' | *w\*; UAS-crb[RNAi]/+; fkh-GAL4/+* |
| D | *w\*; Rab30-YFP, UAS-crb[RNAi]/+* |
| E | *w\*; Rab30-YFP, UAS-crb[RNAi]/+; fkh-GAL4/+* |
| F | *w\*; UAS-crb[RNAi]/+; Rab11-YFP/+* |
| G | *w\*; UAS-crb[RNAi]/+; Rab11-YFP/fkh-GAL4* |
| I | *w\*;; fkhGAL4, ubiGAL80[ts]* |
| J | *w\*; UAS-crb[RNAi]; fkhGAL4, ubiGAL80[ts]* |
| K | *w\*;; fkhGAL4, UAS-SerpCBD-GFP-GFP/+* |
| L | *w\*; UAS-crb[RNAi]/+; fkhGAL4, UAS-SerpCBD-GFP-GFP/+* |

*Figure 1—figure supplement 1*

| | |
|---|---|
| A,C,E,G,I,K | *w\*; UAS-crb[RNAi]/+* |
| B,D,F,H,J,L | *w\*; UAS-crb[RNAi]/+; fkh-GAL4/+* |
| M | *w\*;; UAS-CD8-RFP/fkhGAL4* |
| N | *w\*; UAS-crb[RNAi]/+; UAS-CD8-RFP/fkhGAL4* |
| O | *w\*;; UAS-PNA-GFP/fkhGAL4 ubiGAL80[ts]* |
| P | *w\*;; UAS-crb[RNAi]/+; UAS-PNA-GFP/fkhGAL4 ubiGAL80[ts]* |
| Q: Control | *w\*; UAS-crb[RNAi]/+; Rab11-YFP/+* |
| Q: Crb KD | *w\*; UAS-crb[RNAi]/+; Rab11-YFP/fkh-GAL4* |
| S,U,U',W,Y | *w\*;; UAS-std[RNAi]/+* |
| T,V,V',X,Z | *w\*; UAS-sdt[RNAi]/fkh-GAL4* |
| AA | *w\*;; Rab11-YFP, UAS-sdt[RNAi]/Rab11-YFP* |
| BB | *w\*;; Rab11-YFP, UAS-sdtRNAi/Rab11-YFP, fkhGAL4* |
| CC | *w\*;; fkhGAL4, ubiGAL80[ts]/+* |
| DD | *w\*; UAS-sdt[RNAi]; fkhGAL4, ubiGAL80[ts]/+* |
| EE | *w\*;; fkhGAL4, UAS-CD8-RFP/+* |
| FF | *w\*;; fkhGAL4, UAS-CD8-RFP/UAS-sdt[RNAi]* |
| GG | *w\*;; fkhGAL4, UAS-SerpCBD-GFP/+* |
| HH | *w\*;; fkhGAL4, UAS-SerpCBD-GFP/UAS-sdt[RNAi]* |
| II | *w\*;; fkhGAL4, UAS-PNA-GFP/+* |
| JJ | *w\*;; fkhGAL4, UAS-PNA-GFP/UAS-sdt[RNAi]* |

*Figure 1—figure supplement 2*

| | |
|---|---|
| A,A',B | *w\*; UAS-crb[RNAi]/+; Rab11-YFP/+* |
| C,C',D' | *w\*; UAS-crb[RNAi]/+; Rab11-YFP/fkh-GAL4* |
| E,G,I | *w\*; UAS-crb[RNAi]/+* |
| F,H,J | *w\*; UAS-crb[RNAi]/+; fkh-GAL4/+* |
| K,K' | *w\*; Fas3-GFP/Fas3-GFP; fkhGAL4/+* |
| L,L' | *w\*; Fas3-GFP/Fas3-GFP, UAS-crb[RNAi]; fkhGAL4/+* |
| M,M' | *w\*; Fas3-GFP/Fas3-GFP; fkhGAL4/UAS-gfp[RNAi]* |

*Figure 2*

| | |
|---|---|
| A,D | *w\*; UAS-crb[RNAi]/+* |
| B,E | *w\*; UAS-crb[RNAi]/+; fkh-GAL4/+* |
| G | *w\*; UAS-gfp[RNAi]/+; crb-GFP-A/crb-GFP-A* |

*Table 2 continued on next page*

*Table 2 continued*

*Figure 1*

| | |
|---|---|
| H | *w\*; UAS-gfp[RNAi]/+; crb-GFP-A/crb-GFP-A, fkh-GAL4* |
| I | *w\*;; fkhGAL4, ubiGAL80[ts]/tub::Sas-Venus* |
| J | *w\*; UAS-crb[RNAi]; fkhGAL4, ubiGAL80[ts]/tub::Sas-Venus* |
| K,K' | *w\*;; fkhGAL4, UAS-PLCdelta-PH-EGFP/+* |
| L,L' | *w\*; UAS-crb[RNAi]/+; fkhGAL4, UAS-PLCdelta-PH-EGFP/+* |

*Figure 2—figure supplement 1*

| | |
|---|---|
| A,C,E | *w\*; UAS-gfp[RNAi]/+; crb-GFP-A/crb-GFP-A* |
| B,D,F | *w\*; UAS-gfp[RNAi]/+; crb-GFP-A/crb-GFP-A, fkh-GAL4* |
| G,I | *w\*; UAS-crb[RNAi]/+* |
| H,J | *w\*; UAS-crb[RNAi]/+; fkh-GAL4/+* |

*Figure 2—figure supplement 2*

| | |
|---|---|
| A-C' | *w\*; UAS-crb[RNAi]/+; fkhGAL4, UAS-PLCdelta-PH-EGFP/+* |

*Figure 2—figure supplement 3*

| | |
|---|---|
| A | *w\*; UAS-crb[RNAi]/+* |
| B | *w\*; UAS-crb[RNAi]/+; fkh-GAL4/+* |
| C | *w\*; UAS-sdt[RNAi]/+* |
| D | *w\*; UAS-sdt[RNAi]/+; fkh-GAL4/+* |

*Figure 2—figure supplement 4*

| | |
|---|---|
| A,C,E | *w\*;; fkhGAL4, ubiGAL80[ts]* |
| B,D,F | *w\*; UAS-kst[RNAi]; fkhGAL4, ubiGAL80[ts]* |

*Figure 3*

| | |
|---|---|
| A | *w\*;; fkhGAL4, ubiGAL80[ts]* |
| B | *w\*; UAS-crb[RNAi]; fkhGAL4, ubiGAL80[ts]* |
| C | *w\*; UAS-kst[RNAi]; fkhGAL4, ubiGAL80[ts]* |
| E,G | *w\*;; fkhGAL4, ubiGAL80[ts]/+* |
| F,H | *w\*; UAS-didum[RNAi]/+; fkhGAL4, ubiGAL80[ts]/+* |
| I | *w\*;; fkhGAL4, ubiGAL80[ts]/tub::Sas-Venus* |
| J | *w\*; UAS-didum[RNAi]/+; fkhGAL4, ubiGAL80[ts]/tub::Sas-Venus* |
| K | *w\*;; fkhGAL4, UAS-SerpCBD-GFP/+* |
| L | *w\*; UAS-didum[RNAi]/+; fkhGAL4, UAS-SerpCBD-GFP/+* |

*Figure 3—figure supplement 1*

| | |
|---|---|
| A | *w\*;; fkhGAL4, ubiGAL80[ts]/UAS-MyoV-GFP* |
| B | *w\*; UAS-crb[RNAi]/+; fkhGAL4, ubiGAL80[ts]/UAS-MyoV-GFP* |

*Figure 3—figure supplement 2*

| | |
|---|---|
| A | *w\*;; fkhGAL4, ubiGAL80[ts]/UAS-SerpCBD-GFP* |
| B | *w\*; UAS-kst[RNAi]; fkhGAL4, ubiGAL80[ts]/UAS-SerpCBD-GFP* |

*Figure 4*

| | |
|---|---|
| A | *w\*; Rab6-YFP, UAS-crb[RNAi]/+* |
| B | *w\*; Rab6-YFP, UAS-crb[RNAi]/+; fkh-GAL4/+* |
| C | *w\*;; Rab11-YFP, fkhGAL4, ubiGAL80[ts]/Rab11-YFP* |
| D | *w\*; UAS-crb[RNAi]/+; Rab11-YFP, fkhGAL4, ubiGAL80[ts]/Rab11-YFP* |
| E | *w\*; Rab30-YFP/Rab30-YFP; fkhGAL4, ubiGAL80[ts]/+* |

*Table 2 continued on next page*

*Table 2 continued*

*Figure 1*

| | |
|---|---|
| F | *w\*; UAS-crb[RNAi], Rab30-YFP/Rab30-YFP; fkhGAL4, ubiGAL80[ts]/+* |
| G | *w\*; UAS-crb[RNAi]/+; Rab1-YFP/+* |
| H | *w\*; UAS-crb[RNAi]/+; Rab1-YFP/fkh-GAL4* |
| I: Rab1 Control | *w\*; UAS-crb[RNAi]/+; Rab1-YFP/+* |
| I: Rab1 Crb KD | *w\*; UAS-crb[RNAi]/+; Rab1-YFP/fkh-GAL4* |
| I: Rab6 Control | *w\*; Rab6-YFP, UAS-crb[RNAi]/+* |
| I: Rab6 Crb KD | *w\*; Rab6-YFP, UAS-crb[RNAi]/+; fkh-GAL4/+* |
| I: Rab11 Control | *w\*; UAS-crb[RNAi]/+; Rab11-YFP/+* |
| I: Rab11 Crb KD | *w\*; UAS-crb[RNAi]/+; Rab11-YFP/fkh-GAL4* |
| I: Rab30 Control | *w\*; UAS-crb[RNAi], Rab30-YFP/+;* |
| I: Rab30 Crb KD | *w\*; UAS-crb[RNAi], Rab30-YFP/+; fkhGAL4/+* |

*Figure 4—figure supplement 1*

| | |
|---|---|
| Rab1 Control | *w\*;; Rab1-YFP/fkhGAL4* |
| Rab1 Crb KD | *w\*;UAS-crb[RNAi]/+; Rab1-YFP/fkhGAL4* |
| Rab2 Control | *w\*; Rab2-YFP/+; fkhGAL4/+* |
| Rab2 Crb KD | *w\*; Rab2-YFP, UAS-crb[RNAi]/+; fkhGAL4/+* |
| Rab4 Control | *w\*; Rab4-YFP/+; fkhGAL4/+* |
| Rab4 Crb KD | *w\*; Rab4-YFP, UAS-crb[RNAi]/+; fkhGAL4/+* |
| Rab5 Control | *w\*; Rab5-YFP/+; fkhGAL4/+* |
| Rab5 Crb KD | *w\*; Rab5-YFP, UAS-crb[RNAi]/+; fkhGAL4/+* |
| Rab6 Control | *w\*; Rab6-YFP/+; fkhGAL4/+* |
| Rab6 Crb KD | *w\*; Rab6-YFP, UAS-crb[RNAi]/+; fkhGAL4/+* |
| Rab7 Control | *w\*;; Rab7-YFP/fkhGAL4* |
| Rab7 Crb KD | *w\*;UAS-crb[RNAi]/+; Rab7-YFP/fkhGAL4* |
| Rab8 Control | *w\*;; Rab8-YFP/fkhGAL4* |
| Rab8 Crb KD | *w\*;UAS-crb[RNAi]/+; Rab8-YFP/fkhGAL4* |
| Rab10 Control | *w\* Rab10-YFP/+;; fkhGAL4/+* |
| Rab10 Crb KD | *w\* Rab10-YFP/+; UAS-crb[RNAi]/+; fkhGAL4/+* |
| Rab11 Control | *w\*;; Rab11-YFP/fkhGAL4* |
| Rab11 Crb KD | *w\*;UAS-crb[RNAi]/+; Rab11-YFP/fkhGAL4* |
| Rab18 Control | *w\* Rab18-YFP/+;; fkhGAL4/+* |
| Rab18 Crb KD | *w\* Rab18-YFP/+; UAS-crb[RNAi]/+; fkhGAL4/+* |
| Rab21 Control | *w\* Rab21-YFP/+;; fkhGAL4/+* |
| Rab21 Crb KD | *w\* Rab21-YFP/+; UAS-crb[RNAi]/+; fkhGAL4/+* |
| Rab35 Control | *w\* Rab35-YFP/+;; fkhGAL4/+* |
| Rab35 Crb KD | *w\* Rab35-YFP/+; UAS-crb[RNAi]/+; fkhGAL4/+* |
| Rab39 Control | *w\* Rab39-YFP/+;; fkhGAL4/+* |
| Rab39 Crb KD | *w\* Rab39-YFP/+; UAS-crb[RNAi]/+; fkhGAL4/+* |
| Rab40 Control | *w\* Rab40-YFP/+;; fkhGAL4/+* |
| Rab40 Crb KD | *w\* Rab40-YFP/+; UAS-crb[RNAi]/+; fkhGAL4/+* |

*Figure 4—figure supplement 2*

| | |
|---|---|
| A | *w\*; Rab6-YFP/Rab6-YFP; fhkGAL4/+* |
| B | *w\*; Rab6-YFP, UAS-kst[RNAi]/Rab6-YFP; fhkGAL4/+* |

*Table 2 continued on next page*

*Table 2 continued*

*Figure 1*

| | |
|---|---|
| C | *w*;; Rab11-YFP, fkhGAL4, ubiGAL80[ts]/Rab11-YFP* |
| D | *w*; UAS-kst[RNAi]/+; Rab11-YFP, fkhGAL4, ubiGAL80[ts]/Rab11-YFP* |
| E | *w*; Rab30-YFP/Rab30-YFP; fhkGAL4/+* |
| F | *w*; Rab30-YFP, UAS-kst[RNAi]/Rab30-YFP; fhkGAL4/+* |
| G | *w*;; Rab1-YFP/fkhGAL4, ubiGAL80[ts]* |
| H | *w*; UAS-kst[RNAi]/+; Rab1-YFP/fkhGAL4, ubiGAL80[ts]* |

*Figure 4—figure supplement 3*

| | |
|---|---|
| A-A'' | *w*; Rab6-YFP/Rab6-YFP; fhkGAL4, UAS-CD8-RFP/+* |
| B-B'' | *w*; Rab6-YFP/Rab6-YFP, UAS-gfp[RNAi]; fhkGAL4, UAS-CD8-RFP/+* |
| C-C'' | *w*;; Rab11-YFP, fkhGAL4, UAS-CD8-RFP/Rab11-YFP* |
| D-D'' | *w*; UAS-gfp[RNAi]/+; Rab11-YFP, fkhGAL4, UAS-CD8-RFP/Rab11-YFP* |

*Figure 5*

| | |
|---|---|
| B | *w*;; fkhGAL4, UAS-PLCdelta-PH-EGFP/+* |
| C | *w*; UAS-crb[RNAi]/+; fkhGAL4, UAS-PLCdelta-PH-EGFP/+* |
| D | *w*;; fkhGAL4, UAS-PLCdelta-PH-EGFP/UAS-pten[RNAi]* |
| E | *w*; UAS-crb[RNAi]/+; fkhGAL4, UAS-PLCdelta-PH-EGFP/UAS-pten[RNAi]* |
| F | *w*;; fkhGAL4, UAS-PLCdelta-PH-EGFP/UAS-pi3k92E[RNAi]* |
| G | *w*; UAS-crb[RNAi]/+; fkhGAL4, UAS-PLCdelta-PH-EGFP/UAS-pi3k92E[RNAi]* |
| I | *w*;; UAS-pten2-GFP/fkh-GAL4, ubiGAL80[ts]* |
| J | *w*; UAS-crb[RNAi]/+; UAS-pten2-GFP/fkh-GAL4, ubiGAL80[ts]* |
| L | *Ocrl-RFP, w*/+;; fkh-GAL4, ubiGAL80[ts]/+* |
| M | *Ocrl-RFP, w*; UAS-crb[RNAi]/+; fkh-GAL4, ubiGAL80[ts]/+* |
| O | *w*;; fkhGAL4, UAS-PLCdelta-PH-EGFP/+* |
| P | *w*;; fkhGAL4, UAS-PLCdelta-PH-EGFP/UAS-ocrl[RNAi]* |

*Figure 5—figure supplement 1*

| | |
|---|---|
| A | *w*; DE-cad-mTomato/+; UAS-PLCdelta-PH-EGFP/fkhGAL4, ubiGAL80[ts]* |
| B | *w*;; fkhGAL4, UAS-PLCdelta-PH-EGFP/UAS-sdt[RNAi]* |
| C | *w*; UAS-kst[RNAi]/DE-cad-mTomato; fkhGAL4, ubiGAL80[ts]/UAS-PLCdelta-PH-EGFP* |
| D | *w*; UAS-didum[RNAi]/+; fkhGAL4 UAS-PLCdelta-PH-EGFP/+* |
| F | *w*;; UAS-PLCdelta-PH-EGFP/fkh-GAL4, ubiGAL80[ts]* |
| G | *w*; UAS-crb[RNAi]/+; UAS-PLCdelta-PH-EGFP/fkh-GAL4, ubiGAL80[ts]* |
| I | *w*;; fkhGAL4, UAS-PLCdelta-PH-EGFP/+* |
| J | *w*; UAS-crb[RNAi]/+; fkhGAL4, UAS-PLCdelta-PH-EGFP/+* |
| K | *w*;; fkhGAL4, UAS-PLCdelta-PH-EGFP/UAS-sktl[RNAi]* |
| L | *w*; UAS-crb[RNAi]/+; fkhGAL4, UAS-PLCdelta-PH-EGFP/UAS-sktl[RNAi]* |
| N,P | *w*;; fkhGAL4, ubiGAL80[ts]/UAS-PLCdelta-PH-EGFP* |
| O,Q | *w*; UAS-crb[RNAi]/+; fkhGAL4, ubiGAL80[ts]/UAS-PLCdelta-PH-EGFP* |

*Table 2 continued on next page*

*Table 2 continued*

| *Figure 1* | |
|---|---|
| *Figure 5—figure supplement 2* | |
| A | *w*;; fkhGAL4, UAS-PLCdelta-PH-EGFP/+* |
| B | *w*;; fkhGAL4, UAS-PLCdelta-PH-EGFP/UAS-pten2* |
| C | *UAS-Sktl w*/+;; fkhGAL4, UAS-PLCdelta-PH-EGFP/+* |
| D | *w*;; fkhGAL4, ubiGAL80[ts]/tub::GPR1-PH-EGFP* |
| E | *w*; UAS-crb[RNAi]/+;*<br>*fkhGAL4, ubiGAL80[ts]/tub::GPR1-PH-EGFP* |
| *Figure 6* | |
| A | *w*;; fkhGAL4, UAS-SerpCBD-GFP/+* |
| B | *w*; UAS-crb[RNAi]/+; fkhGAL4, UAS-SerpCBD-GFP/+* |
| C | *w*;; fkhGAL4, UAS-SerpCBD-GFP/UAS-pten[RNAi]* |
| D | *w*; UAS-crb[RNAi]/+;*<br>*fkhGAL4, UAS-SerpCBD-GFP/UAS-pten[RNAi]* |
| E | *w*;; fkhGAL4, UAS-SerpCBD-GFP/UAS-pi3k92E[RNAi]* |
| F | *w*; UAS-crb[RNAi]/+;*<br>*fkhGAL4, UAS-SerpCBD-GFP/UAS-pi3k92E[RNAi]* |
| H | *w*;; Rab11-YFP, fkhGAL4, ubiGAL80[ts]/Rab11-YFP* |
| I | *w*; UAS-crb[RNAi]/+;*<br>*Rab11-YFP, fkhGAL4, ubiGAL80[ts]/Rab11-YFP* |
| J | *w*; UAS-pten[RNAi]/+;*<br>*Rab11-YFP, fkhGAL4, ubiGAL80[ts]/Rab11-YFP* |
| K | *w*; UAS-crb[RNAi]/UAS-pten[RNAi];*<br>*Rab11-YFP, fkhGAL4, ubiGAL80[ts]/Rab11-YFP* |
| L | *w*; UAS-pi3k92E[RNAi]/+;*<br>*Rab11-YFP, fkhGAL4, ubiGAL80[ts]/Rab11-YFP* |
| M | *w*; UAS-crb[RNAi]/UAS-pi3k92E[RNAi];*<br>*Rab11-YFP, fkhGAL4, ubiGAL80[ts]/Rab11-YFP* |
| O | *w*; Rab30-YFP/Rab30-YFP; fkhGAL4, ubiGAL80[ts]/+* |
| P | *w*; UAS-crb[RNAi],*<br>*Rab30-YFP/Rab30-YFP; fkhGAL4, ubiGAL80[ts]/+* |
| Q | *w*; Rab30-YFP/Rab30-YFP;*<br>*fkhGAL4, ubiGAL80[ts]/UAS-pten[RNAi]* |
| R | *w*; UAS-crb[RNAi], Rab30-YFP/Rab30-YFP;*<br>*fkhGAL4, ubiGAL80[ts]/UAS-pten[RNAi]* |
| S | *w*; Rab30-YFP/Rab30-YFP;*<br>*fkhGAL4, ubiGAL80[ts]/UAS-pi3k92E[RNAi]* |
| T | *w*; UAS-crb[RNAi], Rab30-YFP/Rab30-YFP;*<br>*fkhGAL4, ubiGAL80[ts]/UAS-pi3k92E[RNAi]* |
| *Figure 6—figure supplement 1* | |
| A | *w*;; Rab11-YFP, fkhGAL4, ubiGAL80[ts]/Rab11-YFP* |
| B | *w*;; Rab11-YFP, fkhGAL4,*<br>*ubiGAL80[ts]/Rab11-YFP, UAS-pten2* |
| C | *w*; Rab30-YFP/Rab30-YFP; fkhGAL4, ubiGAL80[ts]/+* |
| D | *w*; Rab30-YFP/Rab30-YFP;*<br>*fkhGAL4, ubiGAL80[ts]/UAS-pten2* |

**Table 3.** List of antibodies and probes employed.

| | Dilution | Fixation | Source |
|---|---|---|---|
| DAPI | 1:200000 | FA | Invitrogen Cat. D1306 |
| Phalloidin Alexa Flour 488, 555 | 1:2000 | FA | Invitrogen Cat. A12379, A34055 |
| Alexa Flour 488-, 568- and 647 -conjugated | 1:1000 - 1:2000 | | Invitrogen |
| **Mouse antibodies** | | | |
| Anti-α-Spectrin | 1:100 | MeOH | DSHB 3A9 |
| Anti-Coracle | 1:200 | MeOH | DSHB C566.9 |
| Anti-Disc large | 1:500 | MeOH | DSHB 4F3 |
| Anti-FasIII | 1:4 | MeOH | DSHB 7G10 |
| Anti-αTubulin | 1:2000 | MeOH/Acetone | MPI-CBG Antibody facility, P. Keller |
| **Rabbit antibodies** | | | |
| Anti-aPKC (C-20) | 1:500 | MeOH | Santa Cruz Biotechnology Cat. sc-216-G |
| Anti-Bazooka | 1:200 | MeOH | kindly provided by A. Wodarz (*Wodarz et al., 1999*) |
| Anti-Stardust | 1:2000 | MeOH | (*Berger et al., 2007*) |
| Anti-Cadherin99C | 1:250 | FA | kindly provided by D. Godt (*Glowinski et al., 2014*) |
| Anti-GFP | 1:1000 | FA | Invitrogen A-11122 |
| Anti-Sinuous | 1:8000 | MeOH | kindly provided by G.J. Beitel (*Wu et al., 2004*) |
| Anti-$\beta_H$Spectrin | 1:5000 | MeOH | kindly provided by G. Thomas (*Thomas and Williams, 1999*) |
| Anti-KuneKune | 1:5000 | MeOH | kindly provided by M. Furuse (*Nelson et al., 2010*) |
| Anti-Phospho-Ezrin (Moesin) | 1:500 | FA | Cell Signaling Technology Cat. 3141 |
| Anti-Moesin (Q480) | 1:400 | FA | Cell Signaling Technology Cat. 3150 |
| Anti-MyosinV | 1:2000 | MeOH | (*Pocha et al., 2011a*) |
| Anti-*DPatj* | 1:1000 | FA | (*Richard et al., 2006a*) |
| **Rat antibodies** | | | |
| Anti-Yurt | 1:500 | MeOH | kindly provided by U. Tepass (*Laprise et al., 2006*) |
| Anti-Stardust | 1:2000 | FA | (*Berger et al., 2007*) |
| **Chicken antibodies** | | | |
| Anti-GFP | 1:100 | FA | Abcam Cat. Ab13970 |
| **Guinea pig antibodies** | | | |
| Anti-Crumbs 2.8 | 1:500 | MeOH | (*Richard et al., 2006a*) |
| Anti-Par6 | 1:500 | FA | kindly provided by A. Wodarz (*Shahab et al., 2015*) |

DSHB - Developmental Studies Hybridoma Bank (Iowa city, Iowa, USA)

Invitrogen, Molecular Probes (Eugene, Oregon, USA)

Santa Cruz Biotechnology, Inc (Dallas, Texas, USA)

Cell Signaling Technology (Danvers, Massachusetts, USA)

Abcam plc (Cambridge, United Kingdom)

plugin in Fiji Software. The labeled images obtained were subsequently analyzed using the 3D Object Counter plugin to obtain the values for surface and volume.

## Transmission electron microscopy (TEM) and high-pressure freezing (HPF)

Control and experimental larvae were collected as described above. Salivary glands were dissected on ice in 1xPBS and fixed with 2.5% glutaraldehyde, 2% paraformaldehyde in 1xPBS for 2 hr at RT, washed with 1xPBS, 3 times for 5 min at RT, post-fixed with 1% osmium tetroxide, 1.5% Potassium ferricyanide in water for 1 hr at 4°C. Samples were dehydrated in serial steps (30%, 50%, 70%, 90%, and 100%) Ethanol (EtOH) 5 min/step at 4°C, infiltrated with 1:3 EPON LX112/EtOH for 1 hr, 1:1 EPON LX112/EtOH for 1 hr, 3:1 EPON LX112/EtOH 1 hr, pure EPON LX112 overnight, and pure EPON LX112 for 2 hr. The salivary glands were embedded in rubber mold and polymerized for 24 hr at 60°C. 70 nm cross sections were obtained using an ultramicrotome and were picked up with formvar coated copper slot grid. Grids were stained with 2% uranyl acetate in water for 10 min and lead citrate for 5 min at RT.

For HPF, salivary glands were dissected on ice in 1xPBS and frozen afterwards using a Leica ICE high pressure freezer (Leica Microsystems, Germany). Media of frozen samples was substituted with a cocktail containing 0.1% uranyl acetate and 4% water in acetone at $-90°C$. Samples were transferred into ethanol at $-25°C$. Then, samples were embedded into a Lowicryl HM20 resin (Polysciences, Inc, Germany) followed by UV polymerization at the same temperature. Semi-thin sections (300 nm) were cut and contrasted as described above for chemically fixed samples.

To quantify the density of microvilli, five lines, 1 µm in length each, were drawn adjacent to the apical membrane and distributed over the span of a cell. Five identical lines were drawn parallel to the first ones but at exactly 1 µm away from the first group, that is 1 µm above the apical membrane. The microvilli crossed by these lines were counted and the average per cell is presented in the *Figure 2M*.

Image acquisition was done using a Tecnai 12 (FEI, Thermo Fisher Scientific) with a standard single tilt holder with a TVIPS TemCam F214A (TVIPS, Gauting, Germany) digital camera at 440x for an overview of the whole salivary gland cross section, and 1200x for single-cell overview and 13000x for subcellular structures. Images are representative of 3 independent experiments, at least 3–5 different salivary glands were analyzed per genotype.

## Dextran-permeability assay, lysosomal activity and treatment with inhibitors

For Dextran permeability assays we adapted the method from *Lamb et al. (1998)*. Briefly, the salivary glands were dissected as described above, and incubated 15 min at RT in Grace's medium containing 40 µg/ml Dextran-Rhodamine B 10,000 MW (Molecular Probes D1824), and immediately imaged after incubation. For lysosomal activity analysis, the salivary glands were incubated 30 min at RT in Grace's medium containing 150 nM LysoTracker Red DND-99 (Molecular Probes L7528), and immediately imaged after incubation. For the inhibition of PTEN, the salivary glands were incubated 30 min at RT with 10 µM VO-OHpic trihydrate (Santa Cruz Biotechnology sc-216061). DMSO was used as vehicle and its final concentration was 0.25 µL/mL in Grace's medium. Images are representative of 3 independent experiments, with at least three technical replicates in each experiment.

## Western blot

Control and experimental larvae were collected as described above. At least 15 whole salivary glands were dissected per genotype on ice in 1xPBS, immediately frozen in liquid nitrogen and kept at $-80°C$. For protein extraction, the glands were homogenized with a plastic pestle in 1% PBT lysis buffer and pelleted at 20,000 × g for 5 min at 4°C. Protein content from recovered supernatants was measured using BCA (manufacturer protocol, Invitrogen) and equal protein amounts were loaded per lane and separated on 12.5% SDS-PAGEs. Proteins were transferred to nitrocellulose membranes, blocked with 5% milk powder in 0.1% Triton X-100 in 1xPBS and blots were probed for GFP (rabbit anti-GFP 1:2000, Molecular Probes A11122), Crb (rat anti-Crb2.8 1:1000, see supplementary *Table 3*) and Tubulin (mouse anti-αTubulin 1:1000, see supplementary *Table 3*).

## Food intake assay and puparium formation rate

For the food intake assay we adapted a protocol reported by *Deshpande et al. (2014)*. Briefly, eggs from the appropriate genotypes were collected overnight on apple juice agar plates and transferred into normal food containing blue bromophenol (500 mg/L). As indicated in *Figure 8*, after 2 days of incubation at 29°C, larvae were briefly rinsed in iced cold PBS to remove attached food. Then, for each replica, 60 larvae were manually transferred into an Eppendorf tube containing 220 µL PBS + 0.1% Triton X-100 (PBST), and frozen immediately in dry ice. The samples were thawed and homogenized with a rotor pestle, centrifuged at 10 000 x g at 4°C for 10 min. The supernatant was diluted 1:2 into PBST for absorbance measurement at 680 nm. The standard curve was made by diluting 200 µL of liquefied bromophenol-containing food into 800 µL PBST, mixed in a ThermoMixer (Eppendorf, Germany) block at 900 rpm 80°C for 30 min, followed by centrifugation at 10 000 x g. The supernatant was serially diluted in PBST and the serial dilutions measured at 680 nm using a FLUOstar Omega (MBG Labtech, Germany).

For the assessment of the pupariation rate, eggs from the appropriate genotypes were collected for one hour on apple juice agar plates. Afterwards, 20 eggs were transferred to a new apple juice plate containing fresh yeast paste. To score the puparium formation, the plates with the embryos were incubated at 29°C for 72 hr and afterwards were assessed every 3 hr (excluding the overnight period). All newly appearing pupae were counted until all larvae had pupariated. To determine the puparium formation rate, the number of newly formed pupae at a given time point are divided by the total number of pupated animals. For the graphs of larval development speed (*Figure 1H*, *Figure 1—figure supplement 1KK*, *Figure 6W*) percentages were added up for the consecutive time points (also see source data).

## Statistical analyses

All statistical analyses were performed using GraphPad Prism 8. Statistical significance was calculated in unpaired *t*-test or a one-way analysis of variance (ANOVA) followed by a Dunnett's multiple-comparison when experimental groups are specifically compared only to control conditions, or a Tukey's multiple comparison test when all groups are compared to each other. *P* values are indicated in each corresponding graph.

## Acknowledgements

We thank S Eaton, A Wodarz, GJ Beitel, G Thomas, M Furuse, and U Tepass for kindly providing antibodies for our studies. We thank the MPI-CBG facilities: Patrick Keller, antibody facility, for antibody production; Light Microscopy Facility, in particular Jan Peychl and Sebastian Bundschuh, for microscopy guidance; the Electron Microscopy Facility, in particular Michaela Wilsch-Bräuninger and Tobias Fürstenhaupt, for discussions and troubleshooting. Stocks were obtained from the Bloomington *Drosophila* Stock Center (NIH P40OD018537) and from the Vienna *Drosophila* Resource Center. Antibodies were obtained from the Developmental Studies Hybridoma Bank, created by the NICHD of the NIH and maintained at The University of Iowa, Department of Biology, Iowa City, IA 52242. We thank the fly keepers Sven Ssykor, Cornelia Mass and Stefan Wernicke for excellent care of our flies. MB grants BR5490/2, BR5490/3. The work was supported by the Max-Planck Society and the Deutsche Forschungsgemeinschaft. (DFG)

## Additional information

### Competing interests

Elisabeth Knust: Reviewing editor, *eLife*. The other authors declare that no competing interests exist.

## Funding

| Funder | Grant reference number | Author |
|---|---|---|
| Max-Planck-Gesellschaft | | Johanna Lattner<br>Weihua Leng<br>Elisabeth Knust<br>David Flores-Benitez |
| Deutsche Forschungsgemeinschaft | BR5490/2 | Marko Brankatschk |
| Deutsche Forschungsgemeinschaft | BR5490/3 | Marko Brankatschk |

The funders had no role in study design, data collection and interpretation, or the decision to submit the work for publication.

## Author contributions

Johanna Lattner, Conceptualization, Formal analysis, Investigation, Visualization, Methodology, Writing—original draft, Writing—review and editing; Weihua Leng, Investigation; Elisabeth Knust, Conceptualization, Resources, Funding acquisition, Writing—review and editing; Marko Brankatschk, Conceptualization, Resources, Funding acquisition, Investigation, Methodology, Writing—original draft, Project administration, Writing—review and editing; David Flores-Benitez, Conceptualization, Data curation, Formal analysis, Supervision, Investigation, Visualization, Methodology, Writing—original draft, Project administration, Writing—review and editing

## Author ORCIDs

Johanna Lattner (iD) http://orcid.org/0000-0003-3421-9134
Elisabeth Knust (iD) http://orcid.org/0000-0002-2732-9135
Marko Brankatschk (iD) https://orcid.org/0000-0001-5274-4552
David Flores-Benitez (iD) https://orcid.org/0000-0001-8244-9335

## Decision letter and Author response

Decision letter https://doi.org/10.7554/eLife.50900.sa1
Author response https://doi.org/10.7554/eLife.50900.sa2

# Additional files

## Supplementary files

• Transparent reporting form

## Data availability

We provide as source data files all the data used for statistical analyses and generation of all graphs. These files are sorted according to the figure and the corresponding figure supplements, which correspond to Figure 1 and its supplements, Figure 2, Figure 3 and its supplements, Figure 5 and its supplements, and Figure 6.

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
