## [Decision Letter]

**Acceptance summary:**

Crumbs is a widely studied molecule for its role in defining apical aspects of epithelial organisation. However, this study distinguishes itself in convincingly demonstrating that Crb (and other components of the Crb complex) are required for the apical organization of β-Spectrin and actin, which in turn are required for the recruitment of Myosin 5 and the Rabs involved in apical secretion (Rab6, Rab11 and Rab30). Combined with data that link lipid phosphatase function, these studies potentially reveal a mechanistic aspect of Crb in the organisation of the apical domains of polarized epithelia, linking Crb activity to both organization of the cytoskeleton and to lipid homeostasis. Given the importance of CRB and these processes to development and cancer, this work demands further analysis and validation.

**Decision letter after peer review:**

[Editors’ note: this article was originally rejected after discussions between the reviewers, but the authors submitted for reconsideration. The first decision letter after peer review is shown below.]

Thank you for submitting your work entitled "Crumbs organizes the apical transport machinery by negatively regulating Pten in *Drosophila* larval salivary glands" for consideration by *eLife*. Your article has been reviewed by a Senior Editor, a Reviewing Editor, and three reviewers. The reviewers have opted to remain anonymous.

Our decision has been reached after consultation between the reviewers. Based on these discussions and the individual reviews below, we regret to inform you that your work will not be considered further for publication in *eLife* at this time.

As you can see from the reviews, your manuscript generated a lot of discussion and suggestions. Following extensive post review consultation, the referees were unanimous in their view that the work is not far enough along to warrant an *eLife* publication at this time. Also, since the questions raised pertain as deeply to mechanistic issues, as they do to specific experiments, it was not possible to create a simple set of experiments doable over a two-month revision period that would satisfy the reviewers. This sentiment was more apparent in the discussions than even those in the reviews below. As you know, *eLife* policy does not recommend multiple revisions, and the reviewers (including reviewer 3, who was more positive) really felt that it will not be fair to the authors to unreasonably delay the process if you wish to submit elsewhere.

Reviewer #1:

The article by Lattner, Leng, Knust, Brankatschk and Flores-Benitez describes the role of Crumbs in organizing the apical membrane of the *Drosophila* salivary gland cells.

Crumbs is a known apical large transmembrane protein that interacts with Stardust and binds FERM proteins via its FERM binding domain. It also binds β-Spectrin.

In the first part of the manuscript, the authors show that in salivary glands, loss of Crumbs does not affect the localization of cytoplasmic apical markers but affect the delivery of a generic TM protein CD8-RFP. Loss of Crumbs does not affect AJ but affects a bit SJ, albeit not the extent of compromising the epithelial integrity.

However, loss of Crumbs leads to the appearance of plasma membrane apical sacs (PAMS), per apical PM. PAMS bears all the characteristics to PM including the presence of some microvilli.

Loss of Stardust leads to similar defects. Loss of β-Spectrin, also, albeit to a lesser extent. Crumbs now present in these PAMS.

Moe binds Crumbs and loss of Moe leads to similar phenotype to a certain extent. The results are not as convincing.

In the second part of the manuscript, in what appears to be a jump from one subject to another, the authors examine the localization of many Rabs and found that the localization of 3 is particularly affected in a Crumbs mutant. Namely the localization of Rab6, Rab11 and Rab30 near the apical cortex is lost and that they end up at the basal side.

In a third part, the authors address the role of PI(4,5)P_2_ in the apical membrane. For this, they use a probe that specifically binds PI(4,5)P_2_ and show that the probe localizes mostly to the apical membrane. In the absence of Crumbs, PI(4,5)P_2_ is no longer localized apically. Inhibiting Pten pharmacologically and genetically modulates the pool of PI(4,5)P_2_ at the apical membrane. Importantly, Pten depletion suppress the formation of the PAMs that are induced by Crumbs depletion.

They conclude that Crumbs through its binding to Stardust, β-Spectrin and Moesin controls integrity of the apical membrane by downregulating Pten activity. When Pten is no longer downregulated, it leads to apical membrane extension leading to PAMS.

The manuscript represents a lot of work combining quite a lot of techniques from fly genetics to light and electron microscopy. It is also well performed. As it is, however, it is hard to get a sense of what the paper is about, what the question is, what is new and important, and overall how Crumbs is required in the maintenance of the apical membrane microvilli and prevents the formation of plasma membrane sacs (PAMS). The story telling appears disconnected. The depletion of many factors leads to the formation of PAMS but the reader is not given a mechanistic sense of how these factors are connected in promoting PAMS formation.

Second, the relationship between apical secretion and PAMS formation is not clear. As a result, the reviewer does not have a sense of their role. How are they connected to apical secretion of saliva formation? True the larvae grow slower by a couple of hours but how does it matter?

The reviewer is not even sure what are the PAMS about. Are they a consequence of PM expansion? If so, why does it invaginate inward? Also, the microvilli do not form. Is it how the PM looks elongated? Why are the microvilli not forming? Are TM proteins such as Prominin not delivered properly? Is this lack of delivery related to PI(4,5)P_2_ level? How does Crumbs regulates Pten activity?

Third, none of the microscopy is not quantified. Especially PAMS appear quite heterogeneous, at least from one genotype to the next. Are they all the same? How penetrant?

Last, some sections of the manuscript are written in a pushy manner and this should be remedied.

These points are detailed below:

1) The choice of apical cargo is problematic. Both rely on generic probes.

1.1) The first is CD8-RFP that is not an endogenous protein and that is by default transported to the apical membrane.

What are the dots observed in Crumbs mutant?

1.2) The second is a lectin, PNA that binds the carbohydrate sequence Gal-β(1-3)-GalNAc.

The reviewer would perhaps understand the figure better if labeled PNA is used as a labeling tool (as an antibody). Instead PNA is expressed as a GFP fusion protein under the control of UAS. As it does not have a TM segment, it would localize to the cytoplasm of the salivary gland cells.

If this is the case, the reviewer does not understand how it could bind to glycoproteins as their sugar moieties are in the lumen of organelles (Golgi) and facing the extracellular medium, so away from PNA.

As a result, the reviewer is mighty puzzled as that the dots represent in WT salivary glands Furthermore, the reviewer does not understand the PNA pattern observed in Crumbs mutant? The bright PNA patches appear to be facing the lumen of the tube. Why would glycoproteins be now present at the PM? The reviewer has no notion of what is observed in this figure in term of glycoprotein transport. Is PNA secreted at all? How?

It would not matter much but these two tools are used at many places in the manuscript and the conclusions made are strong. Therefore, they need to be documented in a much better fashion and establish good standard for using this proxy.

1.3) Furthermore, while a large part of the manuscript is performed with endogenous saliva proteins (or endogenously tagged in the genome), why did the authors not study endogenous cargo like the saliva proteins Sgs1-8 (at least some of them). Or the ORF encoded by the gene Saliva.

These are obviously apical cargos and would be greatly preferable than the artificial ones used in the manuscript. Their secretion should be observed in Crumbs mutant. This has been done indirectly in weighing the larvae that cannot forage in the food.

1.4) Also, intriguingly, Crumbs remodeling of the apical PM only occurs in foraging larvae, not wandering when they start synthesizing and secreting the glue. Yet, is Glu protein secretion impaired in Crumbs mutant?

2) Related to the choice of cargo, one clear phenotype of the loss of Crumbs is the loss of microvilli.

2.1) Is it how the PM appears elongated? By losing microvilli?

2.2) Why are the microvilli not forming? Are TM proteins such as Prominin not delivered properly? In mammals, it is clearly required for microvilli formation. Prominin and prominin-like should definitely be followed in WT and Crumbs mutants (in fact prominin appears to genetically interact with Crumbs). The prediction is that it should be delivered to the apical membrane and WT and not in Crumbs mutants.

2.3) The density of the microvilli needs to be quantified in WT and Crumbs mutants. Only then can the authors say that the decrease is dramatic.

3) The Crumbs phenotype is clearly different from embryos where Crumbs have a function in keeping the apico-basal polarity intact as well as the cell adhesion. The reviewer is confused as to why this is and what underlies the difference? Is it because of the absence of microvilli?

Can this be tested? Can other hypothesis put forward to explain the differences? What about other epithelia such as the follicle cells?

4) The PAMS are inward invaginations. But the degree to which they appear depends on the genetic background. They sometimes appear big and deep and sometimes not.

4.1) This needs to be seriously quantified for each background, for instance how many cells exhibit a shallow invagination, a deeper one and a very deep pit which clear morphological definition? For instance, some depletion in WT leads to shallow invaginations. Are those significant?

4.2) Are some PAMs found completely sealed and no longer accessible? This should also be quantified as it is much more connected to the disease.

4.3) If most of them are continuous with the PM, what is their relevance? What does it affect secretion? Or are the two phenotypes not connected? In this regard, looking at prominin or any other TM proteins involved in microvilli formation would result in a clearer understanding of the message.

5) In subsection “Crb controls the apical membrane homeostasis by negatively regulating Pten activity”, the authors state “in addition, these results show that the Crb protein complex is essential for the regulation of Pten activity in SGs.”

“….suggests that accumulation of PI(4,5)P_2_ in Crb KD glands is likely due to an increase in Pten activity rather than a change in its localization.”

To state this, Pten activity should be measured directly, using extracts from WT and Crumbs mutant salivary glands with known substrates such as PI(3,4,5)P_3_. A probe assessing PI(3,4,5)P_3_ level should work if this cannot be done in vitro.

6) Pten is involved in many different pathways. It is a tumor suppressor and acts in negatively regulating the AKt pathway.

6.1) Is there a link between the PAMS and the other phenotype?

6.2) Is there any thoughts or data on how Crumbs regulates Pten activity? Do they form a complex

7) Rabs:

7.1) The reviewer suggests to put this part after the PI(4,5)P_2_/Pten section as the transition from Myo5 to Rabs is really abrupt and does not make a lot of sense.

7.2) Why was Rab35 not considered as it also loses its apical localization in Crumbs mutant?

7.3) What is the role of these Rabs near the apical membrane? Are they active? Rabs mediate membrane traffic steps and these in particular from the TNG to the PM (Rab6) or from endosomes to the PM (Rab11), but none of this is described and tested. How does loss of Crumbs leads to their mislocalisation? Is it only PI(4,5)P_2_ composition? How?

Is any of the Rab depletion lead to PAMs formation?

So overall, the manuscript is the description of a number of factors (related to the cytocortex-Rab network) whose depletion leads to PAMS. But there is very little mechanistic insights or sense of what is new.

Reviewer #2:

In this paper, the authors remove Crb (or other Crb complex protein) function in mid-third instar larval SGs and determine the consequences. What they observe is that many aspects of SG organization are unaffected; overall SG morphology, cell polarity, barrier function, as well as the localization of a number of membrane domain-specific proteins appear largely normal. Apical secretion of both an apical membrane protein (CD8-RFP) and glycoproteins (those detected by PCNA binding) are diminished, however, with the reduction in Crb or with the reduction of Sdt (a component of the Crb complex). Reduced Crb also results in decreases in apical accumulation of F-actin, β_Heavy_-Spectrin and MyoV (the myosin that has been implicated in secretion). Also observed are decreases in apical localization of three Rab proteins that have previously been implicated in apical secretion: Rab6, Rab11 and Rab30. The reduction in Crb also results in the invagination of the apical domain to form structures the authors refer to as PAMS (PI(4,5)P_2_ and Phospho-Moesin enriched apical membrane sacs). To make sense of the observed molecular changes, the authors also determined the effects of reducing β_Heavy_-Spectrin, MyoV, and the enzymes involved in controlling PI(4,5)P_2_ levels in cells: Skittles (a PIP kinase that makes PI(4,5)P_2_), Pten (which dephosphorylates PI(3,4,5)P_3_ to make PI(4,5)P_2_) and PI3K (which phosphorylates PI(4,5)P_2_ to make PI(3,4,5)P_3_). The reduction in β_Heavy_-Spectrin and of MyoV also diminishes secretion and promotes formation of PAMs, although not to the same level as reduction in Crb. The authors also show that reduction of Pten activity as well as increases in P13K can reduce the formation of PAMs associated with reduced Crb function, as well as restoring the apical localization of Rab6 and Rab11. Based on these experimental findings, the authors propose the following model: Crb is required to maintain the proper amount and organization of the apical domain by stabilizing the apical cytocortex (through β_Heavy_-Spectrin and F-actin), that the Crb-β_Heavy_-Spectrin complex facilitates apical secretion in larval SGs by maintaining apical MyoV. The authors also conclude that Crb normally limits PI(4,5)P_2_ accumulation by limiting Pten activity (but not Pten accumulation since levels appear unchanged between controls and Crb knockdown).

Essential revisions:

1) Most of the phenotypes are not quantified and should be. When the authors say that PAM formation is reduced or increased or that secretion is reduced, we are relying on the limited number of images that can be shown and an impression the authors have of the data, rather than any quantification.

2) The authors make a good case that PI(4,5)P_2_ is required for the formation of PAMs, but they do not make a very good case that Crb works through Pten to increase PI(4,5)P_2_ levels. From the images provided, it is not at all clear that overall apical PI(4,5)P_2_ levels are any higher in the Crb knockdown than in WT control SGs, although the apical distribution does seem to change – less in the PM and more in PAMs. Reducing PI(4,5)P_2_ (which is known to bind and activate Moesin) does indeed appear to reduce PAM formation and increasing PI(4,5)P_2_ does appear to increase PAM formation but that does not indicate that Crb normally acts to limit Pten activity; it only says that high levels of PI(4,5)P_2_ are required for this loss of Crb phenotype. Indeed, the Pten overexpression phenotypes does seem to increase the apical domain area but it doesn't seem to create structures that have the same morphology as the PAMs. So, either change the conclusions/title of the manuscript or provide better evidence that PI(4,5)P_2_ levels are indeed higher with loss of *crb*.

3) Skittles is not the only PIP kinase encoded in the *Drosophila* genome, perhaps explaining the relatively mild effects of skittles knockdown on the loss of Crb phenotypes. The authors should acknowledge the existence of this other protein when they note the relatively mild effects.

4) Although the authors claim that loss of Crb does not affect AJs (based on TEMs and *D*E-cad immunostaining), it does appear from the images that levels of *D*E-cad found near the apical surface are notably reduced (perhaps there is also an increase in vesicular staining of *D*E-cad). Thus, the authors need to look at this more carefully and indicate that there is some reduction of the AJ-localized *D*E-cad, or clearly demonstrate that levels are unaffected.

5) There seem to be some PAMs with microvilli-like structures and other similarly large vesicular structures with no microvilli-like structures. Are the latter only found with Crb knockdown or are they also found in controls? If they are only found in the Crb knockdown, what are they proposed to be?

6) This may be challenging to determine, but it would be helpful to know if the Crb knockdown actually changes the total amount of apical surface area. Do control SG cells have the same amount of apical area if, in the Crb knockdown cells, one were to include the apical area found within the PAMs? This is related to the observation that the pleated septate junctions in the Crb knockdown SGs seem to be much more convoluted than those in WT (Figure 1—figure supplement 2), suggesting that there might be an increase in membrane in those domains. This might suggest that with loss of Crb, some of the apical membrane is being converted to basolateral membrane.

7) Importantly, is this study showing us that Crb is doing what it has been known to do – controlling the balance of apical versus basolateral domains and providing some clues as to the mechanism or is this study really uncovering a new and unrelated role for Crb? Either way the study is interesting, but exploring the first possibility is worthwhile since it has not been clear how Crb normally controls apical specification.

Reviewer #3:

This manuscript explores the roles of the polarity protein Crumbs (Crb) in a particular type of secretory epithelium: the salivary glands of the *Drosophila* larva. The authors provide data consistent with a model whereby Crb regulates apical secretion by (1) maintaining the active pool of a subset of Rab proteins at the apical domain and (2) negatively modulating Pten activity, thereby modulating the lipid composition of the apical membrane. Based on a series of loss-of-function/knockdown experiments, the authors propose that both these roles are least in part mediated by the effects of Crb – complexed with β-Spectrin – on Myosin V localisation.

The manuscript is clearly written and the data is generally convincing. The following points/conclusions are only weakly supported by the data provided in the current version of the manuscript, and would benefit from a few additional experiments:

1) All conclusions re: *crb* loss-of-function phenotypes (e.g. "defects in apical secretion are not due to an overall disruption of cell polarity", or "this strategy does not affect embryonic development") are currently based on knockdown experiments. These may only reduce – rather than abrogate – Crb expression, and do so at a currently undefined developmental point. Clonal mutation of *crb* in salivary glands and/or temporally controlled knockdown experiments would allow the authors to draw stronger conclusions about the roles of Crb (or lack thereof) in establishing vs. maintaining overall epithelial polarity in the salivary gland, as well as their temporal requirement. Related to this, different *fkh*-Gal4s lines have been generated with distinct expression. Is the expression of the one used in this particular study really confined to salivary glands? Most lines are also expressed in other portions of the intestine.

2) The authors explore physiological aspects of Crb-mediated salivary gland secretion very superficially. This seemed to me a missed opportunity. If they wish to make any statements about Crb regulating the "physiological activity of the salivary gland" or promoting "efficient apical secretion of glycoproteins" (as currently stated in the abstract), the authors need to show effects on endogenous secretion – for example by visualizing the effect of *crb* knockdown on a protein normally secreted by the salivary gland. Related to this, the authors assume but do not show a reduction in food intake following salivary gland-specific *crb* knockdown. Is food intake reduced following *crb* knockdown? Does Pten knockdown rescue the feeding and developmental timing defects of *crb* knockdown larvae? Finally, what happens to Crb expression/localisation and/or downstream targets around the saliva to glue switch? In other words, is Crb permissive or instructive in the context of salivary gland secretion?

3) The authors should try to rule in/out increased endocytosis (rather than reduced apical secretion) as one of the reasons for the observed phenotypes; it seems to me that at least some of the Rab targets are involved in recycling as well as secretion. Also, the appearance of these novel intracellular microvilli-containing sacs and the concurrent reduced density of microvilli on the cell surface may both also be consistent with increased endocytosis; I am not an expert in this but, as far as I know, microvilli are not "trafficked" to the membrane in secretory compartments? Related to this, it may be informative to test the Rab signature of PAM sacs in both Crb/β-Spectrin knockdowns – might this shed light on the origin of these compartments?

---

## [Author Response]

Reviewer #1:The article by Lattner, Leng, Knust, Brankatschk and Flores-Benitez describes the role of Crumbs in organizing the apical membrane of the Drosophila salivary gland cells.Crumbs is a known apical large transmembrane protein that interacts with Stardust and binds FERM proteins via its FERM binding domain. It also binds β-Spectrin.In the first part of the manuscript, the authors show that in salivary glands, loss of Crumbs does not affect the localization of cytoplasmic apical markers but affect the delivery of a generic TM protein CD8-RFP. Loss of Crumbs does not affect AJ but affects a bit SJ, albeit not the extent of compromising the epithelial integrity.However, loss of Crumbs leads to the appearance of plasma membrane apical sacs (PAMS), per apical PM. PAMS bears all the characteristics to PM including the presence of some microvilli.Loss of Stardust leads to similar defects. Loss of β-Spectrin, also, albeit to a lesser extent. Crumbs now present in these PAMS.Moe binds Crumbs and loss of Moe leads to similar phenotype to a certain extent. The results are not as convincing.

We have the impression that the reviewer is mistaken at this point. Although certainly an interesting idea, we did not and do not show any data for silencing of Moesin, but for silencing Myosin V. We used Moesin or phospho-Moesin as marker for the apical cortex.

In the second part of the manuscript, in what appears to be a jump from one subject to another, the authors examine the localization of many Rabs and found that the localization of 3 is particularly affected in a Crumbs mutant. Namely the localization of Rab6, Rab11 and Rab30 near the apical cortex is lost and that they end up at the basal side.

Rab6 and Rab11 have a complex cellular distribution, as these proteins localize close to the apical plasma membrane, but they are also found associated with punctae-like compartments throughout the cell (Dunst et al., 2015). In Crumbs knock-down exclusively the apical portion of Rab6, 11 and 30 is lost, without affecting their localization elsewhere in the cell. Thus, we would like to clarify that we do not observe nor stated that they “end up at the basal side”.

*In a third part, the authors address the role of PI(4,5)P_2_* in *the apical membrane. For this, they use a probe that specifically binds PI(4,5)P_2_ and show that the probe localizes mostly to the apical membrane. In the absence of Crumbs, PI(4,5)P_2_ is no longer localized apically. Inhibiting Pten pharmacologically and genetically modulates the pool of PI(4,5)P_2_ at the apical membrane. Importantly, Pten depletion suppress the formation of the PAMs that are induced by Crumbs depletion.*

Our results show that PI(4,5)P_2_ apical levels are increased in the absence of Crumbs. This was stated along the first version of the manuscript and illustrated as well (old Figure 5). However, to solidify our findings, we added the quantifications (Figure 5 of the current version) of our microscopy data.

They conclude that Crumbs through its binding to Stardust, β-Spectrin and Moesin controls integrity of the apical membrane by downregulating Pten activity. When Pten is no longer downregulated, it leads to apical membrane extension leading to PAMS.The manuscript represents a lot of work combining quite a lot of techniques from fly genetics to light and electron microscopy. It is also well performed. As it is, however, it is hard to get a sense of what the paper is about, what the question is, what is new and important, and overall how Crumbs is required in the maintenance of the apical membrane microvilli and prevents the formation of plasma membrane sacs (PAMS). The story telling appears disconnected. The depletion of many factors leads to the formation of PAMS but the reader is not given a mechanistic sense of how these factors are connected in promoting PAMS formation.

In brief, Crb is a well characterized determinant essential for the establishment of apico/basal polarity. However, it is less clear to what extent Crb is required to maintain the apical identity in polarized cells and if so, how Crb mechanistically regulates the apical membrane domain. Here we show in *Drosophila* salivary gland cells that Crb negatively regulates the phosphatase Pten required to maintain the apical PI(4,5)P_2_/PI(3,4,5)P_3_ ratio essential for the membrane identity. Moreover, deregulated Pten activity is changing the localization of (at least) the apical Rab machinery and disrupts the link between the plasma membrane and the cytoskeleton. In consequence, the apically-targeted secretion of proteins is impaired resulting in consequences for the whole organism.

Second, the relationship between apical secretion and PAMs formation is not clear. As a result, the reviewer does not have a sense of their role. How are they connected to apical secretion of saliva formation?

The reviewer is correct, it is not easy to explain the formation of PAMS with our results at hand. PAMS are membrane sacs enriched in PI(4,5)P_2_ and Phospho-Moesin that are never present in control salivary gland cells (see subsection “Crb regulates apical membrane levels of PI(4,5)P_2_”). We consider PAMS as morphological manifestations of failed apical membrane transport, although we cannot completely exclude the possibility that a weakened apical cytocortex contributes to their formation. In fact, the link between PAMS, secretion and PI(4,5)P_2_ levels is such that when PI(4,5)P_2_ levels are lowered in Crb-depleted SGs (by knocking down Pten or by using its inhibitor VO-OHpic), the formation of PAMS is suppressed (Figure 5) as well as the secretion defects (Figure 6).

True the larvae grow slower by a couple of hours but how does it matter?The reviewer is not even sure what are the PAMS about. Are they a consequence of PM expansion? If so, why does it invaginate inward? Also, the microvilli do not form. Is it how the PM looks elongated? Why are the microvilli not forming? Are TM proteins such as Prominin not delivered properly? Is this lack of delivery related to PI(4,5)P2 level? How does Crumbs regulates Pten activity?Third, none of the microscopy is not quantified. Especially PAMS appear quite heterogeneous, at least from one genotype to the next. Are they all the same? How penetrant?Last, some sections of the manuscript are written in a pushy manner and this should be remedied.These points are detailed below:

Detailed answers to these points are given below in the specific comments.

Our experimental approach is based on the organ-specific KD of Crb. As a matter of fact, induced systemic Crb knock-down is lethal. The defects in SG secretion induced by loss of Crb do matter at the level of the whole organism, as we show by quantifying the effects on developmental speed (Figure 1H, Figure 1—figure supplement 1KK and Figure 6W) and food intake (Figure 6V). While others have suggested that SGs might be dispensable for larval survival (Jones et al., 1998), our assays are sensitive enough to report these effects. It is important to note that, in the work of Jones et al. (1998), several technical details of the assays reported in Table 2 are missing (for example, stage of the larvae when isolated, duration of the assay, and incubation conditions), but nevertheless they report a lethality of 72.8% in trans-heterozygous *eyg^C1^*/*eyg^C53^*ductless larvae, which highlights the importance of the SGs in larval survival.

We re-structured our manuscript to improve the delivery of our message. We addressed the role of the Rab proteins in the formation of the PAMS (Figure 4—figure supplement 3) and the relevance of proper secretion for the pupation rate and food intake (Figure 6V,W). We included an analysis of the microvilli density (Figure 2M) as well as the size and frequency of PAMS (Figure 5—figure supplement 1E and Subsection “Crb regulates apical membrane levels of PI(4,5)P_2_”) and included these points along our Discussion about their significance and their relation to the secretion defects.

Instead of Prominin, we analyzed the localization of endogenous Cadherin99C, a TM protein regulating microvilli structure (Figure 1I,J and Figure1—figure supplement 1CC,DD).

We provide additional evidence that Crb regulates PI(4,5)P_2_ levels by regulating Ocrl apical localization (Figure 5L-Q) and updated our Discussion about the connections between Crb, β_Heavy_-Spectrin and MyosinV with Pten2, and about the role of Crb in maintaining the localization of Pten at the apical membrane (Figure 5I-K). We provide quantifications (and the corresponding source data) to most relevant microscopy data.

1) The choice of apical cargo is problematic. Both rely on generic probes.1.1) The first is CD8-RFP that is not an endogenous protein and that is by default transported to the apical membrane.

We include the analysis of two different apical proteins. By immunolocalization of endogenous Cadherin99C and live imaging of a Venus-tagged version of Stranded at Second, we confirmed our previous results using CD8-RFP (Figure 1, Figure 1—figure supplement 1, Figure 2 and Figure 3).

What are the dots observed in Crumbs mutant?

These represent most likely endosomes and lysosomes. Similar punctae have also been observed in models for MVID (see for example Mosa et al., 2018). Other published work also suggests that impaired secretion results in enhanced lysosomal activity to degrade the accumulated material. We have included this point in subsection “Crb regulates apical membrane organization via the apical cytocortex”.

1.2) The second is a lectin, PNA that binds the carbohydrate sequence Gal-β(1-3)-GalNAc.The reviewer would perhaps understand the figure better if labeled PNA is used as a labeling tool (as an antibody). Instead PNA is expressed as a GFP fusion protein under the control of UAS. As it does not have a TM segment, it would localize to the cytoplasm of the salivary gland cells.If this is the case, the reviewer does not understand how it could bind to glycoproteins as their sugar moieties are in the lumen of organelles (Golgi) and facing the extracellular medium, so away from PNA.As a result, the reviewer is mighty puzzled as that the dots represent in WT salivary glands Furthermore, the reviewer does not understand the PNA pattern observed in Crumbs mutant? The bright PNA patches appear to be facing the lumen of the tube. Why would glycoproteins be now present at the PM? The reviewer has no notion of what is observed in this figure in term of glycoprotein transport. Is PNA secreted at all? How?It would not matter much but these two tools are used at many places in the manuscript and the conclusions made are strong. Therefore, they need to be documented in a much better fashion and establish good standard for using this proxy.

We used the PNA line listed in the Bloomington Stock Center as a tool to analyze glycoproteins. However, the FlyBase has no further information to clarify the concerns raised by the reviewer. Therefore, we opted to use other available lines to analyze apical secretion. After testing different lines, we decided to repeat all experiments using a construct containing the chitin binding domain of Serpentine tagged with GFP (UAS-SerpCBD-GFP). This construct is a well-established apical secreted cargo and has been used as a proxy to analyze apical secretion (Luschnig et al., 2006; Kakihara et al., 2008; Förster et al., 2010; Petkau et al., 2012; Dong et al., 2013; Dong et al., 2014; Bätz et al., 2014). Using this tool, we confirmed the results obtained with the PNA line, and thus exchanged all figures with the ones obtained with SerpCBD-GFP (Figure 1, Figure1—figure supplement 1, Figure 3, Figure 3—figure supplement 2, Figure 6).

1.3) Furthermore, while a large part of the manuscript is performed with endogenous saliva proteins (or endogenously tagged in the genome), why did the authors not study endogenous cargo like the saliva proteins Sgs1-8 (at least some of them). Or the ORF encoded by the gene Saliva.These are obviously apical cargos and would be greatly preferable than the artificial ones used in the manuscript. Their secretion should be observed in Crumbs mutant. This has been done indirectly in weighing the larvae that cannot forage in the food.

As we described in the manuscript, we analyzed the salivary glands at a stage where the larvae are feeding, which is almost two days before the expression of Sgs1-8 proteins is detectable. Therefore, none of the glue proteins are useful for our analyses, and indeed, glue secretion is not affected by the loss of Crb (see Video 1 and Video 2). Regarding the suggested ORF encoded by the gene Saliva, it is important to note that the product of this gene does not form part of the saliva, but it is a transmembrane transporter of sugar that is highly expressed in the salivary gland (FlyBase). Due to this fact, it was named Saliva, but it is not predicted to be a secreted protein. See also answer to reviewer #3 point 2.

1.4) Also, intriguingly, Crumbs remodeling of the apical PM only occurs in foraging larvae, not wandering when they start synthesizing and secreting the glue. Yet, is Glu protein secretion impaired in Crumbs mutant?

Glue secretion (monitored following Sgs3-GFP) occurs normally (see Video 1 and Video 2). In fact, pupae attach properly to the walls of the vials.

2) Related to the choice of cargo, one clear phenotype of the loss of Crumbs is the loss of microvilli.2.1) Is it how the PM appears elongated? By losing microvilli?

We have quantified the density of microvilli (Figure 2M). Unfortunately, we cannot measure the *absolute* amount of apical and lateral membranes, including the amount of membrane in the microvilli. For this, specific algorithms and computational analysis would be necessary, which is out of the scope of our study. As to how or why the microvilli are not forming, we have included different possibilities into the updated Discussion section.

2.2) Why are the microvilli not forming? Are TM proteins such as Prominin not delivered properly? In mammals, it is clearly required for microvilli formation. Prominin and prominin-like should definitely be followed in WT and Crumbs mutants (in fact prominin appears to genetically interact with Crumbs) The prediction is that it should be delivered to the apical membrane and WT and not in Crumbs mutants.

We appreciate the suggestions of using Prominin to analyze the microvilli phenotype, however this protein is barely expressed in the salivary glands according to FlyBase. Indeed, the genetic interaction between Prominin and Crumbs has been reported in the eye, where Prominin is very abundant. Instead, we decided to analyzed the localization of Cadherin99C, a protein that regulates microvillar length (Chung and Andrew, 2014 and references within). Our results indicate that Cad99C is not properly localized to the apical aspect of Crb deficient cells (Figure 1 and Figure 1—figure supplement 1) and we included a Discussion section about these results.

2.3) The density of the microvilli needs to be quantified in WT and Crumbs mutants. Only then can the authors say that the decrease is dramatic.

We have quantified the density of microvilli (Figure 2M).

3) The Crumbs phenotype is clearly different from embryos where Crumbs have a function in keeping the apico-basal polarity intact as well as the cell adhesion. The reviewer is confused as to why this is and what underlies the difference? Is it because of the absence of microvilli?Can this be tested? Can other hypothesis put forward to explain the differences? What about other epithelia such as the follicle cells?

If the reviewer refers to the work by Chartier et al., 2011, which proposes that Crumbs controls epithelial integrity by inhibiting Rac1 and PI3K, it is difficult to provide a concrete answer. That work relies mostly on genetic interactions and analyses of cuticle phenotypes and did not analyze markers for PI(4,5)P_2_, cytoskeleton components, intracellular trafficking or showed any EM to get an idea of the defects at the cellular level. Therefore, it is difficult to speculate what part of the mechanisms described in our work might or might not be involved in the embryonic phenotype. Also see comment in point 6.1 below.

As for the follicle cells, the interactions between Crumbs, Moesin, aPkc and the cytoskeleton have been shown to be highly dynamic, during the morphogenesis of the follicles. In this case, the role of Crb is more complex, as this protein appears to have other roles in epithelial morphogenesis, e.g. in dorsal closure in the *Drosophila* embryo (Flores-Benitez and Knust, 2015, *eLife*). Further, it is well-known that Crb, and other polarity proteins, have tissue specific functions (Tepass, 2012; Flores-Benitez and Knust, 2016) some of which are mentioned in our Introduction and along our Discussion section.

4) The PAMS are inward invaginations. But the degree to which they appear depends on the genetic background. They sometimes appear big and deep and sometimes not.4.1) This needs to be seriously quantified for each background, for instance how many cells exhibit a shallow invagination, a deeper one and a very deep pit which clear morphological definition? For instance, some depletion in WT leads to shallow invaginations. Are those significant?

We have quantified the morphology and frequency of PAMS in the different genetic backgrounds and the results are shown in Figure 5—figure supplement 1E as well as in subsection “Crb regulates apical membrane levels of PI(4,5)P_2_” We have also addressed the significance of these results in the Discussion section. It is important to note that in some of the images it may appear that the PAMS seem to have different depths, but that is also due to the perspective generated in the single optical sections and maximal projections. The morphology of the PAMS can be better appreciated in the new Video 8.

4.2) Are some PAMs found completely sealed and no longer accessible? This should also be quantified as it is much more connected to the disease.

As recent work shows, these inclusion bodies are dynamic and can form at the apical or the basolateral membrane, but also in the cytoplasm (see Mosa et al., 2018). We have included this point in the Discussion section in context with the dynamics of the formation of the inclusion bodies. See also answer to reviewer #3 point 3.

4.3) If most of them are continuous with the PM, what is their relevance? What does it affect secretion? Or are the two phenotypes not connected? In this regard, looking at prominin or any other TM proteins involved in microvilli formation would result in a clearer understanding of the message.

See answer in point 2 above.

5) In subsection “Crb controls the apical membrane homeostasis by negatively regulating Pten activity”, the authors state “in addition, these results show that the Crb protein complex is essential for the regulation of Pten activity in SGs.”“….suggests that accumulation of PI(4,5)P_2_ in Crb KD glands is likely due to an increase in Pten activity rather than a change in its localization.”To state this, Pten activity should be measured directly, using extracts from WT and Crumbs mutant salivary glands with known substrates such as PI(3,4,5)P3. A probe assessing PI(3,4,5)P3 level should work if this cannot be done in vitro.

Although there are commercially available kits to measure PI(4,5)P_2_ and PI(3,4,5)P_3_ levels, they require a vast amount of sample material, which is not possible to isolate by manual dissection of the larval salivary glands. In addition, phosphoinositides are very transient molecules with short-life time ex vivo. We used a probe for quantifying PI(3,4,5)P_3_ levels and the results are shown in Figure 5—figure supplement 2D-F. We quantified also the PI(4,5)P_2_ levels by live imaging of glands expressing the probe for this phospholipid. Similarly, we quantified the distribution of Pten2-GFP and Ocrl-RFP (Figure 5). According to these measurements we made more conservative statements and updated our conclusions.

6) Pten is involved in many different pathways. It is a tumor suppressor and acts in negatively regulating the AKt pathway.6.1) Is there a link between the PAMS and the other phenotype?

Pten is a major negative regulator of the insulin pathway and its activity prevents the recruitment of the protein kinaseB/AKT to the plasma membrane. We performed Western blot analyses to investigate whether AKT phosphorylation (activity) is changed in Crb KD salivary glands. We tested the specificity of our assay by over-expressing PI3K, presuming that high PI3K levels will increase PI(3,4,5)P_3_ levels and activate AKT (i.e. GOF). Indeed, the PI3K gain of function increases both the salivary gland size (as expected from its role in cell growth – Scanga et al., 2000) and the amount of phosphorylated AKT (see Author response image 1, lane 1). However, wild type and Crb KD salivary gland cells do not show detectable AKT activity (lanes 3 and 4, respectively) although the enzyme is clearly present. Thus, we conclude that the PI(4,5)P_2_ increase in Crb deficient salivary gland cells does not seem to affect AKT signaling significantly.

Regarding the role of Pten as a tumor suppressor, we did not find any evidence suggesting that cell proliferation is modified by the loss of Crb, and as our results suggest that Pten activity is enhanced by the loss of Crb, it would be expected that tumorigenesis is downregulated, which is difficult to test in the context of our work, which was performed in a non-proliferating epithelium.

6.2) Is there any thoughts or data on how Crumbs regulates Pten activity? Do they form a complex

We performed immunoprecipitation experiments and found no complex containing Crumbs and Pten2. We also re-analyzed samples of mass spectrometry experiments of pull-downs from Crb and Sdt made in previous studies from our lab and did not find peptides for Pten2 or any other clear candidate that could mediate the regulation of Pten. The regulation of Pten is known to be extremely complex (see for example: PTEN Methods and Protocols, edited by Leonardo Salmena, DOI 10.1007/978-1-4939-3299-3) and we have addressed this point in the Discussion section, and provided references that identified β_Heavy_-Spectrin and MyosinV as interactors of Pten.

7) Rabs:7.1) The reviewer suggests to put this part after the PI(4,5)P_2_/Pten section as the transition from Myo5 to Rabs is really abrupt and does not make a lot of sense.

Thanks for the suggestion, we made the transition clearer.

7.2) Why was Rab35 not considered as it also loses its apical localization in Crumbs mutant?

The loss of apical Rab35 is not reproducible. Therefore, the corresponding panel in Figure 4—figure supplement 1 has been replaced for a representative image where the apical localization of Rab35 is visible in the Crb deficient glands.

7.3) What is the role of these Rabs near the apical membrane? Are they active? Rabs mediate membrane traffic steps and these in particular from the TNG to the PM (Rab6) or from endosomes to the PM (Rab11), but none of this is described and tested. How does loss of Crumbs leads to their mislocalisation? Is it only PI(4,5)P2 composition? How?Is any of the Rab depletion lead to PAMs formation?

Fluorescently YFP-tagged Rab proteins that we detect are membrane associated, active Rab proteins. They are GTP-bound and regulate a plethora of membrane trafficking steps. Cytoplasmic, inactive GDP-Rabs are complexed with “escort-proteins” dubbed GDI. To do so, Rab proteins represent localized binding platforms capable to recruit specific effector proteins (Caviglia et al., 2017). Interestingly, activated Rab6, Rab11, and probably Rab30, are found to bind MyoV (Lindsay et al., 2013). Thus, Rab proteins and Crb might share direct interactors. On the other hand, Crb is directly involved in the organization of the membrane-associated apical cytocortex (see Figure 2 and Figure 2—figure supplement 1). Most vesicular transport is executed by motor proteins which are bound to the tubulin (kinesin/dynein) or actin (myosins) meshwork. In consequence, the localization of Rab proteins close to the membrane domain is dependent also on the organization of the cytoskeleton (Eaton and Martin-Belmonte, 2014).

The phosphoinositide composition of the plasma membrane is known to regulate every step of the membrane traffic machinery (Mayinger, 2012; Martin, 2015; Posor, Eichhorn-Grünig and Haucke, 2015). As we write in our Introduction, “PI(4,5)P_2_ is directly implicated in the regulation of exocytosis (Milosevic et al., 2005; Gong et al., 2005; Massarwa et al., 2009; Rousso et al., 2013) and in all forms of endocytosis (Antonescu et al., 2011; Mayinger, 2012; Jost et al., 1998)”. However, as we also write in our Discussion section, we did not identify localization defects of Rab proteins involved in the endocytic route.

Regarding the role of the Rab proteins in the formation of PAMS, we show that the KD of Rab11 leads to PAMS formation, supporting the conclusion that defective apical secretion results in PAMS formation (Figure 4—figure supplement 3).

So overall, the manuscript is the description of a number of factors (related to the cytocortex-Rab network) whose depletion leads to PAMS. But there is very little mechanistic insights or sense of what is new.

The relation between Crb and the regulation of phosphoinositides, as well as the requirement of Crb to organize the apical secretion machinery have not been described before. We made this message clearer along the manuscript. Moreover, we expanded the evidence that support these conclusions by including several new experiments and quantifications that show that these interactions are important for the physiology and development of the larvae.

Reviewer #2:In this paper, the authors remove Crb (or other Crb complex protein) function in mid-third instar larval SGs and determine the consequences. What they observe is that many aspects of SG organization are unaffected; overall SG morphology, cell polarity, barrier function, as well as the localization of a number of membrane domain-specific proteins appear largely normal. Apical secretion of both an apical membrane protein (CD8-RFP) and glycoproteins (those detected by PCNA binding) are diminished, however, with the reduction in Crb or with the reduction of Sdt (a component of the Crb complex). Reduced Crb also results in decreases in apical accumulation of F-actin, βHeavy-Spectrin and MyoV (the myosin that has been implicated in secretion). Also observed are decreases in apical localization of three Rab proteins that have previously been implicated in apical secretion: Rab6, Rab11 and Rab30. The reduction in Crb also results in the invagination of the apical domain to form structures the authors refer to as PAMS (PI(4,5)P2 and Phospho-Moesin enriched apical membrane sacs). To make sense of the observed molecular changes, the authors also determined the effects of reducing βHeavy-Spectrin, MyoV, and the enzymes involved in controlling PI(4,5)P2 levels in cells: Skittles (a PIP kinase that makes PI(4,5)P2), Pten (which dephosphorylates PI(3,4,5)P3 to PI(4,5)P2) and PI3K (which phosphorylates PI(4,5)P2 to make PI(3,4,5)P3). The reduction in βHeavy-Spectrin and of MyoV also diminishes secretion and promotes formation of PAMs, although not to the same level as reduction in Crb. The authors also show that reduction of Pten activity as well as increases in P13K can reduce the formation of PAMs associated with reduced Crb function, as well as restoring the apical localization of Rab6 and Rab11. Based on these experimental findings, the authors propose the following model: Crb is required to maintain the proper amount and organization of the apical domain by stabilizing the apical cytocortex (through βHeavy-Spectrin and F-actin), that the Crb-βHeavy-Spectrin complex facilitates apical secretion in larval SGs by maintaining apical MyoV. The authors also conclude that Crb normally limits PI(4,5)P2 accumulation by limiting Pten activity (but not Pten accumulation since levels appear unchanged between controls and Crb knockdown).Essential revisions:1) Most of the phenotypes are not quantified and should be. When the authors say that PAM formation is reduced or increased or that secretion is reduced, we are relying on the limited number of images that can be shown and an impression the authors have of the data, rather than any quantification.

We provide quantifications and corresponding source data in the new manuscript. All statistical analyses are described in the Materials and methods section.

2) The authors make a good case that PI(4,5)P2 is required for the formation of PAMs, but they do not make a very good case that Crb works through Pten to increase PI(4,5)P2 levels. From the images provided, it is not at all clear that overall apical PI(4,5)P2 levels are any higher in the Crb knockdown than in WT control SGs, although the apical distribution does seem to change – less in the PM and more in PAMs. Reducing PI(4,5)P2 (which is known to bind and activate Moesin) does indeed appear to reduce PAM formation and increasing PI(4,5)P2 does appear to increase PAM formation but that does not indicate that Crb normally acts to limit Pten activity; it only says that high levels of PI(4,5)P2 are required for this loss of Crb phenotype. Indeed, the Pten overexpression phenotypes does seem to increase the apical domain area but it doesn't seem to create structures that have the same morphology as the PAMs. So, either change the conclusions/title of the manuscript or provide better evidence that PI(4,5)P2 levels are indeed higher with loss of crb.

We provide the quantification of PI(4,5)P_2_ levels by measuring the fluorescence intensity using a well-established probe for PI(4,5)P_2_ (see for example Jouette et al., 2019, *eLife*). Besides, by quantifying the distribution of an overexpressed Pten2 tagged with GFP we found that its distribution (the apical-to-lateral ratio) decreases in the Crb-deficient salivary glands (Figure 5I-K). Moreover, we extended our analyses to include the inositol polyphosphate 5-phosphatase Ocrl (Oculocerebrorenal syndrome of Lowe), that regulates PI(4,5)P_2_ homeostasis by dephosphorylating PI(4,5)P_2_. Our new results show that Ocrl localizes to the apical aspect of the salivary gland cells and that Crb is necessary for Ocrl apical localization. Accordingly, we have changed the title and conclusions of our manuscript to reflect that Crumbs is indeed involved in regulation of PI(4,5)P_2_ metabolism at different levels.

3) Skittles is not the only PIP kinase encoded in the Drosophila genome, perhaps explaining the relatively mild effects of skittles knockdown on the loss of Crb phenotypes. The authors should acknowledge the existence of this other protein when they note the relatively mild effects.

Thanks for the suggestion. As noted in the previous point, we now include Ocrl in our analyses.

4) Although the authors claim that loss of Crb does not affect AJs (based on TEMs and DE-cad immunostaining), it does appear from the images that levels of DE-cad found near the apical surface are notably reduced (perhaps there is also an increase in vesicular staining of DE-cad). Thus, the authors need to look at this more carefully and indicate that there is some reduction of the AJ-localized DE-cad, or clearly demonstrate that levels are unaffected.

In fact, *D*E-cad levels at the AJ seem to be reduced. At the same time, the intracellular localization is increased. We think that this phenotype is a manifestation of the defects in the trafficking machinery due to the loss of Crb. KD of *D*E-cad-GFP (knock-in allele) using the *gfp^RNAi^* effectively reduced the levels of *D*E-cad-GFP, but surprisingly, the distribution of Dlg-mTagRFP and the overall shape of the salivary glands are not affected. Therefore, we believe that the reduction of *D*E-cad-GFP levels at the AJ in the Crumbs deficient glands does not contribute to the defect in secretion, and that these results are redundant to the main message of our manuscript and therefore do not add more relevant information to the main conclusions. Therefore, we removed those results and relied on the EM analyses instead in order to streamline the manuscript.

**Author response image 2. respfig2:** 

5) There seem to be some PAMs with microvilli-like structures and other similarly large vesicular structures with no microvilli-like structures. Are the latter only found with Crb knockdown or are they also found in controls? If they are only found in the Crb knockdown, what are they proposed to be?

These structures resemble lysosomes and have also been observed in models for MVID (see for example Mosa et al., 2018). Other works have also suggested that impaired secretion results in enhanced lysosomal activity to degrade the accumulated material. We have included these points in our results (Figure 2—figure supplement 3 and subsection “Crb regulates apical membrane organization via the apical cytocortex”).

6) This may be challenging to determine, but it would be helpful to know if the Crb knockdown actually changes the total amount of apical surface area. Do control SG cells have the same amount of apical area if, in the Crb knockdown cells, one were to include the apical area found within the PAMs? This is related to the observation that the pleated septate junctions in the Crb knockdown SGs seem to be much more convoluted than those in WT (Figure 1—figure supplement 2), suggesting that there might be an increase in membrane in those domains. This might suggest that with loss of Crb, some of the apical membrane is being converted to basolateral membrane.

This is a wonderful idea but extremely difficult to tackle. We have included an estimation of the microvilli density in the Crb deficient cells, which do have less microvilli when compared with the controls (Figure 2M). Also, we added a quantification of the apical surface from confocal z-stacks and EM analyses (Figure 5, Figure 5—figure supplement1E, subsection “Crb regulates apical membrane organization via the apical cytocortex”, and subsection “Crb organizes the apical trafficking machinery by controlling apical PI(4,5)P_2_ levels”). Nevertheless, without a complete tomogram of the salivary gland cell, it is impossible to know the precise amount of membrane present in each domain (of course assuming that we will have developed the computational methods to extract the required information from such tomograms). Other super-resolution microscopy approaches could help to measure the amount of membrane in the different domains, but these methods are limited because of the considerable size of the salivary gland cells. Additionally, we have no way to answer whether some of the membrane contained in the large lysosomes in the Crb deficient glands was originally apical, lateral or just not trafficked to the respective domain and was targeted for degradation instead.

7) Importantly, is this study showing us that Crb is doing what it has been known to do – controlling the balance of apical versus basolateral domains and providing some clues as to the mechanism or is this study really uncovering a new and unrelated role for Crb? Either way the study is interesting, but exploring the first possibility is worthwhile since it has not been clear how Crb normally controls apical specification.

The functional relation between Crb and the regulation of phosphoinositides, as well as the requirement of Crb to organize the apical secretion machinery have not been described before. We have tried to make this message clearer along the manuscript. Moreover, we expanded the evidence that support these conclusions by including several new experiments and quantifications that show that these interactions are important for the physiology and development of the larvae. Regarding the balance of apical versus basolateral domains, to uncover the molecular mechanism(s) behind this role of Crb deserves to be addressed in a whole new project, as we do not even know whether these mechanisms are the same in different epithelia and different developmental stages. Nevertheless, we added our views on these points to the Discussion section, and we also suggest that our work could provide molecular targets for future studies understanding the physiological control of epithelial cell polarity (subsection “Crb organizes the apical trafficking machinery by controlling apical PI(4,5)P_2_ levels”). Also see our answer to point 1 of reviewer #3.

Reviewer #3:[…]1) All conclusions re: crb loss-of-function phenotypes (e.g. "defects in apical secretion are not due to an overall disruption of cell polarity", or "this strategy does not affect embryonic development") are currently based on knockdown experiments. These may only reduce – rather than abrogate – Crb expression, and do so at a currently undefined developmental point. Clonal mutation of crb in salivary glands and/or temporally controlled knockdown experiments would allow the authors to draw stronger conclusions about the roles of Crb (or lack thereof) in establishing vs maintaining overall epithelial polarity in the salivary gland, as well as their temporal requirement. Related to this, different fkh-Gal4s lines have been generated with distinct expression. Is the expression of the one used in this particular study really confined to salivary glands? Most lines are also expressed in other portions of the intestine.

The *fkh*-GAL4 line used in our study drives strong expression in the salivary glands and very weakly in a portion of the hindgut. However, in *crb* NULL embryo mutants the hindgut is not affected (Kumichel and Knust, 2014). Therefore, our Crb KD experiments are likely not affected by any miss-expression of *fkh*-Gal4.

**Author response image 3. respfig3:** 

With respect to our knock-down strategy, Crb-RNAi knock-down reduces Crb levels strongly (see Figure 1—figure supplement 1). Thus, our study is based more on a sensitized Crb background rather than a *crb* null-like setting. However, reduced Crb levels allow us to investigate processes regulated by this protein in more detail, as the complete loss of polarity is an extreme phenotype where it is difficult to analyze more subtle processes. To investigate how total ablation of the Crb protein regulates the morphology of salivary glands we decided to apply the deGradFP-mediated knock out approach. The long-term loss of Crb by employing the deGradFP-mediated knockout (Caussinus et al., 2012) results in polarity defects. We used the deGradFP in combination with the GAL80^ts^ system to control the induction of the nanobodies by shifting larvae to 29ºC. We used flies expressing Crb-GFP on the intracellular or the extracellular domain (Crb-GFP^intra^ or CrbGFP^extra^, both knock-in alleles). In this experimental setup, we observed that Crb-GFP^intra^ is no longer detectable already at 4 hours of incubation at 29ºC. Importantly, even after 7 hours at 29ºC, the glands do not show any gross morphological defects. Only after 12 hours at 29ºC we found defects in the morphology of the salivary glands (bracket in Author response image 4). It is important to note that the high efficiency of the nanobodies in the salivary gland may also be due to the very low turn-over rate of Crb at the apical membrane of the larval salivary glands (Bajur and Knust, 2019). As expected, the nanobodies do not affect the localization of Crb-GFP^extra^.

**Author response image 4. respfig4:** 

To complement these observations and analyze possible polarity defects at earlier times, we used animals expressing *D*E-cad-mTomato (a knock-in allele). In this case we employed the deGradFPmediated knockout of Crb-GFP without the GAL80^ts^. In fact, the larvae raised at 29ºC expressing the nanobodies and Crb-GFP^intra^ are smaller than their counterpart controls expressing the nanobodies with Crb-GFP^extra^. In these animals, the glands appear severely misshaped and disorganized (not shown). However, if we maintained the larvae at 19-20ºC to limit the GAL4 activity, it was possible to obtain larvae with normal looking salivary glands. Under these conditions, we monitored the effects of the deGradFP-mediated knockout by incubating early third instar larvae (feeding animals not wandering) for 4 or 7 hours at 29ºC. As shown in the Z-max projections in Author response image 5, after 7 hours Crb-GFP^intra^ is not detectable at the apical membrane. Despite this, *D*E-cad-mTomato is still localized in a similar pattern as in the controls.

**Author response image 5. respfig5:** 

Taken together, the complete loss of Crb in differentiated cells is fatal and a result of many cellular processes affected. Our “milder” *crb* phenotype allowed us to identify the subtler defects in molecular networks not linked to Crb before, yet important for the physiology of the animal.

2) The authors explore physiological aspects of Crb-mediated salivary gland secretion very superficially. This seemed to me a missed opportunity. If they wish to make any statements about Crb regulating the "physiological activity of the salivary gland" or promoting "efficient apical secretion of glycoproteins" (as currently stated in the abstract), the authors need to show effects on endogenous secretion – for example by visualizing the effect of crb knockdown on a protein normally secreted by the salivary gland. Related to this, the authors assume but do not show a reduction in food intake following salivary gland-specific crb knockdown. Is food intake reduced following crb knockdown? Does Pten knockdown rescue the feeding and developmental timing defects of crb knockdown larvae? Finally, what happens to Crb expression/localisation and/or downstream targets around the saliva to glue switch? In other words, is Crb permissive or instructive in the context of salivary gland secretion?

We analyzed whether the rescue of secretion by Pten knockdown also rescues the pupation as well as the feeding of the larvae (Figure 6). Our new results support the conclusion that defects in apical secretion in the salivary glands are mediated by an increase in PI(4,5)P_2_, as Pten knockdown suppresses the delay in pupation, while the Pi3K knockdown delays pupation even more than the Crb knockdown alone (Figure 6W). Additionally, we estimated the food intake by adapting the protocol reported in Deshpande et al., 2014, a total of 3,060 larva were used for this experiment. We found that Crb knockdown tends to reduce the food intake, although not statistically significant (Figure 6V). Importantly, the Pten knockdown increases the food intake, even when Crb is silenced simultaneously (Figure6V).

Regarding the localization of Crb during the transition to glue secretion, the protein remains apical during these stages (see Author response image 6).

**Author response image 6. respfig6:** 

We recognize that it will be ideal to follow a protein that is normally secreted by the salivary glands. Unfortunately, no endogenous cargo has been characterized at the developmental stages we performed our analyses. In fact, glue secretion occurs almost 2 days later from the stages we analyzed (Video 1 and Video 2). We revised transcriptomic data from FlyBase and found possible gene candidates that may encode secreted proteins including mucins, lysozymes and chitinases (the ORF Saliva suggested by the reviewer #1 is predicted to be a transmembrane transporter of sugar, not a secreted protein). However, none of the mucins, lysozymes and chitinases have been characterized, and there are no available antibodies or insertion lines (i.e. MiMIC) that will allow us to generate GFP tagged proteins easily. Thus, establishing, characterizing and validating these possible secreted proteins is out of the scope of our manuscript.

As noted also by the reviewer #1, the PNA-GFP probe that we used as a proxy for secretion is not well characterized. Therefore, we screened other lines used to track apical secretion and decided to replicate all experiment made with the PNA-GFP probe using instead a construct containing the chitin binding domain of Serpentine tagged with GFP (UAS-SerpCBD-GFP). This construct is a well-established apical secreted cargo and has been used as a proxy to analyze apical secretion (Luschnig et al., 2006; Kakihara et al., 2008; Förster et al., 2010; Petkau et al., 2012; Dong et al., 2013; Dong et al., 2014; Bätz et al., 2014). Using this tool, we have confirmed the results obtained with the PNA line, and thus exchanged all figures for the ones obtained with SerpCBD-GFP (Figure 1, Figure 1—figure supplement 1, Figure 3, Figure 3—figure supplement 2, Figure 6).

3) The authors should try to rule in/out increased endocytosis (rather than reduced apical secretion) as one of the reasons for the observed phenotypes; it seems to me that at least some of the Rab targets are involved in recycling as well as secretion. Also, the appearance of these novel intracellular microvilli-containing sacs and the concurrent reduced density of microvilli on the cell surface may both also be consistent with increased endocytosis; I am not an expert in this but, as far as I know, microvilli are not "trafficked" to the membrane in secretory compartments? Related to this, it may be informative to test the Rab signature of PAM sacs in both Crb/β-Spectrin knockdowns – might this shed light on the origin of these compartments?

We explored the role of endocytosis in three ways. By using the temperature sensitive allele of dynamin (*Shi^ts^*), by ex vivo incubation of glands with Dynasore (cell-permeable inhibitor of dynamin) and by overexpressing dominant active (DA) or dominant inactive (DN) forms of Rab5.

For the first approach we generated the following flies:

- *Shi^ts^; UAS-crb[RNAi]*

- *Shi^ts^;; fkh-GAL4>UAS-SerpCBD-GFP*

- *Shi^ts^;; fkh-GAL4>UAS-PLC*d*-PH-EGFP*

All stocks were kept at 19-20ºC, as adult’s viability decays very fast at temperatures >24ºC. To analyze the role of dynamin in the Crb phenotypes we set crosses (indicated below), and kept them at 19-20ºC. The eggs were collected overnight and then transferred to 28ºC (*Shi^ts^* larvae die at 29ºC), where they were incubated for 2 days. It is important to note that generation of these stocks as well as testing the different iterations for this experiment required several months. The salivary glands were dissected on the third day on ice-cold Grace’s medium and then analyzed by live imaging:

For SerpCBD-GFP:

Control *w*: w**;; *fkh-GAL4>UAS-SerpCBD-GFP X w*/*⦢

*crb^RNAi^ w*: w**;; *fkh-GAL4>UAS-SerpCBD-GFP X w*/*⦢; *UAS-crb[RNAi]* Control *Shi^ts^: Shi^ts^*;; *fkh-GAL4>UAS-SerpCBD-GFP X Shi^ts^/*⦢

*crb^RNAi^ Shi^ts^: Shi^ts^*;; *fkh-GAL4>UAS-SerpCBD-GFP X Shi^ts^/*⦢; *UAS-crb[RNAi]*

For UAS-PLCd-PH-EGFP:

Control w*: w*;; fkh-GAL4>UAS-PLCd-PH-EGFP X w*/⦢

crb^RNAi^ w*:w*;; fkh-GAL4>UAS-PLCd-PH-EGFP X w*/⦢; UAS-crb[RNAi]

Control Shi^ts^:Shi^ts^;; fkh-GAL4>UAS-PLCd-PH-EGFP X Shi^ts^/⦢

crbRNAi Shits: Shi^ts^;; fkh-GAL4>UAS-PLCd-PH-EGFP X Shi^ts^/⦢; UAS-crb[RNAi]

Representative images of the results obtained are shown in Author response image 7. In summary, the secretion defect evaluated with SerpCBD-GFP seems to be ameliorated in the *Shi^ts^* background. However, the formation of PAMS is not completely suppressed in the *Shi^ts^* background. Of note, these results are not modified by further incubating the dissected glands for 15 min at 37ºC (to make sure that *Shi^ts^* is completely inhibited).

**Author response image 7. respfig7:** 

For the second approach, ex vivo incubation of glands with Dynasore, all analyses were inconsistent and no solid conclusion about the role of dynamin could be reached.

For the third approach, we analyzed the formation of PAMS induced by Crb KD in glands co-expressing Rab5 DA or Rab5 DN. However, we obtained inconsistent results, as we found enhancement or suppression of the PAMS phenotype with either Rab5 DA or Rab5 DN. This could be due to pleiotropic effects of dominant versions of Rab proteins, which tend to titer effectors and regulators shared with other Rab proteins (e.g. Rab4).

These results suggest that dynamin-mediated endocytosis could be involved in the defects observed in the Crb-deficient glands, and accordingly we added these pointS to our Discussion section. However, it is beyond the scope of this manuscript to describe the precise role of endocytosis in the phenotypes generated by loss of Crb in the salivary glands. Indeed, we think that these observations deserve a systematic analysis that will make the current manuscript too large to be comprehensible. Regarding the second part or the question, about the formation of the intracellular vesicles containing microvilli. As we describe in our manuscript, these inclusions resemble those found in the intestine of MVID patients. There are published works that provide genetic evidence supporting the hypothesis that defects in apical transport results in formation of similar microvilli-inclusion bodies (Feng et al., 2017; Knowles et al., 2014; Knowles et al., 2015; Mosa et al., 2018; Müller et al., 2008; Ruemmele et al., 2010; Sato et al., 2007; Schneeberger et al., 2015; Sidhaye et al., 2016; Weis et al., 2016).

The mechanism by which these inclusions form is unknown, but the reviewer is referred to a recent work (Mosa et al., 2018) in which intestinal organoids were used to analyze the formation of the microvillus inclusions. Using live imaging, Mosa et al. show that these inclusion bodies are dynamic, and can form inside the cytoplasm, or derive from the apical or basolateral membranes, as well as to fuse with the apical or basolateral membrane, or collapse inside the cell. The time scale for these inclusions from forming to the moment of detachment or collapse is several hours. Their results suggest that indeed, microvilli might be assembled inside the cytoplasm and trafficked to the plasma membrane. Nevertheless, we don’t think that intracellular formation of microvilli occurs in our experimental model. Instead, we have observed some large intracellular vesicles, but these are probably PAMS that detached or cleaved away from the apical membrane. However, we have been unable to follow these events in our live imaging experiments, as these may happen very sporadically and over a long period of time (hours). We have included this additional information in our Discussion section.

Finally, regarding the Rab signature of the PAMS, we could not define a Rab signature for the PAMS. As shown in the Figure 4—figure supplement 1, no specific Rab protein accumulated in a compartment similar to the one marked by the PI(4,5)P_2_ reporter or phospho-Moesin.